



# Impact of atmospheric turbulence on performance and loads of wind turbines: Knowledge gaps and research challenges

Branko Kosović[1]★, Sukanta Basu[2]★, Jacob Berg[3], Larry K. Berg[4], Sue E. Haupt[5], Xiaoli G. Larsén[6], Joachim Peinke[7], Richard J. A. M. Stevens[8], Paul Veers[9], and Simon Watson[10]

[1]Johns Hopkins University, Ralph O'Connor Sustainable Energy Institute, USA
[2]University at Albany, Atmospheric Sciences Research Center, USA
[3]DHI, Denmark
[4]Pacific North West National Laboratory, USA
[5]NSF National Center for Atmospheric Research, USA
[6]Technical University of Denmark, Denmark
[7]University of Oldenburg, Germany
[8]University of Twente, Netherlands
[9]National Renewable Energy Laboratory, USA
[10]Delft University of Technology, Netherlands
★These authors contributed equally to this work

**Correspondence:** Branko Kosović (branko.kosovic@jhu.edu), Sukanta Basu (sbasu@albany.edu)

**Abstract.** Wind energy harvesting from the atmosphere takes place in the atmospheric boundary layer. The boundary layer shear and buoyancy create three-dimensional turbulent eddies spanning a range of scales that form a continuous forward cascade of kinetic energy to the smallest scales of motion where energy is dissipated. Large-scale atmospheric circulations modulate the boundary layer turbulence, characterized by coherence and intermittence. As wind turbines have grown in size
and the integrated control of both turbines and wind farms has spanned greater distances, the relationship between the scales of atmospheric turbulence and the design and operation of wind energy facilities has entered new territory. The boundary layer turbulence impacts both wind turbine power production and turbine loads. Optimizing wind turbine and wind farm performance requires understanding how turbulence affects both wind turbine efficiency and reliability. While the characteristics of atmospheric boundary layer turbulence have been observed and studied in detail over the last decades, there are still significant
gaps in understanding the impact of turbulence on wind power resources and wind farm operations. This paper outlines the current state of turbulence research relevant to wind energy applications and points to gaps in our knowledge that need to be addressed to effectively utilize wind resources.

## 1   Introduction

Most human activity happens in the atmospheric boundary layer (ABL), which extends a few hundred meters to a couple of
kilometers above the surface of the Earth. The flow in the ABL is characterized by turbulent eddies and vortices that contribute to the exchanges of momentum, heat, moisture, and other constituents between Earth's surface and the atmosphere. Wind energy harvesting also takes place in this layer. The wind energy resource at a location is commonly assessed by estimating





hub-height wind speed, considering wind speed averaged over ten-minute intervals (e.g., Global Wind Atlas, Davis et al., 2023). However, shorter fluctuations in wind speed and direction due to turbulence can affect wind turbine power production
and loads, directly affecting wind turbine and wind farm operational efficiency, and therefore, the levelized cost of wind energy (e.g., Yang et al., 2021a).

Turbulence affects the efficiency of wind turbine power generation resulting in fluctuating power output. It also shortens their lifespan by inducing dynamic loads. There are two types of loads acting on a wind turbine, aerodynamic rotor loads and loads acting on the tower. Turbulence impacts aerodynamic loads that result form airfoil lift and drag forces or corresponding normal
and tangential forces responsible for the rotation of a rotor and the bending of blades. Combined effects of wind shear over the rotor plane and turbulence result in blade bending and impact particularly blade root fatigue loads. Accurate characterization of turbulence in the environment where wind farms are being developed is therefore essential for effectively designing wind turbines and wind farms. While the characteristics of atmospheric turbulence have been extensively observed and studied over the last decades, there are still significant gaps in understanding the impact of turbulence on wind power resources and wind
farm operations.

During 2023, worldwide deployment of wind energy reached 1 TW of capacity (Global Wind Report, 2024). Wind capacity has quadrupled over the last ten years, and this trend will continue and possibly accelerate. According to the Global Wind Energy Council (GWEC) projection, another 680 GW of wind power capacity will be added globally between 2023 and 2027, 490 GW onshore and 130 GW offshore. The consequence of such rapid growth is that wind turbines are increasingly
deployed in environments not characterized well by simple analytical formulations. Turbine heights have grown beyond the surface layer where the log law is a good representation of the wind profile. The newly added wind power capacity will be deployed in environments that may not have been considered. For example, offshore, where the size limits have not been reached yet, blade tips now reach 300 m above the surface, beyond the frequently shallow marine ABL. Utility-scale turbines are, therefore, being exposed to conditions including turbulence levels that are not well characterized by current standards.
Widespread deployment of wind farms in complex boundary layer environments requires a more nuanced characterization of flows and turbulence for wind turbine and wind farm design. Therefore, expected growth in wind energy deployment presents a scientific challenge to better understand turbulence impacts on power output and turbine loads. Our review builds on and extends previous fundamental studies (e.g., Hölling et al., 2014; Meneveau, 2019) aiming to identify pertinent research topics that would inform new design standards.

This paper reviews the current scientific understanding of turbulence in the ABL and its resulting impacts on wind farm and wind turbine power production and loads. When considering turbulence impact on wind energy we adopt a broad view of atmospheric turbulence that is not focused only on irregular, chaotic, three-dimensional, small-scale motions but also includes larger-scale atmospheric forcings and phenomena that modulate turbulent flows in the ABL. We start by defining fundamental concepts related to ABL turbulence relevant to wind energy applications and then follow with an overview of boundary
layer phenomena and processes that affect the structure and properties of turbulence. We then present a review of turbulence impacts on power production and loads. Finally, we present an analysis of gaps in the scientific understanding of turbulence characteristics and their impacts that must be addressed to enable the reliable operation of utility-scale turbines and wind farms.





## 2 A Primer on Atmospheric Energy Cascade

Motions in Earth's atmosphere span a range of scales from a few thousand kilometers down to sub-meter scales. The largest
atmospheric motions supporting earth-wide winds and transporting heat from the tropics to polar regions form three cells:
Hadley, Ferrel, and Polar, spanning between the equator and the poles on both hemispheres.

The turbulent motions exhibit three distinct kinetic energy scaling ranges, starting from the largest planetary waves and
synoptic scales through mesoscale to microscales in the ABL depicted in Fig. 1. Quasi-two-dimensional planetary waves or
Rossby waves extend longitudinally over thousands of kilometers. Rossby waves are a consequence of Earth's rotation (Rossby
and Collaborators, 1985; Platzman, 1968), are embedded within global circulations. Large-scale pressure systems and the jet
stream are associated with Rossby waves that drive synoptic-scale cyclones on the order or thousand kilometers, responsible
for what we experience as weather. Weather evolves within the troposphere, which extends from the surface to the lower strato-
sphere (approximately 10-15 km). Since the atmosphere is relatively shallow compared to its horizontal scale at synoptic scale,
quasi-two-dimensional, quasi-geostrophic turbulence is a result of an inertial enstrophy cascade resulting in a $-3$ slope of the
horizontal kinetic energy spectrum with spectral energy versus wavenumber/frequency in log-log coordinates (Charney, 1971;
Herring, 1980; Pedlosky, 1987; Tulloch and Smith, 2006). Below several hundred kilometers to a few kilometers, atmospheric
mesoscale motions are affected by surface heterogeneities, including land/sea breezes, squall lines, mesoscale convective cir-
culations, and thunderstorms. Mesoscale turbulence, defined in the same coordinates by a $-5/3$-slope kinetic energy scaling,
results from a combination of downscale and inverse energy cascade (Lindborg, 2006; Kitamura and Matsuda, 2010; Lovejoy
and Schertzer, 2013) is commonly observed at scales below 600 km (Nastrom and Gagne, 1985). Finally, microscale motions,
characterized by a fully developed, three-dimensional turbulence, range from a couple of kilometers to a sub-meter scale.
Three-dimensional turbulence, driven by shear or buoyancy at the microscale, can occur at any altitude within the boundary
layer. Turbulence is a defining characteristics of the ABL and it follows a $-5/3$ scaling in the inertial range (Elderkin, 1966;
Busch and Panofsky, 1968; Kaimal et al., 1972; Kaimal, 1973, 1978). Figure 1 depicts the atmospheric kinetic energy spectrum,
including three distinct scaling regions.

While we primarily focus on the effects of atmospheric turbulence on the wind turbine and wind farm performance, we will
also address the impact of larger-scale atmospheric motions related to extreme events that affect the characteristics of ABL
turbulence that impact wind farms.

The jet stream, as an example of a large-scale atmospheric phenomenon, is potentially a significant wind resource (Archer
and Caldeira, 2009), however, in addition to challenges presented by harvesting high-altitude wind, extractable energy may be
limited (Miller et al., 2011). Significant wind energy is associated with synoptic-scale tropical cyclones, including hurricanes
and typhoons. However, these large rapidly rotating storm systems can result in surface winds significantly exceeding wind
turbine design wind speeds. Hurricanes or typhoons frequently spawn mesoscale supercells, i.e., rotating thunderstorms that
can create tornadoes. While tornadoes most frequently occur in North America they are observed worldwide. Tornadoes are
characterized by an extreme low-pressure funnel core surrounded by an eyewall where wind speeds can exceed 180 km/h
and reach up to 300 km h$^{-1}$ and, therefore, could exceed wind turbine survival wind speed. Thunderstorms can also cause



damaging downburst winds (e.g., Nguyen et al., 2013). Downslope wind storms are another mesoscale process that results in strong winds, breaking waves, and increased turbulence intensity (Pehar et al., 2019). Mesoscale convective circulations and associated cloud streets, frequently observed during cold air outbreaks over warmer bodies of water, span a range of scales from

turbulent boundary layer updrafts to tens of kilometers wide convective cells and helical rolls extending hundreds of kilometers. Such structures can bridge the gap first observed by Van der Hoven (1957) between mesoscales and ABL turbulence eddies.

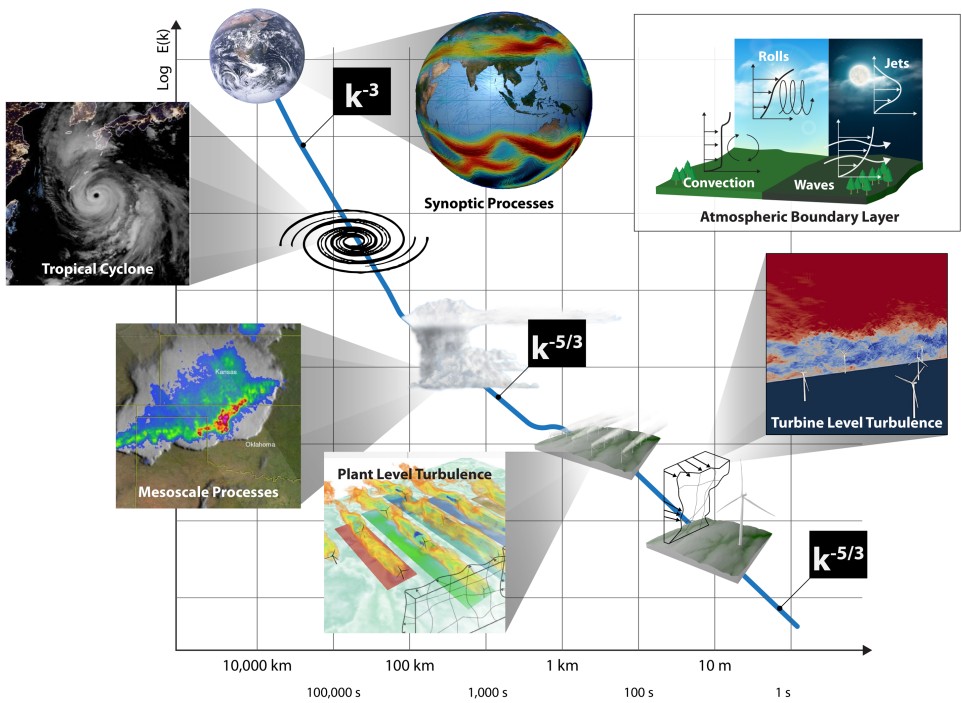

**Figure 1.** Cascade of kinetic energy in the atmosphere. The three main scaling regions include largest atmospheric scales with $k^{-3}$ spectral slope, mesoscale range with $k^{-5/3}$ slope, and small atmospheric scales within Kolmogorov, $k^{-5/3}$, inertial range.

# 3   Atmospheric Boundary Layers

In contact with earth's surface atmospheric flow is impacted by surface forcings: surface drag, heating or cooling, evaporation and transpiration. Surface forcings in the form of shear and buoyancy result in turbulent flow that characterizes atmospheric

boundary layer. Turbulent eddies in the ABL range in size from energetic eddies spanning a few kilometers, i.e., the depth of the boundary layer, to millimeter-scale eddies where energy is dissipated. ABL turbulence does not evolve in the isolation from the rest of the atmosphere, but instead, it is modulated by a range of scales atmospheric motions and phenomena.



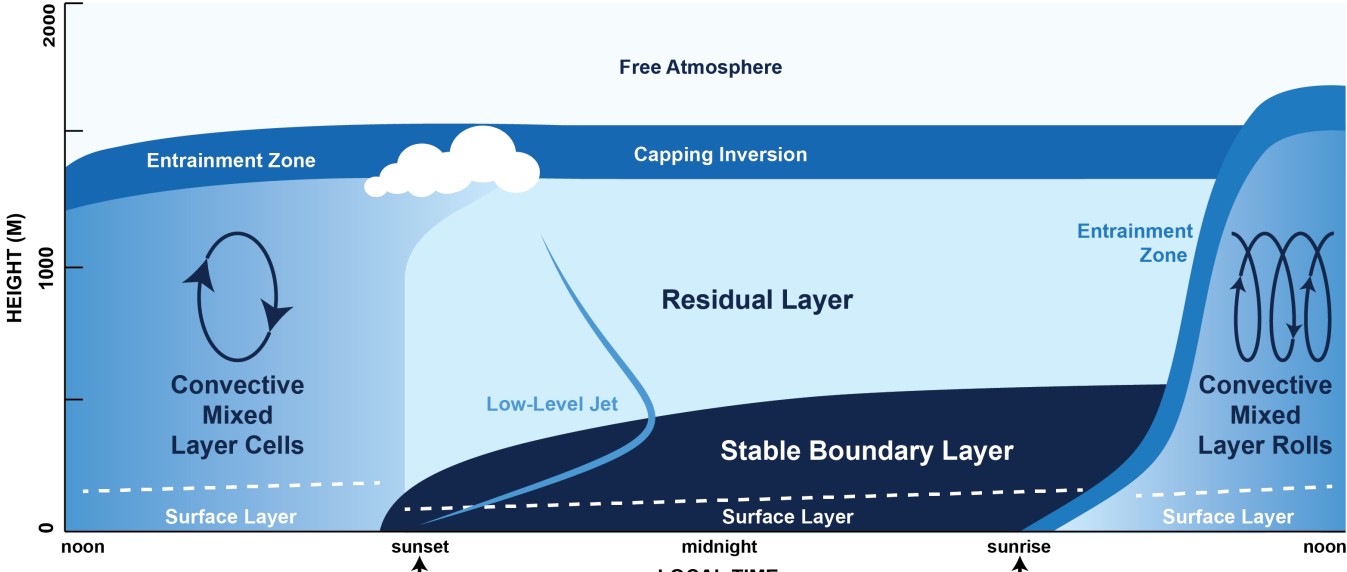

**Figure 2.** The diurnal cycle of the atmospheric boundary layer.

The main drivers of flows in the atmosphere, including the ABL, are large-scale pressure gradients, the apparent Coriolis force, surface heating or cooling, advection, terrain effects, and surface heterogeneities. ABL evolution typically follows a

diurnal cycle due to differences in radiative transfer characteristics during daytime and nighttime. The diurnal cycle is more pronounced over land than over water. In polar regions, however, the diurnal cycle is absent or weak. The diurnal cycle is characterized by faster daytime heating and nighttime cooling of the surface than changes in the overlying atmosphere. The resulting temperature differences between the Earth's surface and the atmosphere result in unstable (convective) daytime and stable nighttime boundary layers (see Fig. 2). During daytime, the convective boundary layer can grow to a depth of a few

kilometers. At the top of convective boundary layer capping inversion develops, characterized by a strong temperature gradient defining entrainment zone through which exchanges between free atmosphere and ABL are mediated. In contrast, when surface cooling happens at nighttime, convective structures collapse, and a stably-stratified boundary layer develops on the order of 10's or 100's of meters. Between the capping inversion above and the stbaly-stratified layer below a residual of a daytime layer persists.

Regardless of atmospheric stability, turbulence in the ABL is created by shear, while buoyancy can produce or destroy turbulence depending on stability conditions. Under unstable conditions, when the surface is warmer than the atmosphere buoyancy contributes to the formation of turbulent eddies. As eddies grow, they create a well-mixed layer capped by a temperature inversion. Under stable conditions, when the surface cools radiatively faster than the atmosphere or when warm air is advected over a cooler surface, buoyancy suppresses turbulence, resulting in reduced mixing and development of a stably-stratified ABL.

The interplay of shear and buoyancy creates a spectrum of turbulent structures, resulting in an inertial down-scale cascade of kinetic energy to smaller eddies. The turbulent kinetic energy is also advected by wind and transported through space by ve-





locity and pressure fluctuations. Ultimately, the kinetic energy is dissipated into heat at millimeter scales due to viscosity. The magnitude of shear and buoyancy depends on both local boundary layer conditions and large-scale atmospheric conditions. Wind energy applications are focused on cases when the wind speed is great enough (e.g., greater than 3 ms$^{-1}$ at hub height)
for wind turbines to generate electricity. These cases are generally associated with significant shear production of turbulence which impacts wind turbine power and loads.

Studies of idealized, canonical ABLs have been used extensively in the research community. They are defined as barotropic turbulent flows over horizontally homogeneous surfaces. Under steady geostrophic forcing, when the Coriolis force balances the large-scale pressure gradient, quasi-equilibrium canonical boundary layers develop and evolve due to heat (and possibly
moisture) exchanges with the surface. This approach has helped us understand the structure of ABLs, their diurnal evolution, and the development of parameterizations in large-scale models. However, in reality, ABLs are embedded in continuously evolving large-scale atmospheric flows. Frequently, large-scale motions evolve at a time scale comparable to the turbulent time scale, resulting in non-equilibrium conditions where the turbulence production is not balanced by dissipation. Thus, real-world cases can have transient events that can result in more extreme turbulence levels and significantly impact turbine performance.
Several meteorological quantities are commonly used to quantify various characteristics of ABL flows. Next, we summarize a few of them to enhance the readability of this paper. A detailed description of these quantities is beyond the scope of this paper. The reader is encouraged to refer to Panofsky and Dutton (1983), Stull (1988), Arya (2001), Wyngaard (2010) and Morales et al. (2012) for details.

### 3.1 Mean Quantities of ABL flows

In this paper, mean velocity components along the longitudinal, lateral, and vertical directions are represented by $\overline{u}$, $\overline{v}$, and $\overline{w}$, respectively.[1] The mean (horizontal) wind speed is: $U = \sqrt{\overline{u}^2 + \overline{v}^2}$. Henceforth, the wind speed at hub height will be denoted as $U_H$. The mean potential temperature, a temperature a parcel of fluid would attain when adiabatically brought to a reference pressure (e.g., standard surface pressure, usually 1,000 hPa), is $\overline{\theta}$. The potential temperature represents the combined stratification of temperature and pressure. At a height $z$ above the surface, $\overline{\theta}(z)$ can be approximated by $\overline{T}(z) + \Gamma z$; where the
mean air temperature is denoted as $\overline{T}$ and $\Gamma$ is the dry adiabatic lapse rate ($\approx -9.8 \times 10^{-3}$ K m$^{-1}$).

As mentioned, atmospheric stability depends on wind shear ($S$) and static stability ($N$). They can be computed as follows:

$$S = \sqrt{\left(\frac{\partial \overline{u}}{\partial z}\right)^2 + \left(\frac{\partial \overline{v}}{\partial z}\right)^2}, \tag{1a}$$

$$N = \sqrt{\frac{g}{\overline{\theta}_\circ}\left(\frac{\partial \overline{\theta}}{\partial z}\right)}. \tag{1b}$$

---

[1] In the atmospheric science literature, a different convention is followed. There, $\overline{u}$ and $\overline{v}$ represent zonal and meridional velocity components, respectively.





In the atmospheric science literature, $N$ is the Brünt Väisäla frequency. The gradient Richardson number ($Ri_g$) is a popular measure of atmospheric stability and is defined as:

$$Ri_g = \frac{S^2}{N^2}. \tag{2}$$

It can quantify the relative importance of shear production and buoyancy production/destruction. When $Ri_g \approx 0$, the atmospheric layer is considered near-neutral. Positive (negative) values of $Ri_g$ signify stable (unstable) conditions. It is generally accepted that the boundary layer flow is quasi-laminar when $Ri_g$ exceeds unity.

Given the sparsity (and generally coarse vertical resolution) of profile measurements, estimating the vertical gradients of meteorological variables in a field experimental setting is challenging. As a viable alternative, one approximates the vertical gradient of any variable $\chi$ as: $\partial \chi / \partial z \approx \Delta \chi / \Delta z$. Thus, a widely accepted bulk parameterization for the Richardson number is defined as follows:

$$Ri_B = \left(\frac{g}{\theta_0}\right) \frac{\Delta \overline{\theta} \Delta z}{(\Delta \overline{u})^2 + (\Delta \overline{v})^2}. \tag{3}$$

If the lower level of the gradient is assumed to be at the surface ($z \approx 0$), one can further simplify Eq. (3) as:

$$Ri_{Bs} = \left(\frac{g}{\theta_0}\right) \frac{(\overline{\theta} - \Theta_s) z}{(\overline{u}^2 + \overline{v}^2)}; \tag{4}$$

where $\theta_0$ and $\Theta_s$ denote reference and surface potential temperatures, respectively. It is further assumed that the wind speed vanishes at the surface. The numerator of $Ri_{Bs}$ contains the term $(\overline{\theta} - \Theta_s)$, which represents the (potential) temperature difference between air ($\overline{\theta}$) and the underlying land/sea-surface ($\Theta_s$); it is commonly called air-surface temperature difference or ASTD. With weak to moderate wind speeds, positive (negative) ASTD leads to stable (unstable) conditions.

Under unstable conditions, both shear and buoyancy effects cause turbulent mixing. In contrast, turbulence is generated by shear and destroyed by (negative) buoyancy in stable conditions. This competition leads to significantly reduced turbulent mixing under stable conditions. In fact, under very stable (strongly stratified) conditions, the flow tends to become quasi-laminar, and turbulence can become globally intermittent, as will be discussed later.

### 3.2 Turbulence Quantities of ABL flows

The three components of velocity variances are denoted as $\sigma_u^2$, $\sigma_v^2$, and $\sigma_w^2$, respectively. Turbulence kinetic energy (TKE; $\overline{e}$) is computed as:

$$\overline{e} = \frac{1}{2}\left(\sigma_u^2 + \sigma_v^2 + \sigma_w^2\right), \tag{5}$$

which takes into account all three components of the wind. The turbulence intensity is more commonly used in the engineering community. The turbulence intensity can be defined in different ways, along the streamwise direction: $\sigma_u/U$, or accounting for the full horizontal wind: $\sqrt{\sigma_u^2 + \sigma_v^2}/U$. The covariances $\overline{u'w'}$, $\overline{v'w'}$, and $\overline{u'v'}$ represent the components of momentum fluxes (closely related to Reynolds stress components); the sensible heat flux is denoted by $\overline{w'\theta'}$.



As an alternative to $Ri_g$ and $Ri_B$ (or $Ri_{Bs}$), atmospheric stability near the surface can also be quantified by the ratio $z/L$, where $L$ is called Obukhov length (Obukhov, 1946, 1971). The Obukhov length includes the ratio of the third power of the surface friction velocity computed from turbulent momentum fluxes ($\overline{u'w'}$, $\overline{v'w'}$) and surface heat flux ($\overline{w'\theta'}$). In convective ABLs, the absolute value of $L$ is defined as the height above which buoyancy effects begin to dominate over the shear effects. Under neutral conditions, $z/L = 0$; for stable conditions, $z/L$ is positive, and for unstable conditions, $z/L$ is negative.

Direct measurement of $L$ requires advanced instrumentation (e.g., sonic anemometers, scintillometer), which are rarely available within or close to wind farms, especially at offshore locations. However, the estimation of $Ri_{Bs}$ only requires measurements of mean meteorological variables at a single elevation and an estimate of surface temperature. Thus, in many field studies, $Ri_{Bs}$ has been computed from observed data, and in turn, $L$ has been indirectly inferred utilizing empirical relationships proposed by Grachev and Fairall (1997) and others. The intrinsic limitations of this indirect approach have been discussed in the literature by Argyle and Watson (2014) and others.

### 3.3 Whither Neutral Conditions?

Contemporary turbine design standards assume inflow turbulence to be neutrally stratified (i.e., $Ri_g \approx 0$, $z/L \approx 0$), but neutral conditions are relatively rare in the atmosphere. In the atmosphere, exact neutral stratification (i.e., $Ri_g = 0$, $z/L = 0$) is mainly associated with the transition between convective conditions and stable stratification, although near-neutral conditions can arise under several scenarios: (a) very windy conditions (i.e., shear generation completely dominates over buoyancy effects); (b) sometimes under cloudy conditions (Oke, 1987; Petersen et al., 1998); and (c) when ASTD is approximately equal to 0 (i.e., virtually negligible buoyancy effects). Over land, this last scenario persists for brief periods during morning and evening transitions (around sunrise and sunset, respectively). Several years ago, Kelley et al. (2006) analyzed 4,676 hours of observational data from Lamar, Colorado, and found that, only 0.3% of the time, the near-surface atmospheric layer (3–116 m above ground, to be specific) was neutral. More recently, Haupt et al. (2019) analyzed observational data from Lubbock, Texas. They reported stable and unstable conditions to dominate at this site (see the panel (a) in Fig. 3). Near-neutral conditions are also infrequent offshore, as indicated by the frequency of $Ri_B$ near zero (see the panels (b) and (c) in Fig. 3).

### 3.4 Wind Speed and Direction Profiles

Mean wind speed profiles in the ABL exhibit a wide variety of shapes. Sometimes, the profile is approximately logarithmic in nature, while at other times, it can take on significantly different forms. Under certain meteorological conditions, e.g., convective, or well-mixed conditions, wind speeds can be relatively uniform with height above the surface layer. Alternately, stable stratification promotes 'jet' shapes with low-level wind maxima (discussed later in detail). Some of these shapes can be seen in Fig. 4 (a-f).

The wind industry commonly uses a power law to represent ABL for heights across the rotor:

$$U_z = U_H \left( \frac{z}{z_H} \right)^\alpha, \tag{6}$$

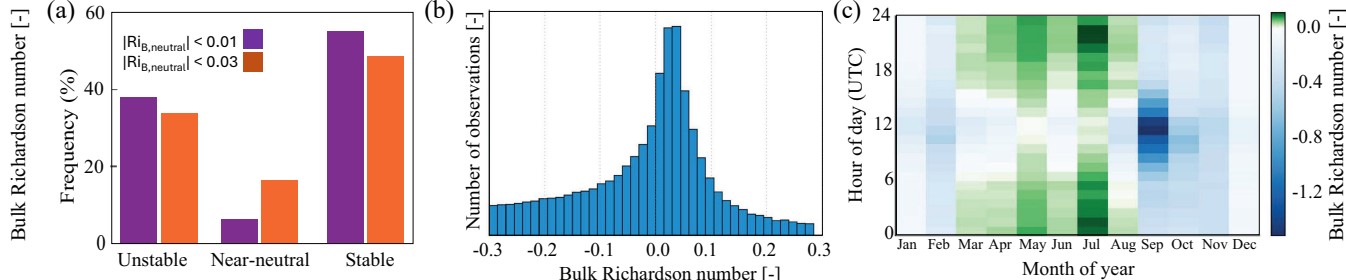

**Figure 3.** (a) Histograms of unstable, near-neutral, and stable conditions at the SWiFT facility, Lubbock, Texas for 730 days between 2012 and 2014. The histogram shows that stable and unstable conditions are dominant at this site; neutral conditions are not common (source: Haupt et al., 2019, © Copyright, 1 December 2019, American Meteorological Society (AMS)). (b) Histogram of bulk Richardson number ($Ri_B$) calculated in the atmospheric layer between 21 m and 90 m. (c) Diurnal and seasonal variation of (median) $Ri_B$ indicates pronounced seasonal cycle. During spring and summer, stable conditions are prevalent, whereas unstable conditions dominate during autumn and winter (source: Kalverla et al., 2017, published under CC BY 3.0 license). Observational data from the IJmuiden tower over the North Sea (85 km from the Dutch coastline) were utilized for the analyses in the panels (b) and (c).

where $U_z$ is the estimated wind speed at height $z$, and $z_H$ is the hub-height. $\alpha$ is the so-called shear exponent or the Hellman exponent. It is well-established in the literature that $\alpha$ strongly varies with atmospheric stability and surface roughness (e.g., Frost, 1947; Sisterson and Frenzen, 1978; Irwin, 1979; Storm and Basu, 2010). Thus, $\alpha$ is expected to exhibit diurnal, seasonal, and inter-annual variations. Using observations from tall towers over the United States (US) Great Plains, Schwartz and Elliott (2006) found that $\alpha$ values are substantially larger at night and smaller during the day. The value of $\alpha$ may also depend on

advection and non-equilibrium conditions, which are common in the coastal zones. Although the sum over power law functions with different exponents does not mathematically lead back to a power law function, constant values of $\alpha$ are often used in wind energy projects, with $\alpha = 1/7 (\approx 0.14)$ and 0.2 being the most commonly used values in wind energy applications.

Wind directional shear (also called veer) is commonly estimated as:

$$\beta = d(z) - d(z_r), \tag{7}$$

where $d(z)$ and $d(z_r)$ are wind turning angles at heights $z$ and $z_r$, respectively. During convective conditions, $\beta$ is typically minimal within the entire boundary layer. However, at night, when the atmosphere is stable, average turning angles of up to 40° (between 20 m and 200 m) have been reported (van Ulden and Holtslag, 1985; Lindvall and Svensson, 2019).

### 3.5 Velocity Variance and TKE Profiles

Turbulence in the ABL is usually generated at the surface and transported upwards. In this scenario, velocity variances and

220 TKE monotonically decrease with height ( Fig. 5 (a)). Turbulence is also generated by shear at the inversion capping convective ABL and supporting entrainment of potentially warmer air from the free troposphere. On occasion, turbulent kinetic energy (TKE) is generated at higher altitudes by meteorological phenomena such as low-level jets, or breaking gravity waves, Kelvin-





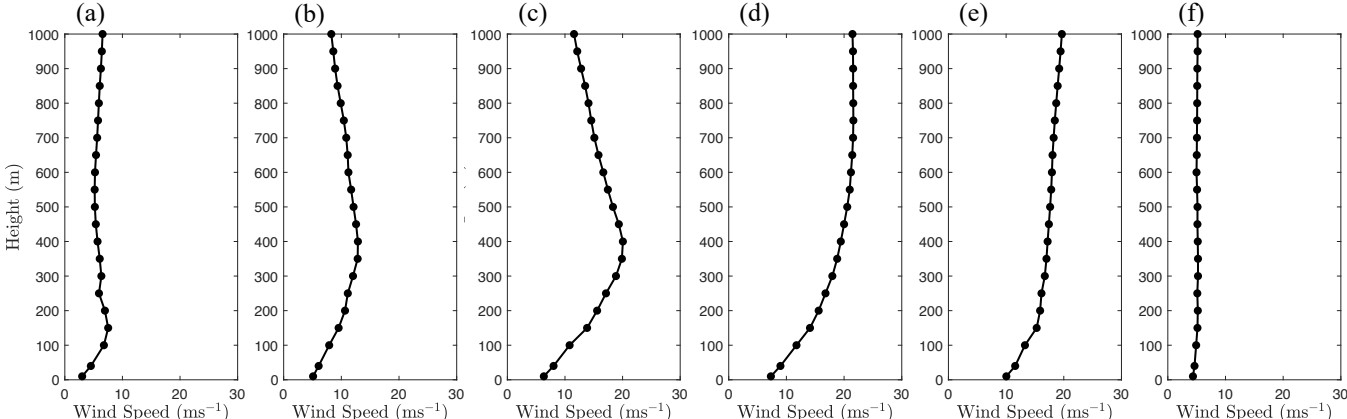

**Figure 4.** (a-f) Diverse wind speed profiles observed at Høvsøre, Denmark during the Tall-Wind Profile experiment (based on the data from Peña et al., 2014).

Helmholtz waves (e.g., Blumen et al., 2001), and then transported toward the ground (as shown in Fig. 5 (a)). Such boundary layers are commonly known as upside-down boundary layers.

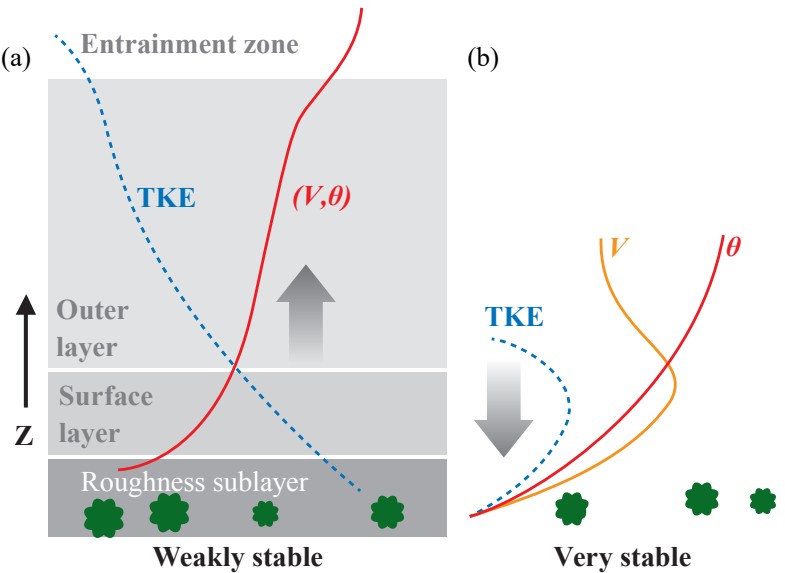

**Figure 5.** Schematic of TKE profiles (and other meteorological variables) in (a) weakly stable and (b) very stable conditions (Used with permission of Annual Reviews, from Annual Review of Fluid Mechanics, "Stably Stratified Atmospheric Boundary Layers", Larry Mahrt, 2014, permission conveyed through Copyright Clearance Center, Inc.)



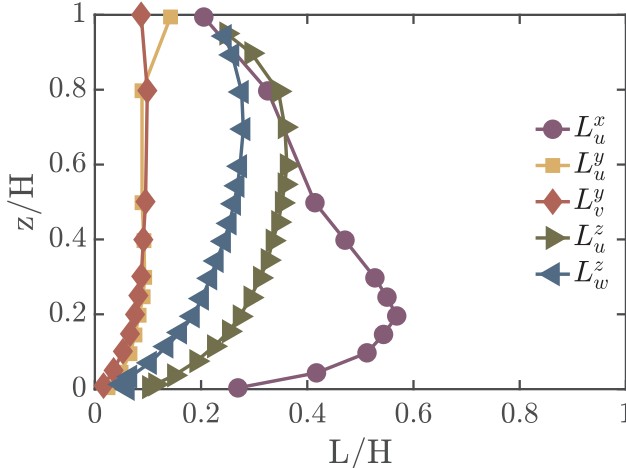

**Figure 6.** Integral length scales in a simulated neutral boundary layer, where $H$ denotes the height of the boundary layer (based on the data used in Fig. 17 from Nandi and Yeo, 2021).

## 3.6 Integral Length Scale

The spatial dimensions of the most energetic eddies are commonly quantified by integral length scales (ILS), $L_k^i$. The ILS, which is usually estimated using correlation functions, also determines the length beyond which the signals become uncorrelated or noise-like, which is a problematic issue in atmospheric turbulence due to the presence of larger structures. The ILS has nine tensorial components corresponding to three velocity components ($k = u$, $v$, or $w$) along the three spatial coordinates ($i = x$, $y$, or $z$). There is a lack of measurement of the different ILS components. In a review paper, Counihan (1975) reported that ILS related to the longitudinal velocity component (i.e., $u$) increases with height in the lower part of the ABL. This trend is expected, as turbulent eddies typically grow larger with increasing distance from the surface. The ILS values are expected to decrease as stability increases, from unstable to neutral and then to stable conditions (for an insightful schematic, see Fig. 2 of van de Wiel et al. (2008)). Based on a thorough analysis of observational data from Kansas, Kaimal (1973) reported that the integral scales (normalized with respect to height) are, in fact, inversely proportional to the gradient Richardson number. Recently, Nandi and Yeo (2021) reported ILS values for neutral boundary layers based on large-eddy simulation (LES) (see Fig. 6). While it is clear that various ILS components vary with height, determining how they vary based on observations is challenging due to the sparsity of required observations. Salesky et al. (2013) have analyzed buoyancy effects on ILS in a surface layer using observations from HATS field study (Horst et al., 2004), while Alcayaga et al. (2022) studied coherent structures, their anisotropy, and related ILS at 50 m and 200 m under a range of atmospheric stability conditions by combining analysis of observations with a dual scanning lidar system and sonic anemometers. The analysis by Alcayaga et al. (2022) demonstrate that ILSs become more isotropic under convective conditions and further from the surface. Syed et al. (2023) conducted analysis of longitudinal and vertical ILSs near a large offshore wind farms based on aircraft observations between



100$m$ and 250$m$ above mean sea level under stably stratified atmospheric conditions. They identified strong correspondence

between the vertical ILS and the length scale of the vertical entrainment over a wind farm.

  In addition to the importance of the ILS, there is also the challenge of determining it precisely from data. In Fuchs et al. (2022) an open source collection of different methods is given.

**Figure 7.** Longitudinal velocity spectra for nine stability regimes. The top, middle, and bottom panels represent unstable, near-neutral, and stable regimes, respectively. Observational data from the FINO1 tower, located in the North Sea, are used for spectral analysis. The variable $\zeta$ represents the so-called stability parameter, a ratio of height and local Obukhov length and $nz/\overline{u}$ denotes the reduced frequency of frequency $n$ (based on Fig. 7 from Cheynet et al., 2018, courtesy of Etienne Cheynet).





## 3.7 Spectra and Coherence

Based on measurements from a flat, uniform field site in Kansas, Kaimal et al. (1972) proposed several generalized spectra and
co-spectra functions for various variables using 10-minute time series. These functions systematically depend on measurement
height, wind speed, and stability (see also Panofsky and Dutton, 1983). The Tchen-Mikkelsen model extends the Kaimal model
for the neutral surface layer by parameterizing the shear production subrange, as discussed by Högström et al. (2002). This
model enhances the Kaimal inertial subrange by incorporating the shear production subrange, which follows a $-1$ power law
scaling to describe the lower part of the surface layer within the eddy surface layer. The Mann model utilizes rapid distortion
theory to model the response of turbulence to shear (Mann, 1994). It provides a spectral tensor based on the principles of flow
incompressibility, the linearized Navier-Stokes equations, small-scale isotropy, and large-scale anisotropy. In addition to the
dissipation, $\epsilon$, the model involves three key parameters: the size of energy containing eddies, $L$, , the spectral Kolmogorov
constant, $\alpha_K$, and the anisotropy parameter, $\Gamma$, to obtain one-point spectra for velocity components, $u$, $v$, $w$, as well as co-
spectra and two-point coherence.

Figure 7 shows observed longitudinal velocity spectra for nine different stability classes for offshore station FINO1 (Cheynet
et al., 2018). The spectra shown can be divided into several regimes. The high-frequency range corresponds to Kolmogorov's
$-5/3$ inertial subrange, followed by the shear production range, characterized by a $-1$ spectral slope. As noted by Cheynet
et al. (2018), this $-1$ slope is particularly evident in the top panels, corresponding to the unstable ABL. The spectral peak
represents the largest eddies in the ABL. Under stably stratified conditions (lower panels), there is a pronounced spectral gap
between the mesoscale range at the lowest frequencies and the spectral peak. Studies have suggested that the gap between the
latter two ranges is not always present. Larsén et al. (2016) and Larsén et al. (2018) demonstrate with measurements from the
surface to about 250 m that the gap exists and can be modeled. In agreement with, e.g., Högström and Smedman-Högström
(1974), Cheynet et al. (2018) showed that the gap can be deep or shallow depending on atmospheric stability. Full-scale spectral
models for the wind components $u$ and $v$ are provided in Larsén et al. (2016) and Larsén et al. (2021). A follow-up study by
(Sim et al., 2023) examined the velocity spectra extending to low frequencies corresponding to a large atmospheric flow scale
of 20-month.

Coherence for both $u$ and $v$ is usually described in terms of frequency $f$ (or wave number), wind speed $U$, distance $\Delta$ and a
decaying coefficient $a$: $Coh(f,\Delta) = -a\frac{f\Delta}{U}$. For typical boundary-layer turbulence, Panofsky and Dutton (1983) found $a \sim 60$
for $u$, and 7 for $v$. For 2D turbulence, Vincent et al. (2013) found $a \sim 7.7$ for $u$, and 5 for $v$, suggesting significantly stronger
coherence in the longitudinal wind component over time and space. For $v$, large-scale flow is also better correlated than the 3D
turbulence, though not as significantly better than $u$.

## 3.8 Global Intermittency and Coherent Structures

One of the intriguing characteristics of boundary layers is the existence of (globally) intermittent turbulent bursts and associated
coherent structures (Shaw and Businger, 1985; Mahrt, 1989). These seemingly random yet highly organized (coherent) features





are generated by various atmospheric processes, such as density current and internal gravity waves (Sun et al., 2004). Illustrative examples can be seen in Fig. 8. The impacts of these features on various turbine loads are discussed in the following sub-section.

In the literature, only a handful of idealized numerical studies (e.g., Zhou and Chow, 2011; Ansorge and Mellado, 2014; He and Basu, 2015) have successfully reproduced some characteristics of observed intermittent turbulence. To the best of our knowledge, realistic simulations, such as LES or gray-zone modeling, of these flow features are still lacking. Since intermittent

turbulence events are often of high amplitudes, but short in duration, traditional stationary or linear signal processing techniques frequently fail to detect or characterize such phenomena. To address this shortcoming, Rinker et al. (2016) developed a technique that introduces temporal coherence which results in a non-stationary signal. Their approach may be utilized to generate turbine inflow generations using stochastic simulations.

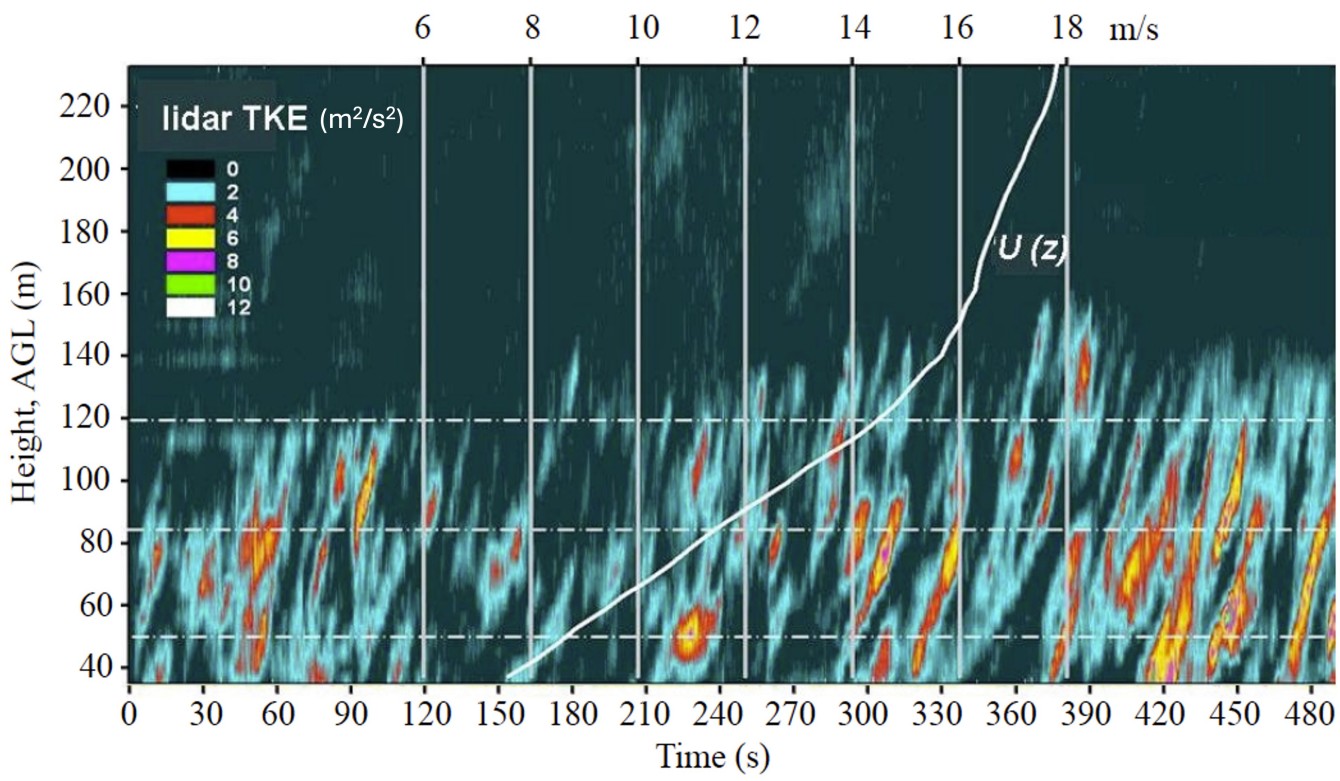

**Figure 8.** Streamwise velocity variance measured by a lidar (adapted from Fig. 7 in: Banta et al., 2008, ©IOP Publishing. Reproduced with permission. All rights reserved). The mean wind speed profile, depicting strong shear, is overlaid. The presence of intermittent turbulence is clearly visible. On several occasions, the turbine design thresholds are exceeded during these bursting events.

### 3.9 Statistical Hierarchy

In addition to the characterization of ABL flows motivated by wind energy, we add here a more mathematical discussion to explain the statistical content of the characterization. We start with one observed scalar quantity $q$. The fluctuations of





$q$ can be captured by the probability density function (pdf) $p(q)$. For the non-fluctuating case, such as a laminar flow, we obtain $p(q) = \delta(q - \overline{q})$. The pdf $p(q)$ determines all nth order moments $\overline{q^n} = \int q^n \, p(q) \, dq$ as well as all central moments $\overline{\mu^n} = \int \mu^n \, p(\mu) \, d\mu$, where $\mu = q - \overline{q}$. In this sense, the mean quantities discussed above and the turbulent kinetic energy as well as the turbulence intensity are only the first two low-order moments of the fluctuation quantities.

In general a pdf has a mean value $\overline{q}$ and a variance $\sigma_\mu^2 = \overline{\mu^2}$ and a form that is best recognized for $p(\tilde{q})$ by transforming $q$ to $\tilde{q} = \frac{q - \overline{q}}{\sigma_\mu}$. In Fig. 9 (a) an example of $p(\tilde{q})$ is shown in which the pdf has pronounced heavy tails. The corresponding Gaussian pdf is completely defined by $\overline{u} = 0$ and $\sigma_\mu$ and displayed as a solid curve. Note the large difference of the probability of large events, a $\mu = 5\sigma$ event is more than 100 times more frequent in the empirical pdf than in a Gaussian distribution. Similarly, the pdf of time increments of velocity has heavy tails (9 (b)). Such a heavy-tailed pdfs are also called intermittent, but this statistical intermittency is not the same as other intermittent phenomena. A central point is that this statistical intermittency can only be captured by higher order statistical moments, i.e. for $n > 2$.

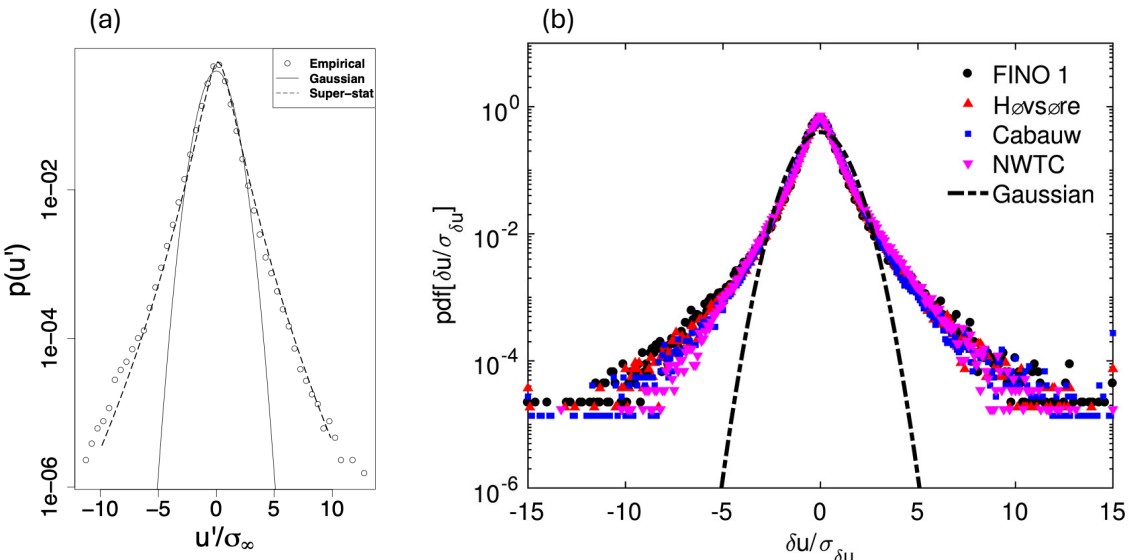

**Figure 9.** (a) A pdf of fluctuating velocities form FINO1 data of January 2006. The fluctuating quantity $q$ is given as $u'$ in units of the standard deviation, Fig. 2 (b) in Morales et al. (2012) reprinted by permission from John Wiley and Sons. Copyright ©2011 John Wiley & Sons, Ltd. (b) A pdf of wind ramps, i.e., time increments, $\delta u$, from four tall-tower sites (FINO1, Høvsøre, Cabauw, and National Wind Technology Center (NWTC)). Multiyear, 10-min averaged wind data measured by the topmost sensors on these towers are used here (source: DeMarco and Basu, 2018, licensed under CC BY 4.0 license).

A pdf $p(q)$ of a data set $\{q_i; i = 1, ..., N\}$ is insensitive to any order in the sequence i. ($i$ can denote a time or space index here.) Only if two values $q_i$ and $q_j$, with $j = i + \Delta$, are statistically independent the characterization by the pdf $p(q)$ is complete. Since structures in turbulence lead to dependencies of $q_i$ and $q_j$, such dependencies or correlations are of interest. Statistically, this is captured by the joint-pdf $p(q_i, q_j) = p(q_i, q_{i+\Delta})$, which for homogeneous data only depend on $\Delta$. The lowest order





moment of this joint-pdf is $\overline{q_i\, q_j}$, which is the autocorrelation up to normalization. Higher order correlations are $\overline{q_i^n\, q_j^m}$. The Wiener–Khintchine theorem states that the power spectrum $S_u(n)$ is proportional to the Fourier transformation of autocorrelation. For the frequency, $n \propto 1/\Delta$ applies. Thus, the power spectrum is a second-order statistical quantity that does not

characterize the well known small-scale intermittency of turbulence. The small-scale intermittency is typically characterized by higher order moments of increments $\overline{(q_{i+\Delta} - q_i)^n}$ or by the pdfs of increments, for more details see Morales et al. (2012).

This discussion can easily be extended to several variables, the so-called multivariate statistics. Here too, joint-pdf or moments of different order (called covariance) can be analyzed. In addition to discussing the order of the moments, it is also important to think about how many independent variables are considered. The initial pdf $p(q)$ can be considered a one-quantity

(or one-point) statistic. If two quantities, such as velocity components in two directions, $u$ and $w$, or at two spatial or temporal locations ($u(x)$ and $u(x+r)$ or $u(t)$ and $u(t+\tau)$ ), are considered, it is a two-quantity or two-point (two-time) statistic. In this sense, a power spectrum is a two-time statistic. From this discussion it becomes clear that for the statistical characterization of turbulent structures, such as gusts, jets, etc., joint multi-point (or multi-time) statistics become necessary. First steps in this direction can be found, for example, in Morales et al. (2012) and Behnken et al. (2020).

## 4 Some Relevant Atmospheric Phenomena

Low-level jets (LLJs), mesoscale convective rolls and cells, gravity waves, terrain-induced circulations, and downslope windstorms are atmospheric phenomena with distinct wind and turbulence structures. The mechanisms and characteristics of these phenomena are briefly introduced here, as their impacts on power production and loads are discussed in sections 7 and 8, respectively. These phenomena occur both on land and offshore. For more detailed information about offshore environments

relevant to wind energy applications, see Shaw et al. (2022), which covers aspects such as the effects of ocean surface waves. As a result, specific details of offshore conditions are not extensively addressed in this paper.

### 4.1 Low-level Jets

LLJs are fast-moving air currents within several hundred meters of the surface. They typically occur at night and are common in many areas. The North American Great Plains LLJ is a well-known example that occurs almost every night over central

North America (e.g., Bonner, 1968; Zhong et al., 1996; Whiteman et al., 1997; Berg et al., 2015). LLJs are also common over central South America (e.g., Marengo et al., 2002, 2004; Gimeno et al., 2016), and many offshore and coastal areas worldwide (e.g., Winant et al., 1988; Burk and Thompson, 1996; Parish, 2000; Kalverla et al., 2019; Lima et al., 2019; Debnath et al., 2021; Aird et al., 2022). LLJs can be caused by various physical mechanisms, such as inertial oscillation, sloping terrain, cold fronts, baroclinicity due to land-sea temperature contrasts in coastal areas, barrier winds, and other topographically

induced adjustments. Figre 10 provides examples of vertical wind profiles in onshore and coastal LLJs. The figure demonstrates that LLJs generate high amplitude wind speeds, creating strong shear. In addition to strong shear significant wind veer is a characteristic of LLJs (e.g., Vanderwende et al., 2015). Above the LLJ nose, the shear is negative, affecting wind turbine loads (see section 8.2.1). Due to their relatively low altitude above ground level, typically 100– 500 m (Fig. 10), LLJs have important





implications for wind resource assessments, wind power forecasting, and turbine loading, particularly as wind turbines become

taller and rotors bigger.

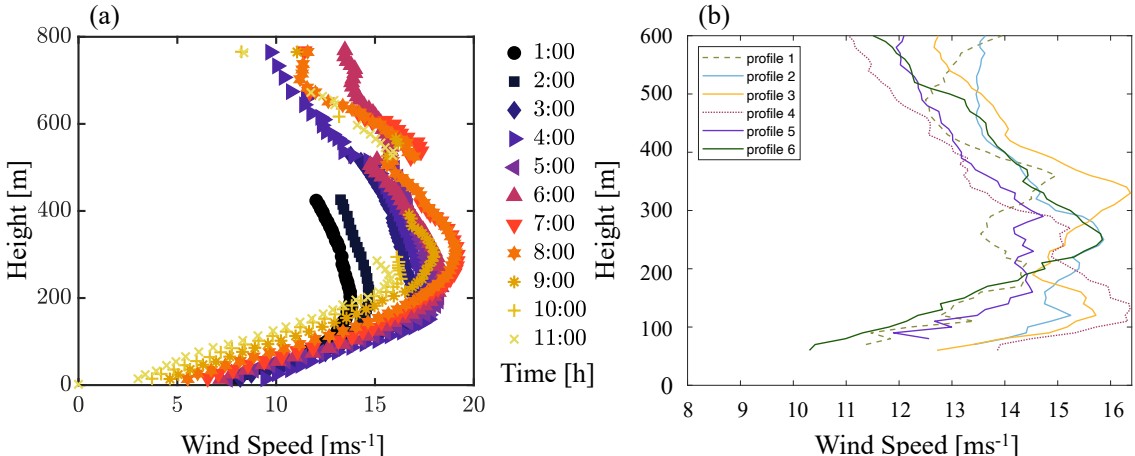

**Figure 10.** (a) A developing LLJ was observed in Colorado during the Lamar Low-Level Jet Project in 2003. Wind speed profiles were estimated by averaging lidar observations over an hour (based on data used in Fig. 7 from Banta, 2008). (b) Wind speed profiles showing LLJ in the German Bight in the North Sea, from the airborne measurements data published in Bärfuss et al. (2019), on 14 October 2017, from about 13:23 to 16:23 o'clock. The corresponding flight tracks to the profiles can be found in Fig. 2 in Larsén et al. (2021), covering the area from 53.85N to 54.20N and 6.95E to 7.65E.

The global distribution and characteristics of onshore and coastal LLJs have been studied using global simulations and reanalysis datasets (e.g., Rife et al., 2010; Ranjha et al., 2013; Lima et al., 2018). In addition, several studies utilized high-resolution mesoscale models to downscale global data (e.g., Soares et al., 2014; Rijo et al., 2018; Soares et al., 2019). However, mesoscale simulations of LLJs are challenging. Storm et al. (2009) investigated the performance of the Weather Research and

Forecasting (WRF) model in simulating the North American Great Plains LLJ. They found that the WRF model, with various physical configurations, can capture some of the observed LLJs' characteristics, such as their location and timing. However, the model overestimated the LLJ height, while underestimating its strength compared to observations. Nunalee and Basu (2014) found that WRF generally underestimated the intensity of offshore LLJs. Using data collected at an Iowa wind farm during the Crop Wind Energy Experiment (CWEX) field study, Vanderwende et al. (2015) showed that the WRF simulated LLJ properties

depended on the initial and boundary conditions as well as the employed boundary layer parameterizations. Aird et al. (2021) used WRF to study seasonal variation of LLJs over Iowa and investigated the dependence of LLJ characteristics on vertical resolution and criteria for identifying LLJs. The shortcomings of WRF in capturing LLJs highlight the need for further research to improve the representation of critical atmospheric boundary layer processes. Muñoz-Esparza et al. (2017) performed high-resolution WRF simulations using nested domains ranging from mesoscale up to LES domains to simulate the diurnal cycle

dynamics of the CWEX field study. They demonstrate that LES nested in a mesoscale domain can accurately capture the magnitude of the LLJ and associated turbulence properties. This seems to suggest that the standard setup of WRF may be





too coarse of a resolution for resolving atmospheric dynamics of LLJ. Using the US Navy's COAMPS model, Ranjha et al. (2016) investigated the resolution dependence of coastal LLJs. As expected, the model produced more realistic flow features with increasing horizontal resolutions from 54 km to 2 km. Interestingly, the model's performance did not show a monotonic behavior concerning spatial resolution in terms of traditional skill scores.

To accurately model LLJs, we need to improve our understanding of the phenomenon and, simultaneously, our understanding of different models' capabilities. The nature of LLJs requires a mesoscale model to resolve the large-scale atmospheric flow characteristics and a high-resolution model to resolve the small-scale turbulence. We need more coordinated measurement campaigns with instruments for both mesoscale and microscale flows. Currently, there is a lack of observed wind speed and direction profiles associated with LLJs, particularly for offshore and coastal conditions Shaw et al. (2022). Towers typically only reach 100-200 m, and sodars cannot penetrate the highly stratified layer near the LLJ nose. While profiling (floating) lidars can provide more information, they are expensive and not routinely used and typically have a vertical range of approximately 200 m. These measurements should also provide more information about the turbulence structures near and above the LLJ nose. The knowledge gained from the analysis of measurements is expected to shed light on how to couple the mesoscale and microscale modeling to fit LLJ's nature. It should also be noted that the LLJ structure has not yet been consistently related to the spectral features described above.

### 4.2 Mesoscale Convective Circulations - Rolls and Cells

Mesoscale convective circulations accompany synoptic weather systems such as fronts or cold and dry air outbreaks from the continent or ice shelves over a neighboring warm surface. Significant air-surface differences in temperature and humidity, combined with large wind shear, can result in helical roll vortices (LeMone, 1973). Rolls with single or multiple thermals within the roll updraft regions can lead to cloud streets (Kuettner, 1971). Observations suggest that, as the distance from the coastline increases, rolls change into three-dimensional, circular open cells with upward motion on the edge and downwards motion in the center (Atkinson and Zhang, 1996; Young et al., 2002; Salesky et al., 2017). These organized atmospheric structures exhibit coherent updrafts and downdrafts, contributing significantly to the vertical turbulent fluxes of momentum, heat, and humidity. Consequently, they influence surface fluxes, the height of the boundary layer, and the spatial correlation of meteorological parameters such as wind, temperature, humidity, and particle concentrations. Typical characteristics of rolls and cells are summarized in e.g. Atkinson and Zhang (1996); Etling and Brown (1993); Young et al. (2002); Banghoff et al. (2020). Their summary suggests that rolls align along or at angles up to 10 degrees from the mean horizontal wind, with typical lengths of 20 to 200 km, widths of 2 to 10 km, and convective depths of up to 3 km. They have been observed both over water and land surfaces. Convective cells, on the other hand, typically have diameters ranging from a few to 40 km and convective layer depth of 1 to 5 km.

The need to study rolls and cells lies in the fact that they are common phenomena frequently observed over the globe, both over water and land, resulting in significant fluctuations in wind speed that impact wind power production and significant levels of turbulence that impact turbine loads. Agee (1987) showed a global distribution of the presence of convective cells. The favorable thermal and dynamical conditions that produce them can occur in connection with cold air outbreaks and fronts





(e.g., Skyllingstad and Edson, 2009; Vincent, 2010), regional flow such as the Mistrals and Tramontane (e.g., Brilouet et al., 2017), and hurricanes (e.g., Zhu, 2008; Worsnop et al., 2017b). They have been observed in the North Sea (e.g., Brümmer et al., 1985; Vincent, 2010), the Mediterranean Sea (Brilouet et al., 2017), the Yellow Sea (Chen et al., 2019), Gulf of Mexico (Young et al., 2002), along the East Coast of US (e.g., Fig. 11 (a)), Siberia and Japan, South of Bering strait (Agee, 1987;

Atkinson and Zhang, 1996), East China Sea (Hsu and yih Sun, 1991). Most often, they were observed under near neutral and convective conditions, though sometimes, rolls have also been found under stable conditions. In Svensson et al. (2017), rolls were observed over the Baltic Sea under stable conditions. These rolls were originally generated under convective conditions over land and then advected and maintained at least 30 – 80 km off the coast.

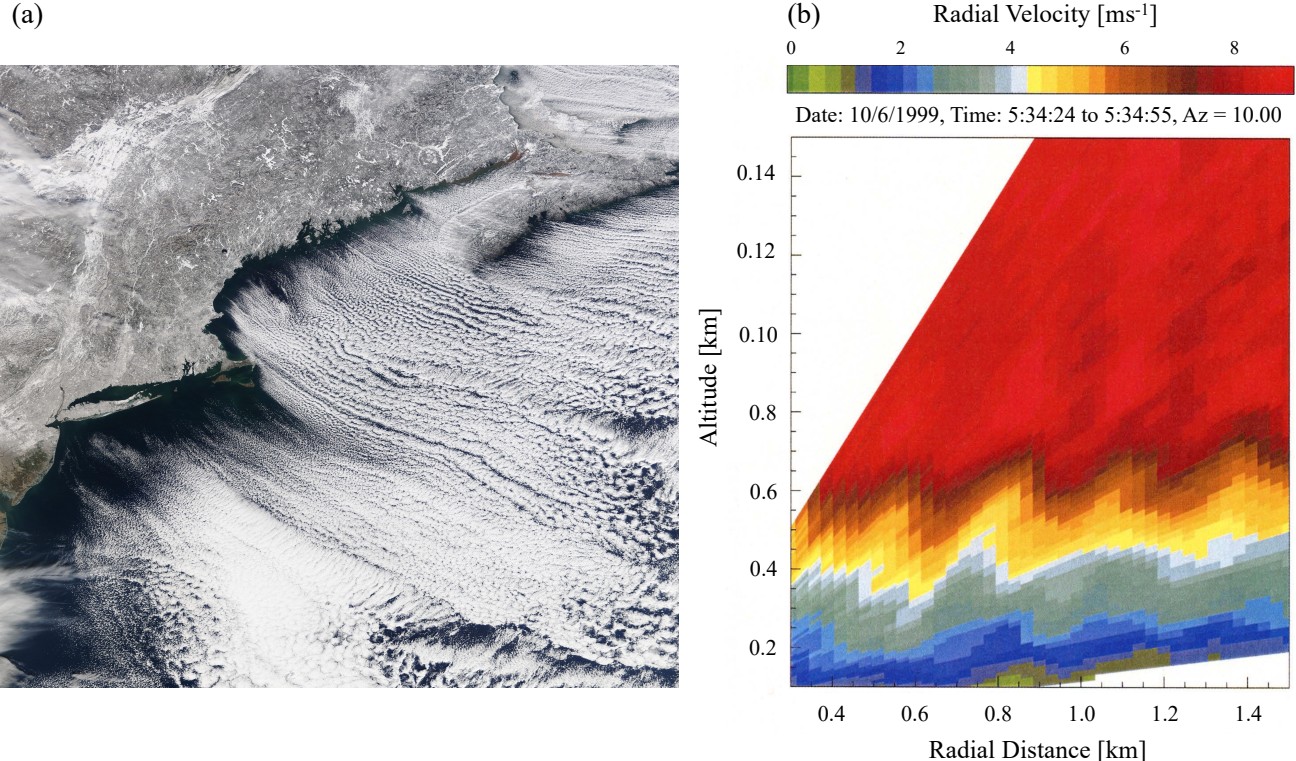

**Figure 11.** (a) Satellite image of clouds over the East Coast of the United States associated with a cold air outbreak on January 24, 2011 (NASA Worldview Snapshots), indicating the presence of mesoscale convective circulations, convective rolls closer to the coast, and convective cells further downwind from the coast. (b) NOAA's wind lidar (HRDL) captured gravity waves on October 6, 1999, during the CASES-99 field campaign (source: Poulos et al., 2002, ©American Meteorological Society. Used with permission.).

## 4.3 Gravity Waves

In a stable atmosphere, wind flow perturbed upwards over a hill or ridge or by a strong convective updraft (e.g., thunderstorm) can experience a restoring force that can create oscillations and trigger waves to be transmitted perpendicular to the streamlines





over the orographic feature. While gravity waves can occur throughout the troposphere, of relevance for wind energy are gravity waves that impact boundary layer flows. In a stably stratified ABL gravity waves can be trapped by a strong potential temperature inversion (e.g., Fig. 11 (b)). Suppose the temperature stratification is sufficiently strong and the winds are moderate, the waves can be reflected back down to the ground and trapped as an atmospheric gravity wave (AGW) known as 'lee waves' (Nappo, 2013). Lee waves can propagate long distances downstream from the orographic feature that generated them, as demonstrated by, for example, Larsén et al. (2011), and impact flow in the boundary layer. AGWs occur widely in the atmosphere (Mahrt, 2014; Urbancic et al., 2021) leading to wind speed variations in the order of minutes to hours, which can cause wind farm power output fluctuations. Although this scale of fluctuation does not impact turbulent loading, wind shear and wave breaking can generate turbulence, sometimes leading to what is sometimes termed an 'upside-down' atmosphere with higher levels of turbulence aloft (Mahrt, 1999); see Fig. 5 (b). Under synoptic conditions, with a strong pressure gradient perpendicular to a mountain range, lee waves can cause downslope wind storms and result in a hydraulic jump.

AGWs enhance wind speed at the wave crests, which increases bulk shear instability, generating eddies that promote strong turbulent mixing from above and potentially create small LLJs; near the surface, wave troughs promote local shear, contributing to the generation of weak turbulence (Sun et al., 2015). However, the impact of AGWs on small-scale turbulence near the surface has not been studied yet.

## 4.4 Terrain induced circulations

Land/sea breezes resulting from differences in surface heating rates between land and water and topographically induced circulations such as speed-ups on ridges or escarpments, gap flows, and drainage flows can represent a significant wind resource. Wind farms frequently deployed in locations with terrain effects result in favorable wind conditions. However, terrain-induced circulations can also be characterized by significant wind variability and non-equilibrium turbulence that impact wind farm performance. Historically, characterization of turbulence in an ABL focused on flows over flat and primarily horizontally homogeneous terrain, while boundary layer studies in mountainous terrain concentrated on characterizing the mean structure of the mountain ABL (e.g., Whiteman, 2000). During the last decades of the 20th century, several field studies focused on flows over isolated hills, e.g., Blashaval Hill (Mason and King, 1985) and Askervein Hill (Taylor and Teunisse, 1987) as well as transport and dispersion (Lavery et al., 1982). The Atmospheric Studies in Complex Terrain (ASCOT) was a major program focused on characterizing drainage flows and turbulent mixing (Orgill and Schreck, 1985) followed by the VTMX 2000 field study (Doran et al., 2002). Over the last few decades, several field studies have been designed to address turbulent ABLs in complex terrain and the range of processes affecting their structure. Some of the examples are: the Mesoscale Alpine Program (MAP) Riviera project (Rotach and Zardi, 2007) focused on a diurnal cycle of valley flows, the T-Rex project, a study of mountain-induced rotors (Grubišić et al., 2008), COLPEX (Price et al., 2011) analyzed cold pool processes, MATERHORN (Fernando et al., 2015) addressed a combination of topography and thermally induced circulations, while stably stratified boundary layers were the focus of METCRAX and METCRAX II field studies (Whiteman et al., 2008; Lehner et al., 2016). The early field studies were also used to validate numerical models. However, only recently were turbulence resolving, large-eddy simulations applied to flows over complex terrain (e.g., Chow et al., 2006; Chow and Street, 2009; Babic and De Wekker,





2019; Arthur et al., 2022). Chow et al. (2019) summarized the advances and challenges in simulating and forecasting flows in complex terrain. While these field and numerical studies were not focused on wind energy applications, they represent an important resource for understanding and characterizing the conditions under which wind farms in complex topography operate. Turbulence characteristics in complex terrain are impacted by the wide range of flow phenomena in addition to spatial
inhomogeneity frequently resulting in non-equilibrium conditions. As indicated above, most of the field studies conducted to date focused on specific flow phenomena, however, the diversity of complex terrains resulting in variety of flow phenomena and frequently interactions between them represent a challenge to developing systematic understanding of turbulence and its evolution in complex terrain.

### 4.5 Downslope Windstorms

Downslope windstorms develop under favorable synoptic conditions and stable stratification when a cross-mountain range flow results in gravity waves and flow acceleration near the surface on the lee side of the mountain. In downslope windstorms wind gusts can exceed 60 m $s^{-1}$ (Čavlina Tomašević et al. 2022). These storms can form when the mountain top is at least 1 km above the lee-side terrain with a steep lee-side slope. A detailed review of mountain wind storms can be found in Durran (1990). Winds associated with mountain ranges that frequently result in wind storms across the world include Chinook in the
Rocky Mountains, Foehn in the Alps (Haid et al. 2020), Bora along Eastern Adriatic (Lepri et al., 2017), Zonda in South Africa (Loredo-Souza et al., 2017), and Santa Ana (Fovell and Cao, 2017), and Diablo winds in California, and Yamajikaze in Japan (Kusaka and Fudeyasu, 2017), a to name a few, are downslope windstorms.

There has been relatively little research into the interactions between downslope wind storms, power production, and turbine loads compared to studies of other extreme wind events. Observational studies, such as those by Sherry and Rival (2015),
studied wind ramps associated with Chinook winds downwind in the Canadian Rockies. One of their findings was that turbulence intensity was generally large during ramp events. Kozmar and Grisogono (2020) provided a review of characteristics of downslope wind storms including mountain wave overturning and quasi periodic oscillations in wind speed and elevate turbulence levels result of Kelvin-Helmholtz instabilities that are particularly relevant for wind energy applications. They point out the need for updating engineering standards to account for large wind velocity fluctuations associated with downslope wind
storms.

### 5  Observing ABL Flows

A wide range of instrumentation, including in-situ and remote sensing, can be used to quantify the characteristics of mean flow and atmospheric turbulence. Over the last 50 years, sonic anemometers have been the workhorse in-situ instruments in boundary-layer studies (Kaimal, 1986). Sonic anemometers can measure all three components of the velocity vector and
sample at high frequency so that the energy and momentum fluxes can be computed using the eddy-covariance method. This approach requires the additional assumption that the turbulence is frozen in time as it advects past the sensor, referred to as the so-called Taylor Frozen Turbulence hypothesis (c.f., Wyngaard, 2010). In other applications, such as resource assessment, cup





or propeller anemometers are commonly used, though their frequency response is inferior to that of sonics. With these sensors, one can compute estimates of the turbulence intensity based on the standard deviation of the wind speed over a time interval of interest. Sonic, cup, and propeller anemometers must be mounted on towers or some other support structure, meaning that they represent only a very small area and are generally located relatively close to the surface.

In-situ measurements can also be made using dedicated aircraft with appropriate instruments (e.g., Schmid et al., 2014; Laursen et al., 2006). These can include crewed aircraft of various sizes, and wind is measured using a pitot tube and the craft's navigation system. Deployment of these aircraft for ABL studies is generally expensive, and past research has focused on intensive measurement campaigns and specific case studies. In addition, it isn't easy to measure the time evolution of the wind using an airborne platform. Recently, there has been interest in the deployment of smaller, uncrewed aircraft to measure boundary-layer winds, and these systems will likely become more common in the future (Pinto et al., 2021).

Remote sensing systems have increasingly been used to address shortcomings associated with in-situ measurements. Radar wind profilers are commonly used to measure profiles of wind speed and wind direction from approximately 70 meters above the surface to many kilometers aloft, depending on the frequency of the radar used in the system. Scanning radar systems, such as Ka-band radar, have been used to study flow in and around wind farms (e.g. Hirth et al., 2012). Doppler sodars, which can measure profiles of wind speed and direction in the lowest several hundred meters of the atmosphere, have been used in a wide range of field studies (e.g., Wilczak et al., 2019). Recently, however, they have largely been replaced by Doppler lidar systems in many applications. Some lidar systems are configured to measure only the profile of wind speed and direction using fixed stare directions. These lidars have also been installed on buoys and other floating platforms to provide profiles of wind speed and direction in maritime environments (e.g., Gorton and Shaw, 2020). In addition to the mean profiles, algorithms have been developed to derive TKE and TI using these systems. The field study 3D Wind conducted over a large wind farm using in situ and remote sensing instruments focused on the structure wind and turbulence in a lower region of an ABL relevant for wind energy and found good agreement between lidar and cup anemometer observations (Barthelmie et al., 2014). However, differences have been observed compared with measurements from sonic anemometers (Sathe et al., 2015). Other systems are configured to scan in arbitrary patterns. They can be used to probe the nature of the flow over a larger area or can be combined to scan in a coordinated pattern to provide measurements akin to virtual towers (e.g., Newsom et al., 2013) or measurements over a relatively large area (e.g., Berg et al., 2017). Scanning lidars can also be mounted on wind turbine nacelles to investigate details of the inflow to the turbine or wakes (e.g., Borraccino et al., 2017; Trujillo et al., 2016) or used as input to turbine controls to reduce loads (e.g., Held and Mann, 2019). Field studies, such as the Planetary boundary-layer Assessment eXperiment (XPIA), provided the opportunity to compare measurements from Doppler lidar and sonic anemometers and showed good agreement between measurements made in-situ or using lidars and radars (Lundquist et al., 2017).

There have been several recent field studies to characterize mean and turbulent flow in complex terrain with an emphasis on resource assessment and forecasting, including the Wind Forecast Improvement Project 2 (WFIP2 Shaw et al., 2019; Wilczak et al., 2019) and a series of New European Wind Atlas field studies: Perdigao experiment (Fernando et al., 2019), Alaiz experiment (Santos et al., 2020), and Kassel Forested Hill experiment (Pauscher et al., 2017). These studies have produced a wealth of unique observations that will continue to provide opportunities for insightful analysis of turbulence characteristics



as they may be modulated by complex terrain and thus deviate from observations in flat terrain. The challenge is to develop and demonstrate effective methodologies to estimate turbulence characteristics over heterogeneous terrain that could be used

to optimize wind farm layout and turbine design. For example, Wildmann et al. (2019) used lidar range-height indicator (RHI) scans during the Perdigao experiment to retrieve turbulence dissipation rate using the modified Doppler spectral width method validated against tethered lifting system based hot-wire anemometer measurements and demonstrated a good agreement between remote sensing and in situ observations of dissipation. Wildmann et al. (2020) used dual Doppler lidar observations to estimate turbulence intensity, including in the wake of a wind turbine located at the western ridge of the Perdigao field study.

Peña and Santos (2021) combined scanning lidar observations and numerical simulations over a selected period during the Alaiz field study and demonstrated that it is possible to simulate an observed hydraulic jump in the Lee of Alaiz mountain. The hydraulic jump develops under strong downslope wind conditions resulting from cross-mountain flow and associated mountain waves. The effects of complex, hilly terrain on turbulence characteristics in wind farms are also studied using wind tunnel experiments (e.g., Kozmar et al., 2018). There are ongoing field studies focused on complex terrain, including at the

WINSENT research facility (WindForS, 2024) and the coastal research wind farm WiValdi (Research Alliance Wind Energy, 2024).

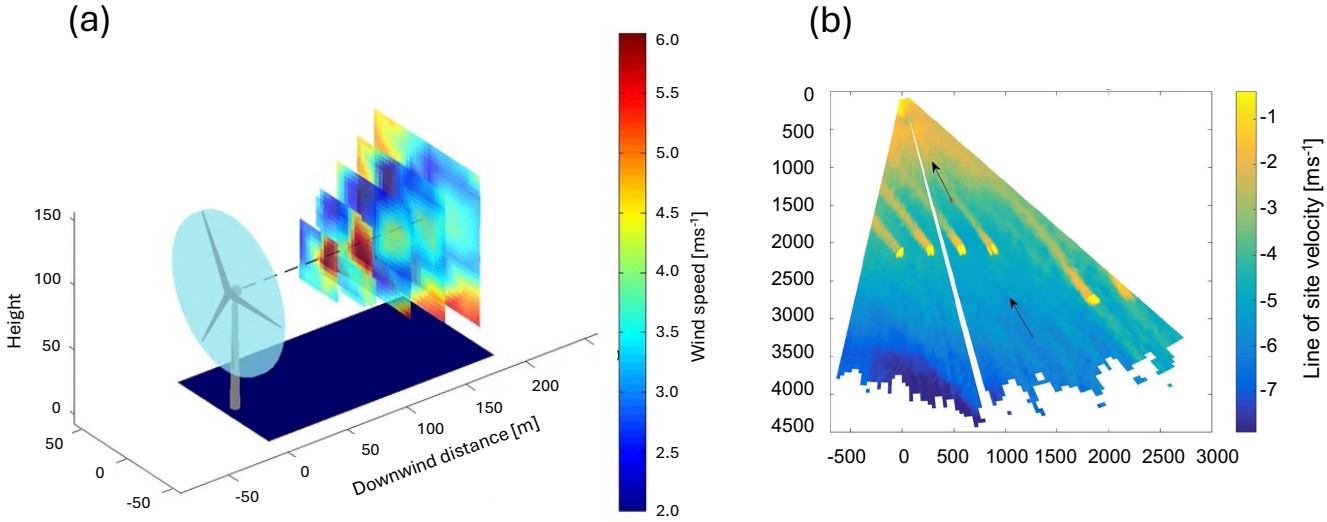

**Figure 12.** (a) Samples of wind speeds measured using nacelle mounted doppler lidar (source: Trujillo et al., 2016, published under CC BY 3.0 license). (b) A ground based scanning lidar horizontal scan (source: Bodini et al., 2017, published under CC BY 3.0 license)).

## 6   Modeling of ABL Flows

High-rate, in-situ atmospheric measurements of turbulence using, for example, sonic anemometers, provide the most reliable characterization of ABL turbulence (e.g., Mann et al., 2009). However, such measurements are either sparse or not always





available when considering wind farm development and wind turbine siting. More often available are only estimates of turbulence intensity based on cup anemometer measurements. Numerical simulations can complement observations or be used as an alternative. However, numerical simulations of atmospheric flows are limited by computational requirements and available computational resources. Currently, simulations that resolve all the scales of atmospheric motions are impossible. Instead, three types of atmospheric simulations are employed, each resolving a different range of scales. Global circulation simulations

resolve the largest scales, planetary waves, and synoptic systems with grid cell sizes down to the order of ten kilometers. Global weather simulations are downscaled using limited area models to resolve regional weather: storms, mesoscale convective systems, and effects of topography and other surface heterogeneities with grid cell sizes on the order of a couple of kilometers. In global and mesoscale simulations, three-dimensional boundary layer turbulence is still fully parameterized and is not a reliable alternative to observations. To resolve ABL turbulence, from the largest boundary layer eddies into the inertial

range of turbulence characterized by Kolmogorov -5/3 spectrum (Kolmogorov, 1941), we can employ large-eddy simulations (LESs). High-resolution LES with properly validated numerical models can complement measurements. LES resolves large turbulent eddies extending into the inertial range. The inertial range is characterized by a forward cascade transferring kinetic turbulent energy down to small scales and is marked by the so-called -5/3 scaling. The universal scaling enables effective parameterization of the effect of small, unresolved eddies on resolved ones. Estimating the direct impact of the ABL turbulence

on wind turbine power production requires resolving eddies spanning two to three orders of magnitude in size. The largest ABL eddies, in general, scale on the ABL height and are in an order of thousand meters, while the smallest eddies impacting turbine performance are in the order of meters or frequency of a few Hertz.

Since its inception, more than fifty years ago Lilly (1966, 1967), LES has been effectively used to study idealized, canonical ABLs (e.g., Deardorff, 1972; Moeng, 1984; Andren, 1995; Kosović and Curry, 2000). LES has been used as a research tool

to study the interaction between ABL flows and operating turbines represented using either actuator disk models or actuator line models (Sørensen and Myken, 1992; Sørensen and Shen, 1996; Troldborg et al., 2007). Actuator disk and line approaches were initially used in idealized Reynolds Averaged Navier Stokes (RANS) simulations that did not account for some of the complexities of canonical ABL flows, such as atmospheric stability. More recently, the representation of operating turbines was implemented in atmospheric LES models, allowing for the analysis of atmospheric stability effect on wind turbine wakes

(Mirocha et al., 2014; Aitken et al., 2014). However, turbulence in an ABL is frequently affected and modulated by surface heterogeneities and larger mesoscale and synoptic scale flows. Accounting for the large-scale effects of ABL turbulence is essential for a range of applications, including wind energy. This requires coupling mesoscale simulations and LES (Haupt et al., 2019). A multiscale simulation approach was recently used to study the effect of frontal passage on wind farm performance (Arthur et al., 2020).

High-performance computing capabilities, including new accelerator technologies (e.g., General Purpose Graphical Processing Units or GPGPUs), enable blade-resolving LES simulations (Sprague et al., 2020). Due to relatively modest computational requirements, RANS simulations are still the tool of choice for design purposes, even though in these simulations, only the mean flow properties are represented, and turbulence is fully parameterized. However, as the HPC approaches exascale computing, LES will become accessible for a range of applications and will not only be used as a research tool (Sanchez-Gomez




et al., 2024). At present, accurately characterizing turbulence affecting wind turbine performance in numerical simulations under variable working conditions represents a challenge.

## 7   Impact of Turbulence on Power Production

The measurement methodology for turbine power performance is specified by the more recent update of the standard EC 61400-12-1:2022. This standard defines the measurement procedure for a single turbine of any type and size. Furthermore,
the standard defines a procedure requiring assessing sources of uncertainty accompanied by measurements of derived energy production and the power curve.

### 7.1   Power Curves

When extracting power from the wind, a modern utility-scale variable-speed wind turbine typically follows a prescribed power curve across its operational range. This is shown schematically in Fig. 13(a). Below cut-in (Region 1), no electrical power is
produced due to insufficient power in the wind. Between cut-in and rated wind speeds (Region 2), the rotational speed of the turbine is controlled as far as possible to maintain maximum aerodynamic efficiency across the range of wind speeds within this operational region. Ideally, assuming a steady laminar flow, the power produced in this region is given theoretically by:

$$P = \frac{1}{2}C_P A_T \rho U^3,\tag{8}$$

where $\rho$ is the density of air and $A_T$ represents the area of the wind turbine rotor disk. The parameter $C_P$ is the so-called
power coefficient. It has a theoretical maximum value of 0.593 and is known as the Betz limit (Betz, 1920). This limit was also derived around the same time by Lanchester and Joukowsky (Lanchester, 1915; Joukowsky, 1920) and is sometimes referred to as the Lanchester-Betz-Joukowsky limit. In practice, a wind turbine rotor does not obtain this value, and $C_P$ depends on the aerodynamic design of the rotor, the tip speed ratio $\lambda = \frac{\omega R}{U}$ (where $\omega$ is the angular velocity, and $R$ is the rotor radius) and the blade pitch angle. Just below the rated wind speed for a turbine (sometimes known as Region 2.5), the wind turbine controller
will deviate from optimum aerodynamic efficiency for practical reasons (e.g., load and noise mitigation). Between rated and cut-out wind speed (Region 3), blade pitch is adjusted to limit power extracted to the rated power of the generator. During periods of very high wind speeds (typically 25-30 m s$^{-1}$), the turbine shuts down for safety reasons by feathering the blades (Region 4).

Figure 13(b) shows a scatter plot of the power output of a wind turbine as a function of wind speed measured upstream of
the turbine rotor sampled at 1 Hz and then statistics calculated within a 10-minute period. It is clear that there is a significant degree of scatter in the data. Many factors contribute to this scatter, which relates both to the unsteady, turbulent nature of the inflow wind profile and the ability of the turbine controller to respond to this rapidly changing wind profile. With stochastic methods, it is possible to partially disentangle these contributions (Lin et al., 2023).

To provide a meaningful estimate of the expected power output of a wind turbine as a function of wind speed, the In-
ternational Electrotechnical Commission (IEC) 61400-12-1:2022 standard (International Electrotechnical Commission, 2022)





provides guidance on the measurement of a power curve. This is based on 10-minute averages of concurrent wind speed and power measurements sampled typically at 1 Hz, which are then averaged in 0.5 m s$^{-1}$ bins. The standard also provides for corrections to be made for air density. A typical bin-averaged power curve is shown in Fig. 13(c).

In its simplest form, the power curve is based on point measurements of 10-minute averaged wind speed at hub height
upstream of the rotor, $\overline{U}_H$. However, the shape of the wind profile will influence the power curve, and the use of a rotor-equivalent wind speed, $\overline{U}_{REWS}$ (Sumner and Masson, 2006; Wagner et al., 2009) is recommended. The level of turbulence intensity may affect the power curve. However, its effect on a power curve is not straightforward to determine as turbulence intensity is closely linked to other parameters that affect the wind profile, such as atmospheric stability, wind shear, and veer, which are interrelated. For example, stable atmospheric conditions will increase wind shear but will also increase veer and
reduce turbulence intensity (and vice versa under convective conditions). Stability has been shown to affect the power curve (Antoniou et al., 2009; Wharton and Lundquist, 2012), but the direct impact of turbulent intensity cannot be easily quantified.

As a first consideration, 10-minute averages do not truly reflect the total power in the wind (de Vries, 1978; Burton et al., 2001). Consider the instantaneous wind speed $U$:

$$U = \overline{U} + U' \tag{9}$$

where $\overline{U}$ is the 10-minute average wind speed and $U'$ is an instantaneous fluctuation from the mean, assumed to be sampled from a normal distribution. Eq. 8 tells us that the power in the wind is proportional to the cube of the wind speed (at least in Region 2) so that the mean power $\overline{P}$:

$$\overline{P} \propto \overline{(U^3)} = \overline{(\overline{U} + U')^3} = \overline{(\overline{U})^3} + 3\overline{\overline{U}^2 U'} + 3\overline{\overline{U}(U')^2} + \overline{(U')^3} = \overline{U}^3 + 3\overline{U}\,\overline{(U')^2} = \overline{U}^3 + 3\overline{U}\sigma_u^2 = \overline{U}^3(1 + 3I_u^2) \tag{10}$$

where $I_u$ is the streamwise turbulence intensity. This means that, in theory, at least in Region 2, the 10-minute wind speed will
underestimate the turbine power, and this underestimate depends on the square of the turbulence intensity. This would hold for any convex curve relationship where $\overline{P} \propto U^n$ and $n > 1$. Where $n < 1$, it is straightforward to show from a Taylor expansion that an increase in $I_u$ would be expected to *reduce* the expected power. This latter case is seen in Region 2.5, where the turbine controller deviates from optimum aerodynamic efficiency. There have been several studies that have borne this out, showing that turbine power output is increased as $I_u$ increases at low wind speeds above cut-in. Power output decreases with increasing
$I_u$ at higher wind speeds just below rated (Honrubia et al., 2012; Dörenkämper et al., 2014; Hedevang, 2014; Bardal et al., 2015; Sakagami et al., 2015; St. Martin et al., 2016; Lee et al., 2020; Kim et al., 2021; Sakagami et al., 2023); several methods have been proposed to correct power curves for this effect, e.g. (Albers, 2010; Clifton and Wagner, 2014; Hedevang, 2014; Lee et al., 2020), and have generally been found to be effective.

Due to the induction zone effect of the rotor, the wind speed seen by the nacelle-mounted anemometer will not be a true
reflection of the free-stream wind speed. By correlating upstream mast measurements with those of the nacelle anemometer, it is possible to derive a nacelle transfer function (NTF) that can be used for power curve verification to check the performance of a turbine when operating on a wind farm. However, the NTF is sensitive to levels of turbulence intensity (St Martin et al., 2017). The authors speculate that this may be due to the interaction between the turbine and levels of turbulence affecting motion, which impacts rotor induction. However, this has not been verified and would require further research.



The detailed effects of the scale, magnitude, and coherent structure of turbulence on the aerodynamic performance of a wind turbine have had less attention. One of the few studies, Gambuzza and Ganapathisubramani (2021), investigated the sensitivity of the power and thrust coefficients of a model horizontal axis wind turbine (0.18 m rotor diameter) in a wind tunnel to conditions of various turbulence intensities and length scales. The $C_P - \lambda$ curve was determined for different values of $I_u$, showing an increase in peak $C_P$ as $I_u$ was increased. However, this increase was significantly greater than expected from the

dependence suggested in Eq. 10. This was attributed to the dependence of the blade aerodynamic properties on free stream turbulence and the resulting torque generated. The scale of the turbulence was also shown to impact performance, with longer turbulent timescales translating into higher power, which was not seen for higher frequency turbulence. This was assumed to be due to the ability of the rotor to exploit the power of the lower frequency fluctuations, which becomes increasingly difficult given inertia limitations as the fluctuations become faster. A similar wind tunnel study was carried out on a model vertical axis

wind turbine (H-Darrieus type of width 0.5 m and height 0.8 m) (Molina et al., 2019). Here, too, an improvement in peak $C_P$ was noted as $I_u$ was increased, which could not be attributed to the additional power in the turbulent fluctuations expected from Eq. 10. Molina et al. (2019) suggested that this is a consequence of more favorable stall characteristics (stall delay) when the blades were at a high angle of attack. This condition occurs more frequently for a vertical axis wind turbines than horizontal axis wind turbines due to the different geometries. No dependence on turbulence length scale was noted, though length scale

and turbulence intensity were not independently varied in this case, in contrast to Gambuzza and Ganapathisubramani (2021). A further study of a model VAWT very similar to that of Molina et al. (2019), concluded that turbulent length scale did have an impact on performance with length scales less than or similar in size to the blade chord length showing enhanced performance compared to a length scale much greater than the chord which showed significantly lower performance (Peng et al., 2019). Clean flow gave rise to even lower performance. The authors speculate that the small-scale turbulence contributes to boundary

layer re-energization around the blades, delaying flow separation and stall. Other work (Chamorro et al., 2015; Tobin et al., 2015) suggests that such findings may also hold for large-scale utility turbines. However, further work would seem justified, particularly the sensitivity of blade aerodynamics to turbulence when close to stall. There are potentially important differences in how HAWTs and VAWTs respond to turbulence, which requires further investigation, particularly if vertical axis technology is used offshore or in the urban environment.

**7.2   High Frequency Power Oscillations**

The effect of turbulence on the temporal behavior of turbine power output is important from the point of view of power quality. A study combining detailed field measurements with LES simulations (Nandi et al., 2017) concluded that turbine power output was sensitive to the passage of energy-containing atmospheric eddies causing fluctuations on the scale of $\sim$ 10–100 s but was relatively insensitive to the higher frequency fluctuations of $\sim$ 1 Hz or greater associated with the sampling of small-scale

turbulence by the blades.

    One central drawback of the common method to estimate the power output is that all fluctuations caused by the response of the wind turbine to the fast wind fluctuations are averaged out by the common 10-minute mean values. Another approach is to use the fluctuations as response information in the wind energy conversion process. Using highly fluctuating data at 1 Hz or




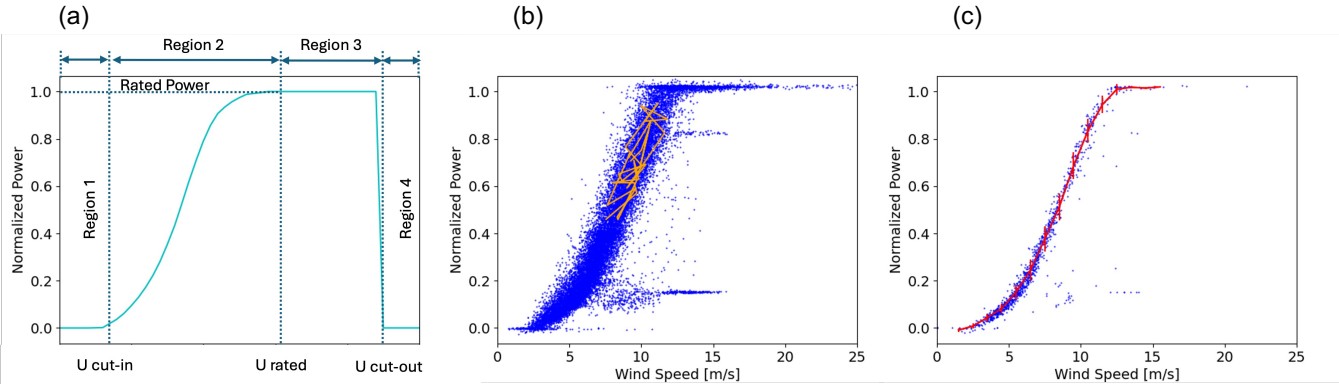

**Figure 13.** (a) Schematic of a wind turbine power curve showing operational regions; (b) power output of a wind turbine as a function of wind speed sampled at 0.5 Hz with orange line connecting a 5 min trajectory of the power wind dynamics (based on data used in Fig. 8 from Mahoney et al., 2012); (c) IEC power curve from 10 min (blue dots) averaged data omitting outliers with error bars (red).

higher, it could be shown that power characteristics can be defined using stochastic process modeling representing the evolution
of a random value based on the Langevin equation. The Langevin approach is independent of the site-specific turbulent wind features Mücke et al. (2015). This methodology can also be extended to load estimations Lind et al. (2014).

Analysis of smaller scales or higher frequencies of the power output of wind turbines shows that the wind energy conversion process is influenced by small-scale turbulence. The power output of even large wind turbines shows fluctuations on the order of a second (Milan et al., 2013, e.g., Table II). The statistical analysis shows that turbulence, including remarkable
features of intermittency (i.e. non-Gaussian statistics), drives power fluctuations. Stochastic models for non-Gaussian frequency fluctuations are presented in Schäfer et al. (2018). Milan et al. (2013) showed that these fluctuations are not averaged out and can significantly impact the power output of entire wind farms. Haehne et al. (2018) analyzed fast frequency fluctuations of the power grid and showed that these are related to wind power fluctuations. Therefore, even country-wide averaged wind power carries the imprint of turbulence.

Power quality is a major challenge for the grid integration of renewable generators (Liang, 2017). The turbulent nature of wind energy led to investigations of the impact of intermittent power feed-in on power grids (Schmietendorf et al., 2017; Auer et al., 2017). Numerical results indicate that intermittency propagates in a power grid and affects the frequency increment distributions of nodes distant to the feed-in. The principle problem and a gap in our ability to develop an effective approach to mitigate the effect of turbulence in high-frequency power fluctuations is that the intermittent, non-Gaussian statistic does
not fall into any well-defined mathematical category. Only phenomenological formulas are known. This results in significant uncertainties when estimating the probability of extreme events of power fluctuations.





## 7.3 Impacts of Certain Atmospheric Phenomena

### 7.3.1 Low-level Jets

Over the past two decades, several billion dollars have been invested in installing large wind farms in the Great Plains region
of the US, including Texas, North Dakota, Oklahoma, and Kansas. Nocturnal LLJs frequently occur in this region, and it is
common knowledge in the wind energy community that LLJs make wind resources more favorable for power production (e.g.,
Kelley et al., 2006; Storm and Basu, 2010; Wimhurst and Greene, 2019). The Wind Forecast Improvement Project report
(Wilczak et al., 2015) stated:

> "Qualitative analysis indicates that the LLJ is a regular, periodic, [...] and dominant feature in the SSA [South-
> ern Study Area in Texas] that drives capacity factors to over 60% (and therefore a large fraction of power produc-
> tion) during the nocturnal hours."

A few modeling studies have attempted to quantify the importance of LLJs more systematically. Cosack et al. (2007) used
observed wind profiles from a SODAR deployed in Germany as input for the FLEX5 aeroelastic simulation tool. Using 1.5
MW and 5 MW wind turbine models, they found that LLJs increase power production, and the effect increases as turbines get
higher. Similar conclusions were reported by Gutierrez et al. (2016) and Weide Luiz and Fiedler (2022). Gutierrez et al. (2016)
used high-frequency measurements from a 200-m tall met-mast near Lubbock, Texas, as input for the FAST (Jonkman and
Sprague, 2024, now OpenFAST) model to model power production. Weide Luiz and Fiedler (2022) used power curves from
utility-scale turbines (Enercon E-126 and Vestas V112), in conjunction with lidar observations from two locations in Germany.

Wind turbines cause enhanced turbulent mixing and, therefore, affect LLJ profiles in the presence of wind farms. In recent
work, Gadde and Stevens (2021b) employed LES with actuator disks representing turbines to simulate wind power production
during LLJ conditions. They found that when the nose of the LLJ is above the turbine, the lost production of downstream
turbines is relatively low, as energy can be entrained from the jet above the wind farm. Also, Doosttalab et al. (2020) found that
wind farm energy production can benefit from enhanced air entrainment from the jet. As a result of the momentum extracted
by the turbines, the jet's strength is reduced while the wind shear at lower altitudes is affected (e.g., Abkar et al., 2016; Larsén
and Fischereit, 2021; Quint et al., 2025). Large wind farms can deflect part of the jet over the wind farm, as large wind farms, in
general, deflect flow; see, e.g., the mesoscale modeling study of Larsén and Fischereit (2021) and idealized LES cases. When
the jet nose lies below the turbine hub, turbulent kinetic energy from the jet is transported upwards instead of downwards, such
that downstream turbines can benefit from the LLJ's energy (Gadde and Stevens, 2021a).

### 7.3.2 Mesoscale Convective Circulations - Rolls and Cells

The fluctuations in wind speed associated with the presence of convective cells represent great challenges for power balancing
systems (Akhmatov et al., 2007; Sørensen et al., 2008). Sørensen et al. (2009) compared the wind power from the offshore
farm Horns Rev and onshore turbines on 18 January 2005, showing that the offshore power fluctuates sharply while it is stable
onshore. Open cells were present that day over the North Sea following a frontal passage.



Vincent (2010) systematically studied the intra-hourly fluctuating wind characteristics in connection with open cells and their impact on offshore wind power for the North Sea region. Larsén et al. (2013) showed that a collection of 18 cell cases have spectral energy five times larger than the climatological mean value in the range of about $10^{-4}$ to $10^{-3}$ Hz. Imberger et al. (2013) showed that the distribution of the wind speed change, $\delta_U$, between two consecutive 10 min values in the presence of open cells suggests significant changes and deviation from those in the absence of cells. Vincent and Trombe (2017) reviewed the topic of forecasting the intra-hourly variability of wind generation. Göçmen et al. (2020) compared the wind power variation of the Horns Rev farm in the presence and absence of open cells using 10-min mean and 1-min SCADA data. While the mean wind conditions are comparable for the two cases, the turbulence intensity, variation in wind direction, and power fluctuation are systematically and significantly larger in the presence of open cells than in the absence of them.

The organized structures such as cells not only bring lare18 fluctuations at one single point, they also provide spatial variations and coherence (Vincent et al., 2013; Larsén et al., 2013; Larsén et al., 2019), which will also challenge some of our methods where standard coherence models are used e.g. in connection with power balancing.

Cells contribute to the intra-hourly variability of the wind and power and the microscale regime. Larsén et al. (2019) analyzed the turbulence characteristics of wind during a cell event using sonic measurements from two stations. They showed that the open-cell-related variability tends to fill the spectral gap with extraordinary energy, affecting the power spectrum from about 0.0002 to 0.01 Hz. Accordingly, using typical boundary layer models, even considering scaling with boundary layer height, significantly underestimates the turbulence intensity. The sizes of open cells have important implications and challenges for the methodologies used in Wind Energy. Currently, when calculating turbulence-related parameters such as load, turbulence intensity, gust, and wake meandering, only scales smaller than the spectral gap are considered using typical boundary-layer models such as Kaimal et al. (1972), Mann (1998) and Veers (1984). The spectral gap is typically assumed to be at a wavelength of 1 km or a period of 10 min – 1 hour. The concept of the gap has been used to filter out larger scale variability (Larsén et al., 2016). Modern wind farm clusters are expanding to areas of thousands square kilometers (e.g., Hornsea area 7240 km$^2$, Minnesota wind farms about 5000 km$^2$), and individual wind turbines are more than two hundred meters tall (e.g., the Haliade-X turbine, 260 m). The flow through such large wind farm clusters involves both micro and mesoscales. With a typical roll width of 2 km to 10 km and cell diameters from a few kilometers to 40 km, it is expected that they both can affect the performance of turbines and wind farm clusters.

While several studies addressed the effects of convective cells on wind energy production focused on the wind farms in the North Sea, the effect of mesoscale convective rolls has not yet been studied. Cold air outbreaks along the US East Coast result in convective rolls, which are evident in satellite images as cloud streets. Atmospheric conditions under which convective rolls occur are characterized by a lower magnitude of the stability parameter, the ratio of mixed layer height, $z_i$, and Obukhov length, $L$, in comparison to convective cells ($|z_i/L| < 20$) as a result of stronger winds and shallower mixed layer. Future studies should focus on the impact of these flows on power production.





### 7.3.3 Gravity Waves

A modeling study of a hypothetical large offshore wind farm suggested that when an AGW of the same size as the wind farm was present, a swing in power production from the highest producing turbine to the lowest of 76% was possible compared to a control situation with a similar upstream wind speed with wake effects only where a swing of just 29% was observed (Ollier et al., 2018). Studies of the impact of AGWs on operating wind farms are scarce, which is surprising considering their prevalence (Nappo, 2013). One of the few such studies looked at a wind farm downstream of the Cascade Range mountain chain in the US (Draxl et al., 2021). In this study, measurements from turbines within the wind farm were compared with mesoscale model simulations. Figure 14 shows the results for one such wind turbine. As the turbine was operating close to rated power, significant fluctuations in both wind speed and power production were observed. Due to the presence of AGWs the aggregated wind farm output fluctuated by 11%.

In addition to experiencing the effect of 'naturally occurring' AGWs, (Smith, 2010) proposed that the wind farms themselves could induce gravity waves which can affect their performance. Smith (2010) developed a linear quasi-analytical model of the effect of a large operating wind farm on atmospheric flow due to turbine drag, resulting in the exchange of momentum with the free atmosphere and generation of gravity waves under stable atmospheric stratification. The combined resistance presented by an array of turbines due to their thrust acts similarly to an orographic feature. More recent research has used LES models to investigate the coupling between AGWs and semi-infinite wind farms, wind farms finite in the downstream direction but infinite in the crosswise direction (Wu and Porté-Agel, 2017; Allaerts and Meyers, 2017, 2018). It has been shown (Allaerts and Meyers, 2017) that a large wind farm can form an internal boundary layer that displaces the ABL aloft exciting AGWs.

Numerical simulations predict significant pressure fluctuations that adversely impact wind farm efficiency. Wu and Porté-Agel (2017) suggested that the first row of turbines in a wind farm can experience greater than a 35% reduction in power output in the presence of a strong inversion. However, turbines at the rear of the wind farm may see an increase due to an advantageous pressure gradient accelerating the wind speed. Allaerts and Meyers (2019) developed an extended and improved version of a model developed by Smith (2010) using three layers to investigate the coupling between wind farm layout and AGWs. The work suggested gravity wave effects are small for very wide or long wind farms. Lanzilao and Meyers (2021) used the three-layer model developed in Allaerts and Meyers (2019) to explore how advanced wind farm control could mitigate the impact of wind farm-induced AGWs on performance, showing that gains of up to 14% could be achieved in certain cases. While numerical studies point to the potentially significant impact of wind farm-induced AGWs on wind farm performance, the significance and impact of these effects on power production have not been confirmed yet by observations. One of the goals of the recent field study, the American Wake Experiment (AWAKEN) is to study and characterize the effect of AGWs on wind power production Moriarty et al. (2024).

Modeling of wind farm-induced AGWs is still a challenge. Avoiding nonphysical wave reflections and their amplification requires very large domains and damping layers. The most optimal way to do this has not been determined, leading to time-consuming simulations with no guarantee that numerical artifacts will not affect the validity of the results. A limited range of atmospheric conditions and wind farm layouts has been investigated. A much wider range of investigations is required to gain a





better insight into the coupling between AGWs and wind farm performance, combining detailed measurements from satellites and lidars with high-resolution mesoscale and microscale modeling. A better understanding of the impact of both naturally occurring and wind farm-induced AGWs on loading and performance is required. However, since spatial fluctuations in wind speed across a wind farm are frequently significant, one of the key challenges is distinguishing the impact of AGWs from other atmospheric phenomena. Access to wind farm data will also be key for model validation and investigating the extent to which

AGWs impact wind farm generation and loads onshore and offshore.

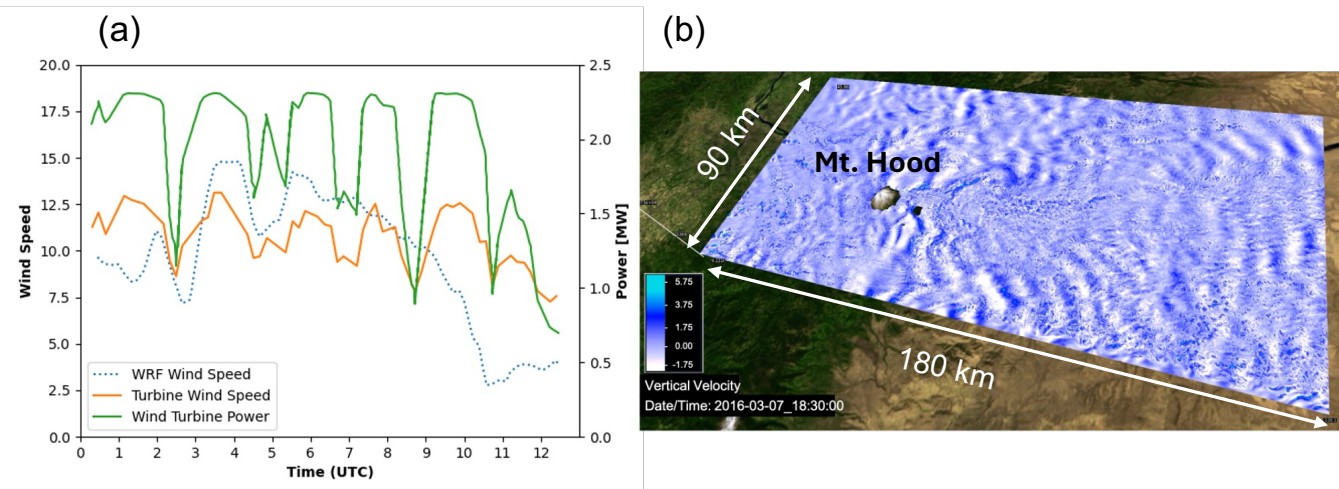

**Figure 14.** (a) Comparison between observations and mesoscale model (WRF) simulations for a wind turbine within a wind farm experiencing AGWs (adapted from  Draxl et al., 2021, licensed under CC BY 4.0 license). (b) LES of a flow in the Columbia River Gorge area, during WFIP2 field study, resolving atmospheric gravity waves (courtesy of Scott Pearse and Pedro Jiménez).

## 7.4  Topographic Effects

A large fraction of wind energy capacity has been deployed in regions with excellent wind resources in relatively flat, homogeneous terrain or offshore. The projected growth of wind power deployment worldwide will, by necessity, lead to more wind farms being developed in complex terrain. In general, we can define complex terrain as terrain characterized by heterogeneity

of topography or land use, possibly including forested terrain. Most of the studies of flows in complex terrain for wind energy applications focused on complex topography, and therefore, in what follows, we will focus on topographic effects on wind power production and wind turbine loads.

Some of the first studies of flows in complex terrain related to wind energy applications were conducted in the early to mid-1990s (Taylor and Smith, 1991; Stefanos et al., 1994, 1996) and focused on measurement and steady numerical simula-

785 tions of topographic wakes. These studies were limited in scope and complexity due to limitations in observing systems and computational resources. Over the last 30 years, the development of remote sensing technology (e.g., doppler lidars) and high-performance computing now enables more detailed studies of flows in complex terrain. In contrast to RANS, which was used in



early studies (e.g., El Kasim and Masson, 2010), nowadays, LES can better account for terrain-induced circulations combined with the effects of atmospheric stability, often resulting in highly unsteady flow. Yang et al. (2021b) reported on one of the first LES of a flow through a utility-scale wind farm in complex terrain, the Invenergy Vantage wind farm in Washington State, US. They used the Virtual Flow Simulator (VFS-Wind) code and computed mean power and the root-mean-square (RMS) of power fluctuations. The agreement between simulations and observations indicated that LES could be used to optimize wind farms sitting in complex terrain. Similarly, Tabas et al. (2021) found good agreement between simulated and measured wind farm power when using the RANS model WindSim. However, they also found that the results were sensitive to model parameters. Because LES models resolve large turbulent eddies, it is likely that LES results would be less sensitive to the choice of turbulence model parameters than RANS.

Although there is a relatively small number of LES of flows through wind farms in complex terrain, these simulations span a range of configurations, from idealized LES of flow over two-dimensional hills and valleys between them Liu and Stevens (2020, 2021) to simulations of observed flows through wind farms in complex terrain. Complex topography can result in a wide range of flow configurations, particularly when combined with atmospheric stability effects. In addition mountain-induced AGWs discussed earlier, some of the flow phenomena that can have a significant impact on the level of turbulence and, therefore, on turbine loads and wind power production are: wind speed up on ridges, topographic wakes, up and down valley flows, cold pools, gap flows, etc. These flow phenomena can result in elevated turbulence or intermittent turbulence levels that can significantly impact turbine performance through power production and increased turbine loads. Turbulence can also play an indirect role in power production by mixing out cold pools characterized by low wind speeds, thus inducing wind ramps. The diversity of phenomena represents a challenge for developing a general approach to characterizing turbulence in complex terrain and its impacts on wind turbine and wind farm performance.

As previously indicated, several field studies have focused on better characterization of topographic effects for wind energy applications in recent years. For example, the WFIP2 field study occurred in the Columbia River Gorge area east of the Cascade Range, where more than 5 GW of wind power is already deployed. This is affected by the mountain wake from Mt. Hood, a cone-shaped, volcanic mountain rising 3,000 m above the surrounding landscape. Under favorable atmospheric stability conditions, von Kármán vortices are frequently shed from islands and isolated mountains like Mt Hood. Mountain wake-induced wind speed reduction or even wind reversal associated with von Kármán vortices directly impact wind power production in the lee of mountains and islands. Under stably stratified atmospheric conditions, topographic effects include forming cold pools in valleys. Cold pools are decoupled from the flow aloft and can persist for extended periods (McCaffrey et al., 2019). Because cold pools are associated with reduced wind speeds and reduced wind power production, accurate forecasting of cold pool formation, maintenance, and eventual mix-out is essential for effective wind resource assessment and wind power utilization (e.g., Olson et al., 2019; Bianco et al., 2019). Arthur et al. (2022) demonstrate that more accurate prediction of cold pool mix outs could be achieved by a mesoscale model accounting for horizontal turbulent mixing using a three-dimensional planetary boundary layer parameterization (3D PBL Kosović et al., 2020; Juliano et al., 2022).

Clifton et al. (2022) recently reviewed research challenges for wind energy in complex locations, including complex terrain, and concluded that focused research and development effort is required to ensure that deployment of wind energy at complex





sites is competitive. To enable more rapid progress, they recommend developing new frameworks for sharing data and test facilities in complex terrains and diverse climate zones spanning a wide range of weather patterns. Furthermore, they point
out that there is no widely accepted definition of "complex terrain" for wind energy applications. The recent IEC 61400-1 standard (IEC, 2019a) categorizes complex terrain as low, medium, and high complexity as a function of slope angles and terrain variance. As challenges, Clifton et al. (2022) point out, there is a lack of standards for power performance testing at complex sites because the free-stream wind speed cannot be identified easily. In addition, wind resource assessment and wind power forecasting are challenging due to the numerical resolution requirements.

## 830   8   Impact of Turbulence on Loads

An understanding of the expected range of turbulent inflow conditions is essential in the design of wind turbines. Turbulence is the principal driver of fatigue, and its spatial and temporal structure has a significant impact on the lifetime of a machine. Response of different materials to turbulence vary significantly, and there needs to be a robust way to calculate the expected lifetime of the different wind turbine components operating in any situation. Calculations of parameters such as damage equiv-
alent load (DEL) are generally based on relatively simple parameterizations of the wind inflow with constant wind shear and veer and a constant mean flow, possibly with a simple stochastic turbulence model or a deterministic gust profile superimposed. However, the turbulent properties of real inflow can deviate significantly from these simple assumptions. Phenomena such as low-level jets, intermittent events, downbursts, or typhoons can create transient turbulent inflow fields that differ markedly from the mean and cannot be captured using simple turbulence models or gust profiles.
In the context of atmospheric turbulence and determining its impact on loads, the standards (IEC 61400-1 (International Electrotechnical Commission, 2019a) and IEC 61400-3-1 (International Electrotechnical Commission, 2019b) for offshore wind energy) play a significant role. The purpose of these standards is to provide a standardized set of definitions, rules, requirements, and guidelines for wind energy applications. In principle, the standards have two major objectives:

1. To classify turbines according to a given set of design load cases dependent on external conditions, and

2. To characterize the local external site conditions, i.e., to verify that the local conditions are not more severe than the turbine class selected.

Design load cases are specified for a large range of wind turbine operational conditions, e.g., power production, fault, start-up, parked, etc., under different environmental conditions determined to be either normal or extreme. The latter is furthermore specified for different gust situations and extreme turbulence. Specific guidelines are given for the generation of synthetic
turbulent inflow.
  The standardized wind conditions and models in the IEC standards are subject to several simplifications, which, given today's huge turbines with rotor diameters of more than 200 m and hub heights of the same order of magnitude, seem too simplistic. The most important simplifications is the prescription of a constant shear exponent across the full rotor as described in Eq. 6 with zero veer across the rotor. Turbulence is assumed to be homogeneous, conforming to either the Kaimal (Kaimal et al.,





1972) or Mann model (Mann, 1994), which does not account for non-neutral atmospheric stability and whose applicability at
higher altitudes more typical of today's machines is not proven.

Actual conditions at turbine sites need to be better understood and characterized since, in some situations, fatigue damage
could exceed values calculated using the assumptions of the prescribed models in the standards. The following sections look
in more detail into the shortcomings of the IEC standards and describe several of the phenomena not (fully) accounted for in
the standards whose turbulent properties require further research.

### 8.1    Limitations of the Design Standards

This section begins by assessing the validity of the assumptions made in the IEC standards in the context of a well-established
literature on boundary layer meteorology, atmospheric turbulence, and wind engineering (Panofsky and Dutton, 1983; Oke,
1987; Stull, 1988; Simiu and Scanlan, 1996; Arya, 1999, 2001; Holmes, 2007). The IEC standard for wind turbine design
recommends two turbulence models to account for atmospheric turbulence: the Mann spectral tensor model (Mann, 1994) and
the Kaimal spectral and exponential coherence model (Kaimal et al., 1972). The corresponding Kaimal and Mann spectral
formulations are stationary models designed for neutral atmospheric conditions in the surface layer. These models and the
employed parameters, which are based on measurements from relatively short meteorological masts (e.g., Kaimal et al., 1972),
have limitations in presenting inflow conditions accurately, especially for modern turbines that nowadays operate outside the
surface layer.

Based on measurements collected from a flat, uniform field site in Kansas, Kaimal et al. (1972) proposed several generalized
spectra (and co-spectra) functions for various variables. The velocity spectra exhibit a pronounced and systematic dependence
on measurement height, wind speed, and atmospheric stability (see also Panofsky and Dutton, 1983). The study derived simpli-
fied velocity spectra functions for neutral conditions from the more general ones to facilitate comparisons with wind tunnel and
other observational data. These simplified functions were solely dependent on height and wind speed. In subsequent work, In-
ternational Electrotechnical Commission (2005) further modified and simplified these longitudinal velocity spectra for design
purposes. However, it is well-documented that the characteristic sizes of turbulent eddies decrease as atmospheric stratification
increases (e.g., Stull, 1988). Consequently, coherence typically diminishes more rapidly under stably and neutrally stratified
conditions than unstable conditions (Panofsky and Dutton, 1983). However, the IEC exponential coherence model International
Electrotechnical Commission (2019a) does not incorporate such stability-dependent variations. Furthermore, for the sake of
simplicity, the IEC guidelines assume that the integral length scale remains unchanged with height for hub height $z_h > 60$
m within the turbine rotor layer, resulting in spectra that no longer account for height dependence. While simplifying height
dependence was a reasonable approximation for smaller rotors (around 20 meters), this assumption is not physically realistic
for modern rotors.

According to IEC standards, standard industry tools such as Turbsim (Kelley and Jonkman, 2007) and the Mann turbulence
generator (Dimitrov et al., 2024, e.g.,) provide stationary Gaussian wind fields with uniform variance and predefined spectrum
and coherence properties. However, wind turbines in the field are subject to atmospheric turbulence, which can significantly
affect their power output, loading, and fatigue life. For high Reynolds number turbulent flows as observed in the atmosphere,





the pdfs of velocity increments portray strong non-Gaussian behaviors (e.g., heavy tails) for small spatial/temporal separations
(Frisch, 1995; Bohr et al., 1998; Basu et al., 2007; Mücke et al., 2011). For large values of separations (on the order of the
integral scale), these pdfs asymptotically approach the Gaussian distribution. The scale dependence of the velocity increment
pdfs, a characteristic feature of the multifractal cascade dynamics of turbulence, should be accurately reflected in inflow as this
turbulence structure is crucial for understanding the structural loads and performance of the turbine under realistic conditions.
Preliminary LES study suggests that a large, utility scale turbine acts as a low-pass filter and therefore the non-Gaussian effect
alone does not impact significantly fatigue loads of the turbine (Berg et al., 2016).

Considering the above facts, it becomes apparent that contemporary turbine inflow generation approaches have many lim-
itations and that a paradigm change is needed. Working towards this goal, researchers from NREL developed the TurbSim
(Jonkman, 2009) stochastic inflow turbulence tool. This tool has been developed to provide a numerical simulation of a
full-field flow that contains coherent turbulence structures that reflect the proper spatiotemporal turbulent velocity field re-
lationships seen in instabilities associated with nocturnal boundary layer flows and which are not represented well by the IEC
Normal Turbulence Models. TurbSim provides the ability to efficiently generate randomized coherent turbulent structures that
are superimposed on the more random background turbulent field as produced by one of the non-neutral spectral models.

LES is an attractive way to generate inflow winds with non-Gaussian statistics as it simulates unsteady, anisotropic turbulent
flows dominated by large-scale structures and turbulent mixing that are all characteristic of the atmosphere (Berg et al., 2020).
Many atmospheric phenomena, including thermal stratification, can be well represented in LES. A comparison with field
data shows that high-frequency content is typically not fully represented in LES and requires even higher fidelity modeling.
Additionally, in idealized canonical LES cases, lower frequencies are underestimated compared to field measurements, as the
impact of mesoscale atmospheric dynamics (e.g., mesoscale convective circulations) is not accounted for. Meanwhile, it has
been shown that accounting for the lower frequencies is important, especially offshore and for bigger rotors. Comparisons
between IEC model spectra and LES and field measurement data reveal that IEC models capture neutral stratification better
than stable and unstable conditions.

Point statistics, such as wind speed and turbulence intensity at hub height, significantly impact a wind turbine's response.
However, for larger rotors where the spatial and temporal distribution of the wind field becomes increasingly important, factors
like wind shear and coherent structures become more important. The absolute coherence can be expressed as $\gamma = \frac{|S_{xy}|}{S_{xx}S_{yy}}$,
where $S_{xx}$ and $S_{yy}$ are one-sided auto-spectra of the wind velocities at two different positions, $x$ and $y$, and $S_{xy}$ represents the
cross-spectrum between these positions. The cross-spectrum can be divided into a real part (co-coherence) and an imaginary
part (quad-coherence). The common approach in the wind energy industry neglects quad coherence as it is generally assumed
to be less significant than co-coherence. While quad-coherence is generally smaller than co-coherence, it is certainly not
negligible. This assumption may be significant in terms of the loads experienced by a wind turbine blade, as turbulent velocity
fluctuations at a certain frequency may not exhibit the same phase along the length of the blade. While the Kaimal model
does not account for quad coherence, the Mann model includes quad coherence. Turbsim uses the Davenport coherence model
(Davenport, 1961), which neglects quad-coherence, however, Hipersim (Dimitrov et al., 2024) provides turbulence generation
capability based on the Mann model.





In recent years the Man model has been extended to account for a wider range of atmospheric phenomena that impact wind
turbine performance. Segalini and Arnqvist (2015) extended Mann model to account for stable stratification, while Chougule
et al. (2017) further extended the Mann model to account for the full range of atmospheric stability and introduced two new
parameters, gradient Richardson number, $Ri$, and temperature variance destruction, $\eta_\theta$. They compared statistics of uniformly
stratified and sheared model generated turbulence to observations from the HATS field study (Horst et al., 2004). Since these
models did not account for low frequency fluctuations induced by two-dimensional, mesoscale motions, Syed and Mann (2024)
extended the Mann model to include length scale corresponding to the peak of mesoscale turbulence, the boundary layer height
and the variance due to two-dimensional, low-frequency fluctuations in addition to the anisotropy parameter. Mann model was
also extended to space-time domain (de Maré and Mann, 2016; Guo et al., 2022). These are important developments toward
better characterization of turbulence in an ABL that warrant further studies to evaluate how these models perform under a range
of conditions.

Although new developments led to more complete and realistic representation of atmospheric conditions, several phenomena are still absent in current inflow models that are likely to impact design loads significantly. For instance, design inflow
conditions do not account for veer. Despite the numerous identified inaccuracies and instances of incomplete or missing data
in current inflow models, the industry has not experienced frequent failures of primary structures. This suggests that assumptions of overly energetic inflow turbulence and generous safety factors have led to conservative designs. However, the scale of
modern-day wind turbines is such that inaccuracies in existing inflow models may be inadequate for cutting-edge design and
optimizing safety margins. For further insights into how inflow conditions influence wind turbine design, we refer to Veers
et al. (2023). Furthermore, we note that extreme atmospheric turbulence events, such as hurricanes, thunderstorms, and downbursts, exhibit spatial and temporal characteristics that differ significantly from IEC design load cases, necessitating separate
considerations.

Achieving better estimates of boundary layer turbulence impact on fatigue loads and wind power production requires better
characterization and representation of turbulence as modulated by complex atmospheric conditions at elevations of modern
wind turbines (i.e., outside the surface layer). This should also include offshore environments. Improving the representation of dynamically evolving turbulent flows in high-resolution simulations (i.e., LES) necessitates including low-frequency,
mesoscale-induced fluctuations. This can be achieved by going beyond idealized setups and embracing fully coupled mesoscale
to microscale simulations discussed in section 3.7.

### 8.2 Impact of Atmospheric Phenomena on Fatigue Loads

The fatigue loading experienced by turbines is a crucial design consideration outlined in the IEC 61400 standard for wind
turbine safety (International Electrotechnical Commission, 2019a). The IEC provides certified standards for various welldefined turbine types. Before construction, it needs to be verified that the site-specific conditions are within the limits of the
turbine type certification. Wind conditions at a site are specified by extreme wind speed, vertical wind shear, flow inclination,
turbulence, and rare gust-like events. The load type is either an ultimate load that damages the turbine or a fatigue load. While





the turbine classification is the responsibility of turbine manufacturers, the site assessment is in the hands of project developers and requires insight into atmospheric turbulence dynamics.

The wind conditions are determined using simplified models, scaled by the hub height values, providing a reference wind speed, and turbulence intensity. The effective turbulence intensity is an *ideal* turbulence intensity, independent of wind direction, to characterize a site such that fatigue damage is expected to be representative of the wind conditions encountered at the site. However, atmospheric turbulence and wind speeds typically depend on wind direction and atmospheric stability. They are often correlated due to upstream conditions at the site or in relation to certain weather phenomena. On the other hand, turbine damage is highly non-linearly related to loading amplitudes and, therefore, to turbulence intensity. As a result, a few extreme events may cause most fatigue damage, and the concept of an effective turbulence intensity fundamentally can not capture the richness of atmospheric turbulence. Furthermore, turbulence generated by upstream turbines impacts downstream turbines. The work by Frandsen (2005) led to the incorporation of wake added turbulence of neighboring turbines to the effective turbulence intensity, and a simple wake turbulence model is provided for this.

Wind turbine wakes affect the dynamic loading on the turbines due to increased turbulence, the created wind speed deficit, and changes in turbulence structure. Thomsen and Sørensen (1999) used the Vindeby experiment combined with the aeroelastic code HAWC to demonstrate that the turbulence intensity, length scales, and coherence are parameters that strongly affect turbine loading. They found that loads for a turbine installed in an array or a farm are approximately $5\%$ and $15\%$ higher than those in free stream conditions, and the increase is similar onshore and offshore. Turbine loads are typically higher onshore than offshore as turbulence intensity is typically lower offshore Dahlberg et al. (1992). This is also supported by LES studies conducted by NREL Churchfield et al. (2012) and Lee et al. (2013), which capture the structural load experienced by wind turbines by using the FAST model coupled to the LES. They found that increased surface roughness, typical for onshore conditions, increases damage equivalent loads on the turbines. Furthermore, they showed that downstream turbines typically experience more significant damage than loads. However, the in-plane blade root moments for downstream turbines are lower due to the lower wind speed.

Thomsen and Sørensen (1999) showed that fatigue loads are higher when the turbulence spectral length scale is smaller. Riziotis and Voutsinas (2000) showed that yaw misalignment can increase fatigue loads. Furthermore, they state that turbulence length scales are smaller in complex terrain, which is why turbine fatigue loads tend to be higher in complex terrain. The Vindeby Frandsen and Thomsen (1997), and Sexbierum Adams (1996) experiments showed that fatigue loads are similar in single and multiple-wake situations, which supports the view that the wake added turbulence intensity of the closest turbine is most relevant. However, it should be noted that LES has shown that, depending on the wind farm layout, turbulence levels in the wind farm can increase with the downstream direction for multiple turbine rows.

Sathe et al. (2013) and Holtslag et al. (2016) showed that the lifetime fatigue loads experienced by a wind turbine depend strongly on the joint probability distribution of hub height wind speed and atmospheric stability. Typically, IEC standards are followed, and atmospheric stability is neglected. Lee et al. (2018) showed that the relative positioning of turbines matters when loading is concerned. Displaced downstream turbines that experience partial wake impingement are subjected to significantly larger fatigue loads compared to a fully waked turbine. Lee et al. (2018) found similar impact on a displaced downwind floating





wind turbine. Englberger et al. (2020) used LES to study what controls downwind wake deflection and found the blade rotation when combined with directional shear (wind veer or backing) result in a significant wake deflection. These combined effects must be taken into consideration when estimating partially waked turbine fatigue loads.

We conclude that we must adequately revise the wind turbine design and safety standards to account for increased fatigue loading in extended wind turbine clusters. The increasing wind turbine size and height require further exploration of the atmospheric flow conditions at higher elevations (100-400 meters). Recent studies based on tall tower and lidar observations are addressing this gap (e.g., Lundquist et al., 2017; Pichugina et al., 2017; Peña et al., 2021), but better characterization of shear, veer, and turbulence in this layer is still needed. The safety standards that are currently employed result from limited

insight and characterization of other turbulent flow characteristics induced by complex topography or extreme events. Given that the impact of turbine loading is highly non-linear, additional fundamental insight into wake-wake interactions is required to improve predictions of the turbulence conditions inside extended wind farms or the interaction between wind farms.

### 8.2.1   Low-level Jets

More than twenty years ago, Neil Kelley and his collaborators began investigating the impacts of LLJs on wind turbine loads.

They incorporated some of the characteristics of observed LLJs into the popular turbine inflow generation code called TurbSim (Jonkman, 2009). Their findings were presented in several publications, including a comprehensive report by Kelley et al. (2004), which provides particularly insightful information. Almost a decade later, Park et al. (2014, 2015) conducted a follow-up investigation on the impacts of LLJs on wind turbine loads. Instead of using synthetically produced stochastic inflows (e.g., via TurbSim), they used inflow data generated through LES to investigate the impact of realistic inflows on various turbine

loads, including out-of-plane bending moment, bending moment causing deformation in direction perpendicular to the rotor plane, and tower-top yaw moment causing the tower to rotate around its vertical axis. All the LES runs included dynamically evolved LLJs. Based on these LES-based inflows, Park et al. (2014, 2015) computed wind turbine loads using the FAST model. For comparative analysis, neutrally stratified inflow fields were generated using TurbSim following the prescriptions by IEC (e.g., Kaimal spectra). The impact of LLJs and associated strong shears (both speed and directional shears) on turbine loads is

evident in Fig. 15.

Gutierrez et al. (2016) utilized high-frequency measurements from a 200-m tall met-mast in Texas to study the impact of LLJs on wind turbine loads. Their findings indicate that LLJs caused increased static and mechanical loads and enhanced fatigue cycles. In a subsequent study, Gutierrez et al. (2017) analyzed blade, nacelle, tower motions, tower forces, moments, and blade loads, in the presence of LLJs. They concluded that negative wind shears (when the LLJ nose is below the hub height)

improved the mechanical loading for the nacelle and tower and recommended using taller turbine towers to take advantage of the benefits of harvesting LLJ-enhanced energy.

Zhang et al. (2019) employed the engineering LLJ inflow model and the von Kármán spectra model as input for the FAST model to show that LLJs increase the aerodynamic loads on the turbine. Gadde et al. (2021) used LES and showed that the shear causes significant azimuthal variation in the external aerodynamic blade loading, increasing fatigue loading on the

turbines. Chatterjee et al. (2022) used mesoscale-driven LES coupled to an aeroelastic solver to demonstrate that the strong



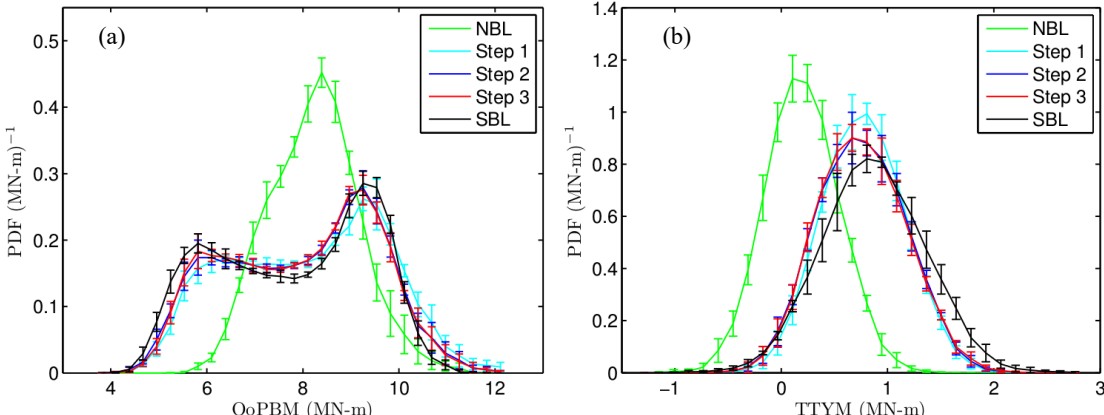

**Figure 15.** Probability density functions (PDFs) of simulated wind turbine loads utilizing IEC-NBL (green lines) and LES-SBL (black lines) inflows representing out of plane bending moment (OoPBM, panel (a)) and tower top yaw moment (TTYM, panel (b)), respectively. Load calculations were performed using the FAST model on a hypothetical 5-MW turbine developed by NREL. The IEC-NBL inflows were generated using a stochastic inflow generation code called TurbSim (Jonkman, 2009). For LES-SBL inflows, a pseudo-spectral LES code employing a dynamic SGS model (Basu et al., 2008) was used to generate the inflow fields and associated LLJs. The hub-height wind speed values from the IEC-NBL and LES-SBL inflows are closely matched (source: Park et al., 2015, published under CC BY 4.0 license)).

veer associated with LLJ significantly increases the damage equivalent hub-loads and tower loads of isolated turbines and turbines placed in wind farms. In wind farms, the reduction of the LLJ strength with downstream distance in the wind farm reduces its impact. This highlights that the results obtained from analyzing the impact of LLJs on isolated turbines may not immediately apply to the entire wind farm.

As larger modern turbines are exposed to a wider range of atmospheric conditions, there is a need to better represent the turbulent inflow that impacts their performance. For practical design applications, TurbSim (Kelley and Jonkman, 2007) is currently a tool that can rapidly generate a range of idealized turbulent fields. To meet the new design needs, TurbSim must be extended to provide a wider range of realistic turbine inflows. Turbulent inflow can also be generated using LES. Until recently, LES has been used to study idealized ABL flows. Coupled mesoscale to microscale (i.e., LES) simulations include

parameterizations of radiative transfer, microphysics, and other physical processes in the atmosphere and can represent a full range of dynamically evolving turbulent flows; however, such simulations are computationally expensive. Therefore, there is still a need for a faster alternative. In addition to a tool like TurbSim, recent developments in Artificial Intelligence and Machine Learning (AI/ML), in particular state-of-the-art physics-informed deep learning approaches, provide an opportunity to develop ML models using vision transformers that could generate realistic turbulent fields at a fraction of a cost of an LES (e.g., Stengel

et al., 2020). Alternatively, coupled simulations could be used to create a public database of ABL flows, similar to the Johns Hopkins Turbulence Database (Johns Hopkins University, 2021).




### 8.2.2 Global Intermittency and Coherent Structures

In Section 3.1.8, the global intermittency phenomenon was briefly introduced. Even though these phenomena are often present in the atmosphere, only a few studies have described their impacts on wind turbine loading. Using observational data from the Long-Term Inflow and Structural Test (LIST) project, Kelley (2011) documented severe transient loading events associated with turbulent bursting events (see Fig. 16 for an example). In an LES study, Park et al. (2015) reported the presence of global intermittency in stable boundary layers. They found that these structures led to strong asymmetric forces on the rotor and, in turn, produced increased tower-top yawing moments.

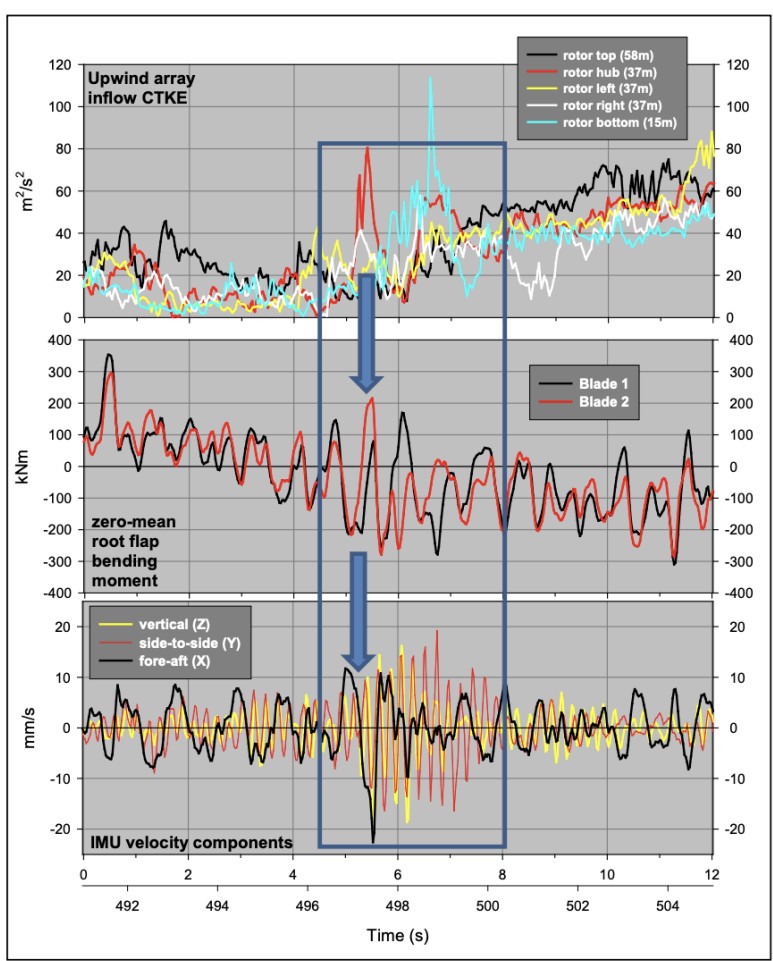

**Figure 16.** Intermittent turbulence-induced severe transient loading event measured on the NWTC Advanced Research Turbine during the LIST project (source: Kelley, 2011).



## 8.3 Wind Farm Generated Turbulence

Wakes in wind farms generate turbulence, which is assumed to supplement ambient turbulence. Empirical models for how this should be calculated and used in fatigue calculations are still very basic. They are predicated on simple ten-minute average measurements from cup or sonic anemometers. One of the most widely used methods, that is still used in wind turbine design standards (International Electrotechnical Commission, 2019a), was developed by Frandsen (Frandsen, 2005). This method was formulated over fifteen years ago when wind turbines were much smaller. It was 'tuned' using a relatively small number of measurements for a handful of what, by today's standards, are considered small wind farms. The model assumes three different cases experienced by a wind turbine in a wind farm, namely exposure to ambient free-stream turbulence, ambient wind farm turbulence (deep within the wind farm), and direct wake turbulence. The model introduces the concept of a representative or $90^{\text{th}}$ centile level of turbulence likely to be experienced by a turbine that can then be used in fatigue load calculations.

There have been few studies to assess the accuracy of empirical models of wind farm induced turbulence. Most of them have only compared with measurements behind single turbines (Crespo and Hernández, 1996) or are based on computational fluid dynamics (CFD) models (Stevens and Meneveau, 2017). One of the few studies to date (Argyle et al., 2018), based on a large offshore wind farm, found that the Frandsen methodology performed well. However, questions remain about its application, including: how to combine ambient free-stream turbulence with that generated by wakes properly; how to account for wake generated variability in turbulence levels, *e.g.*, due to effects such vortex shedding and meandering; whether the concept of a homogeneous ambient wind farm turbulence deep within an array is valid; and how non-neutral stability conditions affect turbulence in wind farms, particularly with regard to how ambient and wake generated turbulence interact.

The fundamental assumption underlying current methods of estimating turbulence impacts, that wind turbine fatigue can be related to point measurements of mean ten-minute turbulence from a cup or sonic anemometer is very simplistic. A more robust measure of turbulence is required, the one which would consider the coherent structure of the inflow. The construction of such a method requires a more detailed assessment of how the 3D inflow affects the loads (and thus the fatigue) on a modern large wind turbine.

## 8.4 Terrain Effects and Loads

Terrain-induced circulations can result in highly variable winds and intermittent turbulence that impact turbine performance. It is, therefore, important to characterize wind gusts in complex terrain. Kawashima and Uchida (2018) investigated the impact of terrain-induced turbulence on blade fatigue loads based on strain measurements. By combining measurements with numerical simulations, they develop a method for optimal turbine placement in complex terrain. Uchida (2018) conducted LES to analyze turbulence intensity impacting a turbine in the Atsumi Wind Farm. The LES output analysis confirmed a higher turbulence intensity induced by the terrain effects. Resolving terrain-induced vortex structure that resulted in high turbulence intensity required a 5 m grid resolution. Uchida (2019) also analyzed the cause of the wind turbine blade damage accident at Shiratakiyama Wind Farm. This study suggested that vortex shedding from the hill located upwind from the wind farm resulted in more intense turbulence that caused blade damage. Hu et al. (2018) analyzed wind gusts in moderately complex ter-





rain and calculated wind gust parameters and their associated parent distribution. They identified a linear relationship between turbulence intensity and gust factors. In a follow-up study, Letson et al. (2019) characterized wind gusts observed during the Perdigão field study and found that gust length scales on ridge tops scale with the height of the ridge and concluded that terrain features impact gust length scales more than their magnitude. In complex terrain, the flow characteristics depend significantly on wind direction due to specific terrain and land cover features and differences in atmospheric stability. Atmospheric stability dependence on the wind direction can be particularly pronounced in coastal environments.

Field studies of turbulence effects can be complemented with wind tunnel experiments. The advantage of wind tunnel experiments is that they allow to vary flow parameters in controlled manner, however, the disadvantage is that the high Reynolds number characteristic for atmospheric flows is not possible to achieve and atmospheric stability and Coriolis effects and their consequences (e.g., shear, veer, LLJs, AGWs) are not replicated. Nevertheless, controlled experiments can yield useful insights into turbulence characteristics in a wind farm (e.g., Kozmar et al., 2018). Vanderwel et al. (2017) designed a wind tunnel experiment to study the effects of surface heterogeneities. They found that surface heterogeneities result in significant differences in the shear stress distribution, which could affect the fatigue life of turbines depending on their placement relative to these heterogeneities.

Considering that it can be expected that wind power capacity deployed in complex terrain will continue to grow in recent years, more attention is being paid to analyzing the effects of complex terrain on wind turbine loads. However, more research is required to better characterize terrain-induced turbulence, including the effects of upwind fetch. This goal can be achieved by analysis of observations from field studies complemented with extensive high-resolution numerical simulations.

## 8.5 Extreme Events and Loads

Wind engineers have studied the effects of high-impact, extreme winds on buildings, bridges, transmission towers, and other structures for decades. Numerous examples are available in several books on wind engineering (e.g., Simiu and Scanlan (1996), Holmes (2007), Solari (2019)). In contrast, the effects of extreme winds on wind turbines and wind farms have been reported only by a handful of papers. A small percentage of these papers documented various damages due to actual meteorological events; others performed idealized numerical simulations and structure load analysis.

Kozmar and Grisogono (2020) argued that the turbulence characteristics of downslope windstorms differ significantly from those in the typical atmospheric boundary layer, and they argued that the current wind engineering standards do not account adequately for the downslope wind storms. This conclusion can be extended to wind turbine design standards as demonstrated by Pehar et al. (2019), who analyzed the measurements of turbulence intensity during a downslope wind event and concluded that their observations do not fall within any of the turbulence classes outlined in the IEC 6100-1 standard. Relative sparsity of observation of downslope windstorms can be overcome using high-resolution, turbulence resolving numerical simulations. However, while LES can quite accurately represent details of downslope windstorms (e.g., Juliano et al., 2024), due to computational complexity, such simulations are presently used as a research tool.

In early 2000, a typhoon made landfall on Miyakojima island, Japan. The recorded peak gust speed was 74.1 m/s. Among the six turbines present, three turbines were completely destroyed, while the remaining turbines suffered significant damage



(Ishihara et al., 2005). A more recent occurrence, documented by Hawbecker et al. (2018), was a thunderstorm featuring multiple downbursts and tornadoes that swept through the Buffalo Ridge Wind Farm in Minnesota, USA, in 2011. The powerful wind gusts not only inflicted damage to turbine blades but also caused a turbine tower to buckle. Tornadoes ripped through Harper County, Kansas, on May 19, 2012 (source: https://www.youtube.com/watch?v=Egdtlnv6Gio, last access: 1 August

2023); five wind turbines were destroyed at a wind farm that was under construction nearby (AbuGazia et al.). In 2017, Hurricane Maria caused significant damage to 13 wind turbines at the Punta Lima wind farm in Puerto Rico. Some blades broke, and others experienced delamination (Kwasinksi et al., 2019). Turbine incidents and failures are underreported due to legal and other proprietary considerations.

In the following subsections, we elaborate on the idealized studies that focused on downbursts and hurricanes.

### 8.5.1 Downbursts

Downbursts are low-level diverging outflows that can generate extreme winds when they occur (Fujita 1985, Wakimoto 1985). Downbursts are often associated with thunderstorms and are dominated by severe horizontal components of the wind field (see the (a) and (b) panels of Fig. 17). However, they are also accompanied by strong vertical motions at low levels. Although downbursts pose a serious threat to wind turbine structures, the IEC or other standards for wind turbine design do not describe

them realistically.

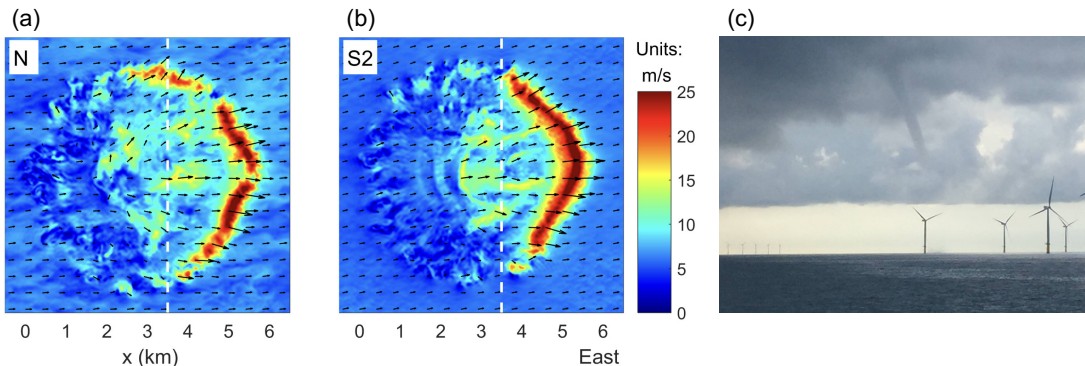

**Figure 17.** (a and b) Downbursts simulated by a pseudospectral LES code employing a dynamic SGS closure. The ambient turbulence condition is neutrally (left) and stably (right) stratified (source: Lu et al., 2019, published under CC BY 4.0 license). (c) A waterspout at Walney wind farms, UK in 2015 (photo by Chris Hall).

Nguyen et al. (2011, 2013) and Nguyen and Manuel (2015a) developed engineering models for simulating velocity fields characteristic of downbursts. The velocity fields were composed of non-turbulent and turbulent parts. The non-turbulent part was generated by using simple analytical models. Meanwhile, stochastic simulation with prescribed power spectral density and coherence was used for the non-turbulent part. The FAST model was used to estimate the tower and rotor loads of utility-

scale wind turbines. The simulated downbursts exhibited strong surface winds coupled with rapid wind direction change. The aeroelastic simulations highlighted the need for blade pitch and nacelle yaw controls to limit turbine loads.



Lu et al. (2019) used a cooling source approach (mimicking evaporative cooling) in conjunction with LES to model downbursts. Compared to the hybrid analytical-stochastic model of Nguyen and Manuel (2015b), the LES-generated velocity fields produced more rapid changes in wind speed and direction. These differences were found to have significant influences on extreme and fatigue loads using the aeroelastic simulation tool, FAST V8 (Jonkman and Jonkman, 2016).

Waterspouts have some morphological similarities with tornadoes and downbursts. Hence, they may induce damage to offshore or coastal turbines. At this point, we are not aware of any such incidents. However, some online images document the occurrence of waterspouts near wind farms; see an example in the Fig. 17 panel (c). As the worldwide growth of offshore wind energy continues, the threats of waterspouts will likely increase at some offshore locations (e.g., the North Sea).

### 8.5.2 Tropical and Extratropical Cyclones

Tropical cyclones (TC), defined as warm-core synoptic-scale cyclones originating over tropical or subtropical waters with organized deep convection and a closed surface wind circulation, can present a damaging series of loads to impacted wind turbines. When the 1-minute average wind exceeds 33 m/s, the TC becomes known as a hurricane or typhoon. Due to significant turbulence levels and associated wind gusts, tropical cyclones can result in turbine loads outside the IEC 61400-3 design standard (Kapoor et al., 2020). In Annex J of IEC 61400-1 (2019), tropical cyclones are addressed based on random sampling from statistical distributions of relevant hurricane parameters from historical track data. While this enables a level of confidence in the statistical prediction of extreme wind for wind turbine design,

A series of studies that analyzed data produced from an LES of a category five hurricane showed that the design standards are inadequate for such a strong hurricane. Worsnop et al. (2017a) showed that this simulation with the Cloud Model 1 (Bryan and Rotunno, 2009, CM1,) well models Hurricane Isabel in terms of mean wind speeds, wind-speed variances, and power spectra. Further work (Worsnop et al., 2017a) demonstrated that the greatest wind speeds occur in the eyewall with peak 3 s gusts exceeding 70 m/s and up to 100 m/s, which would cause extreme aerodynamic and structural loading that could lead to damage or failure, agreeing with analysis of dropsonde wind speeds (Stern et al., 2016). They showed that gust factors can reach 1.7 at the eye-eyewall interface, well beyond what has been considered previously. Rapid changes in wind direction and wind veer during hurricanes can also be quite problematic, increasing both blade and tower loads (Kapoor et al., 2020). Additionally, observational studies with dropsondes (Stern et al., 2016) reveal extreme low-level updrafts in strong hurricanes, with the azimuthal distribution being related to shear orientation and storm movement. Sanchez Gomez et al. (2023) suggested that the IEC standards may be insufficient for defining turbines that could withstand the strongest TCs. This is also the message from Müller et al. (2024b) and Müller et al. (2024a), particularly on Typhoon cases.

Wang et al. (2024a) presented a summary of a workshop conveyed to address needs and challenges related observations, modeling, risk assessment, and climate change impacts of tropical and extratropical cyclones on offshore wind energy in U.S. The workshop highlighted the need for enhanced observational capabilities, coupled atmosphere - wave - ocean modeling, bridging the gap between temporal and spatial scales of weather forecasting and wind turbine operation, and development of effective risk assessment frameworks. A review of relevant work that forms the background for further study appears in Wang et al. (2024b).





Extratropical cyclones, such as a nor'easter along the East Coast of US, develop when cold and warm air collide forming a front and result in a baroclinic instability due to an imbalance in pressure and density levels of a fluid. Maximum wind speed in extratropical cyclones can reach the same levels as in tropical cyclones. As the offshore wind energy is being deployed along the Atlantic Coast of US there is a need to better characterize how extratropical cyclones could impact wind farms.

**8.6   Recommendations**

The projected growth of worldwide wind energy capacity makes the need for better characterization of atmospheric boundary layer wind and turbulence more pressing. The growth of wind energy capacity will inevitably lead to the deployment of wind farms in complex terrain and offshore environments where there is insufficient data and understanding about wind and turbulence characteristics. Furthermore, developing ever larger utility-scale wind turbines presents new design challenges requiring

better characterization of turbulent flow throughout rotor span that may extend above a shallow ABL. Therefore, the projected wind energy deployment goals will require research efforts informing new design standards that would ensure the efficiency and reliability of wind turbines and wind farms. Some main knowledge gaps that must be addressed to achieve projected goals are outlined below.

- Mesoscale-generated turbulence associated with low-level jets, convective cells, convective rolls, and gravity waves are

observed onshore and offshore. The impact on potential wind farm production or turbine loading is less clear, aside from a few isolated examples. A thorough analysis of more wind farm production data is required, coupled with detailed measurement campaigns using remote sensing instruments such as Doppler lidar and numerical modeling to pin down the impact of these different phenomena, particularly the impact of the different time and length scales of the turbulence associated with such mesoscale structures.

- The characterization and quantification of effects of atmospheric stability, non-homogeneity, and non-stationarity on turbulence, coherent structures, and length scales and their impacts on the aerodynamic performance and wind turbine loads is still lacking. More specialized in-field measurements at the blade level are required to understand these impacts fully.

- Presently, most of the turbulence characterization and its impacts on power production and loads are based on the

equilibrium assumption that turbulence production and dissipation are balanced. However, under dynamically evolving atmospheric conditions, the equilibrium is not achieved. Therefore, there is a need to observe and synthesize the effects of non-equilibrium turbulent flows, including intermittent turbulence, on the high-frequency fluctuations of power and loads.

- Improved characterization of extreme wind and ocean conditions, including those associated with hurricanes and topical

cyclones are needed. Values of extreme wind speed and the associated gusts and turbulence levels are often beyond the recommended in the IEC 61400-1 (2019) standard. An important reason for this is the current trend of growing hub





heights of modern-day turbines (the hub height is used as a reference in IEC 61400-1 (2019) ) and, hence, wind speed. Research in the structures of hurricane boundary layers still reveals surprising results, as mentioned in Section 8.5.2.

- The frequency and magnitude of extreme events will likely be different in future climates than what is observed today. Additional research is needed to understand how atmospheric and oceanic conditions will change over the next several decades and more to better inform turbine design. For example, statistical distributions of key hurricane parameters (for example, occurrence rates, geostrophic shear, radius of maximum wind speed, and maximum sustained wind speed) in future climates are needed

- The community needs more validation data to increase our understanding and to evaluate our models. For comprehensive model validation there is a particular need simultaneous observation of the environmental conditions, details of the wind farm operation, and the interactions between the two.

- Most of the field studies described in the literature are episodic and focused on process studies. There is a need for measurements and analysis over an annual cycle, or at least several seasons, to better understand the environment where wind turbines operate. These studies should be conducted in areas with different topographies and land-use/land cover. Considering that the lifetime of a wind turbine should be 20 or more years, there is a need for long-term measurement of turbulence.

- Continuing development of model benchmarks focusing on tests that specifically assess turbulence characterization. An example is recently initiated International Energy Agency (IEA) Wind Task 57 - Joint Assessment of Models (JAM) is a benchmark successor to Wakebench).

- Currently, there are several advanced research tools and computer models under development that utilize advanced computational platforms and algorithms (e.g., GPUs, accelerators, and machine learning) and provide the ability to conduct turbulence-resolving simulations of unprecedented fidelity. There is an opportunity to transition some of these tools from research to applications.

As outlined above, facilitating continued growth of wind energy capacity requires better characterization of turbulence in the ABL under a wider range of large-scale atmospheric forcing including non-stationary conditions, over non-homogeneous terrain, and when turbulence is out of equilibrium. Addressing the gaps in our knowledge would not only contribute to better understanding of turbulent processes in the ABL but also to wider deployment of wind energy.

**Code availability**

The code used in this paper is not publicly available but can be obtained from the author upon request.





## Data availability

The data are publicly available. The data can be obtained at the following link:b https://doi.org/10.1594/PANGAEA.902843, accessed: March 4, 2025.

## Competing interests

SB, JP, and PV are Associate Editors of Wind Energy Science.

## Contributions

All the authors contributed to the conceptualization of the content and the structure of the paper. BK drafted section 1, 2, and 6 and contributed to drafting sections 3, 4 and 7. SB drafted section 3, 4, and 8 and contributed to drafting section 7 and 8. JB contributed to drafting section 7 and 8. LKB contributed to drafting section 3 and drafted section 5. SEH contributed to drafting section 3 and 8. XGL contributed to drafting section 3, 4, 7, and 8. JP contributed to drafting section 3 and 7. RJAMS contributed to drafting section 4, 7, and 8. PV provided leadership in the conceptualization of the paper. SW contributed to drafting section 3, 7, and 8. All authors contributed actively to the editing and review of all sections.

*Acknowledgements.* We thank William Shaw and Jakob Mann for their comments and suggestions on the draft of the manuscript that have significantly improved the manuscript. We also thank Joshua Bauer for the design of Fig. 1 and Etienne Cheynet for sharing the data that Fig. 7 is based on. Sukanta Basu is grateful for financial support from the State University of New York's Empire Innovation Program. Sue Ellen Haupt is partially supported by the NSF National Center for Atmospheric Research, which is a major facility sponsored by the U.S. National Science Foundation under Cooperative Agreement No. 1852977. Xiaoli Larsén was supported by Danish EUDP GASPOC (J. nr. 65020-1043) and Horizon Europe DTWO (101146689) projects.



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
