# Peer review of "Impact of atmospheric turbulence on performance and loads of wind turbines: Knowledge gaps and research challenges"

_Wind Energy Science, 2025_

## Referee Comment (RC1)

Review of manuscript: wes-2025-42 Title: Impact of atmospheric turbulence on performance and loads of wind turbines: Knowledge gaps and research challenges Authors: Kosovic et al

**Overall comments:**

The submitted paper reviews the fundamentals of atmospheric flow from global to microscale, and then zooms in on mesoscale and microscale turbulence and their impact on wind turbines. As turbines continue to grow in size and deployment, this review is timely and important to highlight the need to revisit commonly used assumptions/simplifications. Overall, the paper is well-written and is an especially impressive effort coordinating many authors. There are several areas that can be improved with further integration across the paper, I've noted a few below, where the text repeats or contradicts itself. I hope the authors may consider these comments in a revision.

**General comments:**

- 1. The authors have made a choice to focus the review on ABL effects on wind turbines, rather than wind farms. This is a reasonable choice to keep the paper's scope constrained, but wind turbines are nearly always placed in wind farms, where wake and array level effects will both depend on the ABL in interesting/complex ways and also will change many aspects discussed (i.e. in large wind farms, the effect of ABL turbulence on the loads of the leading row described extensively in Sections 7 and 8 could matter less than the interaction between the ABL flow and wakes/farm-scale effects that will dictate the performance of downwind turbines, of which there are many more than there are leading row turbines usually). My specific suggestion would be to confront the scope of the paper in the introduction and conclusions/recommendations to highlight this focus on wind turbines rather than arrays.
- 2. There is some inconsistency in the degree to which topics are introduced in a simple way through text description versus quantitative measures. For example, first order statistics, shear, and TKE are described completely, whereas Reynolds decomposition, integral length scale, energy spectra, etc. are not as clearly introduced quantitatively. Please consider making the technical descriptions more uniform.
- 3. There seems to be no substantial discussion of the boundary layer height, which plays a critical role for wind farms, aside from a limited discussion in conjunction with gravity waves in Section 7.3.3. The boundary layer height may also play an increasing role for individual turbines (focus of this review) as well, given the growing size of turbines mentioned many times, and the potential operation in shallow marine/stable boundary layers.

**Point comments:**

- Line 14: In paragraph 1, the framing describes that wind resource is typically assessed using 10 min averaged hub height wind speed. Then commentary is made regarding turbulence timescales. I also thought this would be a good place to mention ABL shear (mentioned in abstract).
- Line 22: "Turbulence affects the efficiency of wind turbine power generation resulting in fluctuating power output."
   This sentence is true, but might be misleading as written. Although not defined explicitly yet, we typically understand turbine 'efficiency' as the coefficient of power of the turbine. The primary way turbulence affects fluctuating power output is by changing the magnitude of the wind speed. The coefficient of power (efficiency) can also be affected by turbulence (e.g. [1, 2]) but this will usually be a much smaller impact than the effect of fluctuating wind speeds.
- 3. Line 23: "It also shortens their lifespan by inducing dynamic loads" References are needed for such a sweeping (and impactful) statement.
- Line 59: "Rossby waves are a consequence of Earth's rotation (Rossby and Collaborators, 1985; Platzman, 1968), are embedded within global circulations." Typographical error
- 5. Line 100: "The diurnal cycle is more pronounced over land than over water." While generally true, diurnal cycles can be significant in coastal environments.
- 6. Line 122: consider defining barotropic/baroclinic
- 7. Line 141: The authors could consider first introducing the flux Richardson number, which has a justified derivation from the TKE budget and is therefore a robust measure of stability, before introducing the gradient Richardson number which is its approximate form that is more practically useful. Then, more quantitative statements could be made than this: "It is generally accepted that the boundary layer flow is quasi-laminar when Rig exceeds unity."
- 8. Figure 5: The roughness sublayer, surface layer, and outer layers appear to not be defined in text or in the caption.
- 9. Line 230: "There is a lack of measurement of the different ILS components." Unclear what this sentence means, consider rephrasing
- Line 290: "In addition to the characterization of ABL flows motivated by wind energy, we add here a more mathematical discussion to explain the statistical content of the characterization." I did not follow what is meant exactly by this sentence (and therefore the motivation of the section). Consider rephrasing. More generally, this subsection contains important content but is

written at a more advanced level than the earlier parts of the paper.

- 11. Line 335: Typographical error
- 12. Section 4.1: The authors may add discussion regarding the quantitative identification of LLJs [e.g. 3]
- 13. Section 4.4 and Section 5 have duplicated content on flow over complex terrain
- 14. Line 465: I am unclear what the authors mean when they say that computing turbulent fluxes requires Taylor's hypothesis
- 15. Line 567: "Ideally, assuming a steady laminar flow, the power produced in this region is given theoretically by:" -> "Ideally, assuming a steady laminar **uniform** flow, the power produced in this region is given theoretically by:"
- 16. Line 592: Of relevance: recent evidence suggests the rotor equivalent wind speed model does not fully capture the effects of the wind profile shape (i.e. wind shear and veer), [4. 5]
- 17. Section 7.1: Given the focus of this review paper on turbulence, and the discussion of wind tunnel tests, the authors should consider confronting the issue of dynamic similarity, especially Reynolds number, and how that affects the interpretation of wind tunnel tests [6, 7]
- 18. Section 7.3.1: Motivation is given via Great Plains, but LLJs can also be quite important in coastal regions for offshore wind [8]
- 19. Line 725: This discussion of the scale of wind turbines and farms could be relocated to the introduction
- 20. Line 765: Reference formatting
- 21. Line 923: Typographical error
- 22. Section 8.1 is comprehensive and well-written. Related to the discussion in paragraph starting at Line 935: There has been recent discussion as to

Related to the discussion in paragraph starting at Line 935. There has been recent discussion as to whether failure rates may be increasing, and high profile failure events in the past several years are gaining more attention. The authors may consider a brief summary of knowledge gaps that could be related to these failures. Growing turbine size and veer are mentioned already. Aeroelasticity and the coupling between SIV/VIV and anisotropic/intermittent ABL turbulence are pertinent.

23. Line 974: Reference formatting

- 24. Line 981: More recent and relevant publication [9]
- 25. Line 981: "Furthermore, they state that turbulence length scales are smaller in complex terrain, which is why turbine fatigue loads tend to be higher in complex terrain." I imagine it would be challenging to make such a general statement, especially in light of the discussion earlier on AGW and the lack of a unified standard on what is "complex terrain"
- 26. Line 992: "Englberger et al. (2020) used LES to study what controls downwind wake deflection and found the blade rotation when combined with directional shear (wind veer or backing) result in a significant wake deflection."

This is a good reference but seems out of place in a section about fatigue loads. There are many more studies on how ABL phenomena and turbine operation (shear, stability, Coriolis effects, yaw, ...) affect wakes in general, beyond blade rotation+shear. More generally, this review does not describe wakes/farm scale processes, so this reference is somewhat isolated.

- 27. Figure 15: Unclear what Steps 1, 2, and 3 are
- 28. Line 1030: This paragraph has high overlap with the previous section on ABL turbulence models
- 29. Section 8.3: Relevant to the discussion on 1059, recent LES indicates that wake added turbulence depends on ABL stability [10] which is not well addressed by existing empirical models
- 30. Line 1089: "however, the disadvantage is that the high Reynolds number characteristic for atmospheric flows is not possible to achieve" This statement is not strictly correct, as demonstrated in Refs. [6, 7]. Also, shear and stability is possible to achieve [e.g. 11, 12], but veer, LLJs, and AGWs are certainly more challenging.
- 31. Line 1123: "Turbine incidents and failures are underreported due to legal and other proprietary considerations." Strong statement that may be true, but would require references/proof to include in this paper
- 32. Line 1151: The sentence is incomplete/cutoff

**References**

[1] Elliott, Dennis L., and Jack B. Cadogan. Effects of wind shear and turbulence on wind turbine power curves. No. PNL-SA-18354; CONF-900989-2. Pacific Northwest Lab., Richland, WA (USA), 1990.
 [2] Clifton, Andy, and Rozenn Wagner. "Accounting for the effect of turbulence on wind turbine power curves." In Journal of Physics: Conference Series, vol. 524, no. 1, p. 012109. IOP Publishing, 2014.
 [3] Debnath, Mithu, Patrick Moriarty, Raghavendra Krishnamurthy, Nicola Bodini, Rob Newsom, Eliot Quon, Julie K. Lundquist, Stefano Letizia, Giacomo Valerio Iungo, and Petra Klein. "Characterization of wind speed and directional shear at the AWAKEN field campaign site." Journal of Renewable and Sustainable Energy 15, no. 3 (2023).

[4] Mata, Storm A., Juan José Pena Martínez, Jesús Bas Quesada, Felipe Palou Larrañaga, Neeraj Yadav, Jasvipul S. Chawla, Varun Sivaram, and Michael F. Howland. "Modeling the effect of wind speed and direction shear on utility-scale wind turbine power production." Wind Energy 27, no. 9 (2024): 873-899.
[5] Vratsinis, Konstantinos, Rebeca Marini, Pieter-Jan Daems, Lukas Pauscher, Jeroen van Beeck, and Jan Helsen. "Impact of inflow conditions and turbine placement on the performance of offshore wind turbines exceeding 7 MW." Wind Energy Science Discussions 2025 (2025): 1-18.

[6] Miller, Mark A., Janik Kiefer, Carsten Westergaard, Martin OL Hansen, and Marcus Hultmark. "Horizontal axis wind turbine testing at high Reynolds numbers." Physical Review Fluids 4, no. 11 (2019): 110504.

[7] Miller, Mark A., Subrahmanyam Duvvuri, Ian Brownstein, Marcus Lee, John O. Dabiri, and Marcus Hultmark. "Vertical-axis wind turbine experiments at full dynamic similarity." Journal of Fluid Mechanics 844 (2018): 707-720.

[8] De Jong, Emily, Eliot Quon, and Shashank Yellapantula. "Mechanisms of low-level jet formation in the us mid-atlantic offshore." Journal of the Atmospheric Sciences 81, no. 1 (2024): 31-52.

[9] Damiani, Rick, Scott Dana, Jennifer Annoni, Paul Fleming, Jason Roadman, Jeroen van Dam, and Katherine Dykes. "Assessment of wind turbine component loads under yaw-offset conditions." Wind Energy Science 3, no. 1 (2018): 173-189.

[10] Klemmer, Kerry S., and Michael F. Howland. "Momentum deficit and wake-added turbulence kinetic energy budgets in the stratified atmospheric boundary layer." Physical Review Fluids 9, no. 11 (2024): 114607.

[11] Chamorro, Leonardo P., and Fernando Porté-Agel. "Effects of thermal stability and incoming boundary-layer flow characteristics on wind-turbine wakes: a wind-tunnel study." Boundary-layer meteorology 136 (2010): 515-533.

[12] Bartl, Jan, Franz Mühle, Jannik Schottler, Lars Sætran, Joachim Peinke, Muyiwa Adaramola, and Michael Hölling. "Wind tunnel experiments on wind turbine wakes in yaw: effects of inflow turbulence and shear." Wind Energy Science 3, no. 1 (2018): 329-343.

---

## Referee Comment (RC2)

**Review of the manuscript wes-2025-42, "Impact of atmospheric turbulence on performance and loads of wind turbines: Knowledge gaps and research challenges", by B. Kosovic, S. Basu, J. Berg, L.K. Berg, S.E. Haupt, X.G. Larsen, J. Peinke, R.J.A.M. Stevens, P. Veers, S. Watson.**

This manuscript provides an extensive review of the role of the atmosphere in power capture and loads of wind turbines. This manuscript includes essential information for a general reader to be informed about the main phenomena associated with this wind energy topic, and it can be a valuable resource for our research community as well. These are my main comments on this manuscript:

1. Each section tackles, in more or less depth, a specific related topic by providing a summary of the associated recent literature. Finally, a very brief outlook on the related research is provided in the final section 8.6. I would propose to restructure each section including a summary of the recent research achievements (rather than listing the executed works), then illustrating the current research gaps, and the research projects/tasks needed to address those gaps. This writing approach is sometimes outlined in a few sections, but generally not implemented in most of the sections.

2. Some effort should be made to homogenize this extensive manuscript. Currently, it reads as a collection of various drafts written by different authors with different writing styles connected by their content. As I understand this was a necessary strategy to work on such an extensive manuscript, at the same time, I believe an extra effort should be made to homogenize the writing, avoid potential overlaps, and cross-reference different sections when possible.

3. The manuscript is very extensive and, sometimes, some discussions are rather shallow and could be omitted (see details below). I would suggest revising critically the manuscript to identify those sections/parts that can be removed, merged, or shortened without omitting important information for the reader.

Some detailed comments are reported in the following.

**Comments:**

1. L57 The turbulent motions….To my knowledge (e.g. PERRY, A.E. & MARUSIC, I. 1995 A wall-wake model for the turbulence structure of boundary layers. Part 1. Extension of the attached eddy hypothesis. J. Fluid Mech. 298, 361–388.; HÖGSTRÖM,U., HUNT, J.C.R. & SMEDMAN, A.S. 2002 Theory and measurements for turbulence spectra and variances in the atmospheric neutral surface layer. Boundary-Layer Meteorol. 103 (1), 101–124; Van der Hoven (1957) for the spectral gap) turbulent motions have a specific spectral footprint and are restricted to scales smaller than those belonging to the mesoscale range. I think you should replace the adjective turbulent with atmospheric.

2. L79 – You can merge it with the previous paragraph.

3. L108 – Check for typos.

4. L192-194 – Specify the criterion used in Kelley et al. (2006) to identify neutral conditions.

5. Sect. 3.9 is very disconnected from the rest of the discussion. Maybe it can be removed.

6. L464 – Please add that sonic anemometers typically measure virtual temperature as well. This physical parameter is leveraged for the estimation of the friction velocity and Obukhov length through the eddy-covariance method.

7. L559 – Add here the reference to the IEC standards, which is provided at L 585, instead.
8. L655 – Provide details on the Langevin equation.
9. L656 – Fix references.
10. L721 – Wind Energy, no need for capital letters.
11. Sect. 8.2.2 does not provide a clear explanation of the phenomenon described. I would suggest removing it.
12. Similarly for Sect. 8.3. The discussion is very generic and no critical information is provided.

---

## Community Comment (CC1)

[supplement omitted: unrelated document]

---

## Community Comment (CC2)

[revised manuscript text omitted]

---

## Author Comment (AC1)

**Reviewer #1 Response Letter for WES-2025-42**
**Impact of atmospheric turbulence on performance and loads of wind turbines: Knowledge gaps and research challenges**

Branko Kosović, Sukanta Basu, Jacob Berg, Larry K. Berg, Sue E. Haupt,
Xiaoli G. Larsen, Joachim Peinke, Richard J. A. M. Stevens,
Paul Veers, and Simon Watson

**Response to Reviewer #1**

We thank the reviewer for their time and evaluation of our paper. We have carefully read these comments (shown in blue font) and provided point-by-point responses (shown in magenta font) and the modified text (shown in red font) below. For context, in some instances we included text that was not modified (shown in black font).

The submitted paper reviews the fundamentals of atmospheric flow from global to microscale, and then zooms in on mesoscale and microscale turbulence and their impact on wind turbines. As turbines continue to grow in size and deployment, this review is timely and important to highlight the need to revisit commonly used assumptions/simplifications. Overall, the paper is well-written and is an especially impressive effort coordinating many authors. There are several areas that can be improved with further integration across the paper, I've noted a few below, where the text repeats or contradicts itself. I hope the authors may consider these comments in a revision.

We thank the reviewer for their positive remarks.

The authors have made a choice to focus the review on ABL effects on wind turbines, rather than wind farms. This is a reasonable choice to keep the paper's scope constrained, but wind turbines are nearly always placed in wind farms, where wake and array level effects will both depend on the ABL in interesting/complex ways and also will change many aspects discussed (i.e. in large wind farms, the effect of ABL turbulence on the loads of the leading row described extensively in Sections 7 and 8 could matter less than the interaction between the ABL flow and wakes/farm-scale effects that will dictate the performance of downwind turbines, of which there are many more than there are leading row turbines usually). My specific suggestion would be to confront the scope of the paper in the introduction and conclusions/recommendations to highlight this focus on wind turbines rather than arrays.

The following text (shown in red font) was added to the introduction to more clearly define the scope of the manuscript:

> When considering turbulence impact on wind energy we adopt a broad view of the atmospheric turbulence that is not focused only on irregular, chaotic, three-dimensional, small-scale motions in an ABL, but also includes larger-scale atmospheric forcings associated with quasi-geostrophic turbulence (e.g., Charney, 1971) and mesoscale phenomena (e.g., Lilly, 1983) that modulate turbulent flows in the ABL.

and

Our review focuses on the impact of turbulence on a single wind turbine rather than wind turbine arrays. The impacts of turbulence generated by wind turbine and wind farm wakes as well as turbine and farm control are addressed in companion papers in the "Grand Challenges: wind energy research needs for a global energy transition" series.

There is some inconsistency in the degree to which topics are introduced in a simple way through text description versus quantitative measures. For example, first order statistics, shear, and TKE are described completely, whereas Reynolds decomposition, integral length scale, energy spectra, etc. are not as clearly introduced quantitatively. Please consider making the technical descriptions more uniform.

We merged the subsections 3.1 and 3.2 and addressed the comment by adding information about Reynolds decomposition and turbulent fluxes. The subsection 3.1 now reads:

**3.1 Mean and Turbulence Quantities of ABL flows**

Instantaneous velocity components along the longitudinal, lateral, and vertical directions are commonly denoted by, $u$, $v$, and $w$, respectively. In addition to velocity components, relevant thermodynamic variables: pressure, $p$, temperature, $T$, and water vapor mixing ratio, $q$ determine atmospheric stability. Using these thermodynamic variables one can derive: the virtual temperature, the potential temperature, and the virtual potential temperature. The virtual temperature, $T_v$, is a temperature at which pressure and density of a dry air parcel are equal to the moist air one. The potential temperature, $\theta$, is a temperature a parcel of fluid would attain when adiabatically brought to a reference pressure (e.g., standard surface pressure, usually $p_0 = 1{,}000$ hPa), while the virtual potential temperature also accounts for the effects of water vapor.

$$T_v \approx T(1 + 0.608q); \qquad \theta = T\left(\frac{p_0}{p}\right)^{\frac{R}{c_p}}; \qquad \theta_v \approx \theta(1 + 0.608q)$$

Here, $R$ is the ideal gas constant and $c_p$ is specific heat capacity at the constant pressure.

Ensemble mean values of a variable $\varphi$ are represented by an overline, $\overline{\varphi}$. Since it is difficult to estimate an ensemble mean, in the wind energy community it is common practice to approximate it by a temporal, spatial, or combined spatio-temporal average. When conditions are nearly spatially homogeneous or

temporally stationary Reynolds decomposition of instantaneous values, $\varphi$ into mean and fluctuating quantities is applicable.

$$\varphi = \overline{\varphi} + \varphi'$$

 The mean velocity components are $\overline{u}$, $\overline{v}$, and $\overline{w}$ and the mean potential temperature is $\overline{\theta}$.[1]  The mean horizontal wind speed is: $U = \sqrt{\overline{u}^2 + \overline{v}^2}$. Henceforth, the mean wind speed at hub height will be denoted as $U_H$.

Reynolds decomposition is commonly used to define turbulence quantities including turbulent kinetic energy and turbulent fluxes of momentum and heat. The three components of velocity variances are denoted as $\sigma_u^2$, $\sigma_v^2$, and $\sigma_w^2$, respectively. Turbulence kinetic energy (TKE; $\overline{e}$) is computed as:

$$\overline{e} = \frac{1}{2}\left(\sigma_u^2 + \sigma_v^2 + \sigma_w^2\right)$$

The turbulence intensity is more commonly used in the engineering community.  It is frequently defined along the streamwise direction: $I = \sigma_u/U$.  The covariances $\overline{u'w'}$, $\overline{v'w'}$, and $\overline{u'v'}$ represent the components of momentum fluxes (closely related to Reynolds stress components); the sensible heat fluxes are denoted by $\overline{u'\theta'}$, $\overline{v'\theta'}$, and $\overline{w'\theta'}$.

Frequently flow conditions are not stationary or homogeneous. Multiresolution decomposition was developed for such conditions (e.g., Treviño and Andreas, 1996; Howell and Mahrt, 1997). Turbulence characterization under non-stationary and non-homogeneous conditions is challenging and requires careful consideration, it is currently a very active area of research (e.g., Lehner and Rotach, 2023; Arias-Arana et al., 2024).

We have also added definitions of the correlation function and the integral length scale (ILS) tensor in the subsection 3.5:

The ILS, which is usually estimated using correlation functions. A spatial autocorrelation function for a turbulent quantity $\varphi$ is defined as:

$$\rho_{\varphi\varphi}(\mathbf{x}, r\mathbf{e}) = \frac{\overline{\varphi'(\mathbf{x})\varphi'(\mathbf{x}+r\mathbf{e})}}{\overline{\phi'(\mathbf{x})^2}}$$

Here, $\rho$ is a spatial correlation function of a variable $\varphi$ at point in space $\mathbf{x}$, r is a distance from $\mathbf{x}$ along the direction of the unit vector $\mathbf{e}_\alpha$, and the overline denotes the ensemble average. In case of spatially homogeneous flows
* * *
[1] In the atmospheric science literature, a different convention is followed. There, $\overline{u}$ and $\overline{v}$ represent zonal and meridional velocity components, respectively.

the ensemble average can be replaced with the spatial average, while in case of statistically stationary flows it can be replaced with the time average. The ILS of a variable $\varphi$ along the direction $\alpha$ is then defined as:

$$L_\varphi^\alpha = \int_0^\infty \rho_{\varphi\varphi}(\mathbf{x}, r\mathbf{e}_\alpha)dr$$

There seems to be no substantial discussion of the boundary layer height, which plays a critical role for wind farms, aside from a limited discussion in conjunction with gravity waves in Section 7.3.3. The boundary layer height may also play an increasing role for individual turbines (focus of this review) as well, given the growing size of turbines mentioned many times, and the potential operation in shallow marine/stable boundary layers.

The following text addressing the comment about the boundary layer height was introduced in the subsection 3.1 (originally subsections 3.1 and 3.2):

Another parameter that characterizes an ABL is its height. While there is no single definition of the boundary layer height, it is commonly defined as a level above the surface where either TKE drops below some threshold or, alternatively, a level where the potential temperature gradient exceeds a certain value and forms a capping inversion. In the literature, the heights of the low-level jets are also used as surrogates of stable boundary layer heights. The ABL height is commonly denoted with $h$, while the convective, mixed layer height is frequently denoted with $z_i$. Under strong convective conditions a well mixed boundary layer height can exceed 3 km, while under stably stratified conditions with wind speeds greater than 3 m s$^{-1}$ (when wind turbines produce power) a boundary layer height can be as low as several tens of meters, i.e., below the hub height of a modern utility scale with turbine. To characterize wind turbine operating environment under stably stratified conditions the boundary layer height must be taken into consideration. While the boundary layer height can be inferred from remote sensing observations, direct observations are generally not available. This represents a challenge when estimating turbulence impacts on wind turbine performance since the wind shear and the turbulence level impacting turbine blades can vary significantly through a rotor rotation depending on the ABL height. Puccioni et al. (2024) used observations with a scanning lidar from the AWAKEN field campaign to assess ABL height. In a simulation study, Park et al. (2014) documented the influence of low-level jet heights on wind shear and turbulence intensity, and in turn, how these variables affect various turbine loads.

1. Line 14: In paragraph 1, the framing describes that wind resource is typically assessed using 10 min averaged hub height wind speed. Then commentary is made regarding turbulence timescales. I also thought this would be a good

place to mention ABL shear (mentioned in abstract).

Following reviewer's suggestion, in addition to wind speed (and direction), wind energy resource assessment also includes wind shear, but also turbulence intensity and their variability, the sentence now reads as:

> The wind energy resource at a location is commonly assessed by estimating hub-height wind speed and direction, wind shear, turbulence intensity, and their variability (Murthy and Rahi, 2017) considering wind speed averaged over ten-minute intervals (e.g., Global Wind Atlas, Davis et al., 2023).

2. Line 22: "Turbulence affects the efficiency of wind turbine power generation resulting in fluctuating power output." This sentence is true, but might be misleading as written. Although not defined explicitly yet, we typically understand turbine 'efficiency' as the coefficient of power of the turbine. The primary way turbulence affects fluctuating power output is by changing the magnitude of the wind speed. The coefficient of power (efficiency) can also be affected by turbulence (e.g. [1, 2]) but this will usually be a much smaller impact than the effect of fluctuating wind speeds.

Text is rearranged so that the impact of turbulence on loads is discussed first followed by the discussion of the wind turbine power generation efficiency.

> In addition to the primary effect of wind speed variability, turbulence also impacts the efficiency of wind turbine power generation resulting in fluctuating power output (e.g., Elliot and Cadogan, 1990; Clifton and Wagner, 2014).

3. Line 23: "It also shortens their lifespan by inducing dynamic loads" References are needed for such a sweeping (and impactful) statement.

References are added:

> Turbulence negatively affects wind turbine lifespan by inducing dynamic loads (Leishman, 2002; Veers et al., 2023).

4. Line 59: "Rossby waves are a consequence of Earth's rotation (Rossby and Collaborators, 1985; Platzman, 1968), are embedded within global circulations." Typographical error

Typographical error was corrected, a superfluous "are" was removed.

5. Line 100: "The diurnal cycle is more pronounced over land than over water." While generally true, diurnal cycles can be significant in coastal environments.

The following sentence was added:

> While the diurnal cycle is more pronounced over land than over water in coastal environments it drives sea and land breezes.

6. Line 122: consider defining barotropic/baroclinic

The following sentences were added:

> In barotropic flows density, pressure, and temperature isosurfaces coincide. In contrast, in barotropic flows isosurfaces of density and pressure do not coincide.

7. Line 141: The authors could consider first introducing the flux Richardson number, which has a justified derivation from the TKE budget and is therefore a robust measure of stability, before introducing the gradient Richardson number which is its approximate form that is more practically useful. Then, more quantitative statements could be made than this: "It is generally accepted that the boundary layer flow is quasi-laminar when Rig exceeds unity."

We introduced the flux Richardson number and added the following text:

> The ABL stability results from interplay of turbulence production and suppression. As mentioned, while shear results in production of turbulence, buoyancy can be either a source or a sink of turbulence. A non-dimensional parameter that characterizes atmospheric stability is the flux Richardson number defined as a ratio of buoyancy to shear production of turbulence:
>
> $$Ri_f = \frac{\frac{g}{T_0}\overline{w'\theta'}}{\overline{u'w'}\frac{\partial \overline{u}}{\partial z} + \overline{v'w'}\frac{\partial \overline{v}}{\partial z}}$$
>
> Estimating atmospheric stability using the flux Richardson number requires flux measurements which are frequently not available. Generally, measurements of wind and temperature profiles are more readily available. An alternative non-dimensional stability parameter can be more practically estimated as a ratio of  static stability ($N$) and wind shear ($S$).

8. Figure 5: The roughness sublayer, surface layer, and outer layers appear to not be defined in text or in the caption.

We have added the following sentence to the figure caption:

A roughness sublayer is affected by surface roughness elements, surface layer is a layer throughout which turbulent fluxes of momentum, heat, moisture and other constituents are approximately constant, while the outer layer represents the rest of an ABL extending to the entrainment zone through which an ABL interacts with the upper troposphere.

9. Line 230: "There is a lack of measurement of the different ILS components." Unclear what this sentence means, consider rephrasing

The subsection 3.5 was modified and now includes definition of integral length scales.

**3.5 Integral Length Scales**

The spatial dimensions of the most energetic eddies are commonly quantified by integral length scales (ILSs) $, L_k^i$. The ILSs are commonly estimated using correlation functions. A spatial autocorrelation function for a turbulent quantity $\varphi$ is defined as:

$$\rho_{\varphi\varphi}(\mathbf{x}, r\mathbf{e}_\alpha) = \frac{\overline{\varphi'(\mathbf{x})\varphi'(\mathbf{x}+r\mathbf{e}_\alpha)}}{\overline{\phi'(\mathbf{x})^2}}$$

Here, $\rho$ is a spatial correlation function of a variable $\varphi$ at point in space $\mathbf{x}$, r is a distance from $\mathbf{x}$ along the direction of the unit vector $\mathbf{e}_\alpha$, and the overline denotes the ensemble average. In case of spatially homogeneous flows the ensemble average can be replaced with the spatial average, while in case of statistically stationary flows it can be replaced with the time average. The ILS of a variable $\varphi$ along the direction $\mathbf{e}$ is then defined as:

$$L_\varphi^\alpha = \int_0^\infty \rho_{\varphi\varphi}(\mathbf{x}, r\mathbf{e}_\alpha)dr$$

If we instead of the separation vector $r\mathbf{e}$ we introduce a time offset $\vartheta$ then we can compute the temporal autocorrelation and the corresponding integral time scale.

The unclear statement was clarified as follows:

> There is currently a lack of direct measurements of all autocorrelations required to estimate the individual ILS components which limits our understanding of the ABL structure.

10. Line 290: "In addition to the characterization of ABL flows motivated by wind energy, we add here a more mathematical discussion to explain the statistical content of the characterization." I did not follow what is meant exactly by this sentence (and therefore the motivation of the section). Consider rephrasing. More generally, this subsection contains important content but is written at a more advanced level than the earlier parts of the paper.

The subsections 3.7 and 3.8 were merged into the new subsection 3.6 "Statistical Hierarchy, Spectra, and  Coherence" and the introduction of statistical tools was rewritten. The first two paragraphs now read as follows:

**3.6 Statistical Hierarchy, Spectra, and Coherence**

The chaotic nature of turbulent flows makes flow variables suitable for analysis using statistical tools. The properties of fluctuations of an observed turbulent flow scalar quantity $q$ can be represented by the probability density function (pdf) $p(q)$. If we consider a data set $\{q_i; i = 1, ..., N\}$, a pdf $p(q)$ is insensitive to any order in the sequence $i$ (here $i$ can denote a time or space index). Only if two values $q_i$ and $q_j$, with $j = i + \Delta$, are statistically independent the characterization by the pdf $p(q)$ is complete. While a pdf of a flow quantity in a non-fluctuating laminar flow is represented by a delta function $p(q) = \delta(q - \overline{q})$, the pdf of a turbulent flow quantity determines all its statistical moments $\overline{q^n} = \int q^n \ p(q) \ dq$. Statistical analysis of a turbulent flow quantity is commonly focused on central moments computed with respect to the mean value $\overline{q}$, $\overline{\mu^n} = \int \mu^n \ p(\mu) \ d\mu$, where $\mu = q - \overline{q}$. The second moment or variance is $\sigma_\mu^2 = \overline{\mu^2}$. We can define a transformed variable $\tilde{q} = \frac{\mu}{\sigma_\mu}$ and then the associated pdf is $p(\tilde{q})$. In Fig. 7 (a) an example of $p(\tilde{q})$, where $\tilde{q} = u'$, is shown in which the pdf has pronounced heavy tails. The corresponding Gaussian pdf is completely defined by $\overline{u} = 0$ and $\sigma_\mu$ and displayed as a solid curve. Note the large difference of the probability of large events, a $\mu = 5\sigma$ event is more than 100 times more frequent in the empirical pdf than in a Gaussian distribution. Similarly, the pdf of normalized time increments, $\frac{\delta u}{\sigma_{\delta u}} = \frac{u(t+\Delta t) - u(t)}{\sigma_{\delta u}}$ of velocity

has heavy tails Fig. 7 (b). Quantities characterized by such heavy-tailed pdfs are also called intermittent.

Since turbulent structures lead to dependencies of $q_i$ and $q_j$, such dependencies or correlations are of interest. Statistically, this is captured by the joint-pdf $p(q_i, q_j) = p(q_i, q_{i+\Delta})$, which for homogeneous data depends only on the separation $\Delta$. The lowest order moment of this joint-pdf is autocovariance $\overline{q_i \, q_j}$. The Wiener–Khintchine theorem states that the power spectrum $S_{qq}(n)$ is the Fourier transformation of autocovariance, where wave number or frequency, $n$, is proportional to $1/\Delta$. While a power spectrum characterizes energy content at different spatial or temporal scales, it does not characterize small-scale intermittency of turbulence. The intermittency is characterized by higher-order moments of increments $\overline{(q_{i+\Delta} - q_i)^n}$ (Frisch, 1995, for more details see also Morales et al., 2012). This statistical intermittency must be distinguished from the global intermittency induced by large coherent structures such are, for example, Kelvin-Helmholtz billows. The joint-pdf can be defined for two or more variables, the so-called multivariate statistics. The moments of different order of multivariate joint-pdfs are covariances. A covariance of two variables $q_i$, and $r_j$ is denoted by $\overline{q_i r_j}$ and the corresponding Fourier transform is a complex function, the cross-spectrum $S_{qr} = C_{qr}(n) + iQ_{qr}(n)$, where the real part, $Co_{qr}(n)$ is the cospectrum and the imaginary part $Qu_{rs}$ is the quadrature spectrum. The coherence function is defined as:

$$Coh_{qr}(n) = \frac{|S_{qr}(n)|^2}{S_{qq}(n)S_{rr}(n)}$$

11. Line 335: Typographical error

Corrected: Figure 10...

12. Section 4.1: The authors may add discussion regarding the quantitative identification of LLJs [e.g. 3]

The following text was added to the subsection 4.1:

Since Lettau and Davidson (1957) first identified LLJs during the Great Plains Project there were many attempts to unequivocally define an LLJ (e.g., Bonner, 1968; Whiteman et al., 1997; Banta et al., 2008), however this did not result in a generally accepted

definition. Such a definition was elusive due to relative sparsity and resolution of observations. We need more coordinated measurement campaigns with instruments for both mesoscale and microscale flows. AWAKEN (Moriarty et al., 2024) is a field study that provides such observations. In recent analysis of a long-term observations of LLJs at the Atmospheric Radiation Measurement Southern Great Plains site Debnath et al. (2023) used the following criteria to detect LLJs: location of the wind speed maximum where the difference between the wind speed maximum and the wind speed at the top of the jet is at least 2 m/s and this difference exceeds 10 % of the maximum wind speed. They found that LLJ wind profile cannot be represented by the shear exponent only.

Several attempts to analyze spectral features associated with LLJ structure and relate them to spectra observed in canonical stably-stratified ABLs without a jet (Kaimal, 1973) did not result in consistent findings. While Smedman et al. (2004) and Hallgren et al. (2022) found that low frequencies of the streamwise velocity spectra associated with LLJs are suppressed, however, these results are not consistent with some other studies (e.g., Duarte et al., 2012). The knowledge gained from the further analysis of these and other measurements that include characterization of mesoscale conditions is expected to shed light on how to couple the mesoscale and microscale modeling to fit LLJ's nature.

Also, the following paragraph was moved to the end of the subsection 7.3.1 where it better addresses the gaps in characterization of LLJs:

Although LLJs and their impacts on wind energy have been studied extensively in specific regions (e.g. Emeis, 2014; Aird et al., 2022), there is still a lack of observed wind speed and direction profiles associated with LLJs, particularly for offshore and coastal conditions (Shaw et al., 2022). Towers typically only reach 100-200 m, and frequently sodars are ineffective in the layer near the LLJ nose due to the lack of shear-produced turbulence. While profiling (floating) lidars can provide more information, they are expensive and not routinely used and typically have a vertical range of approximately 200 m. These measurements should also provide more information about the turbulence structures near and above the LLJ nose.

13. Section 4.4 and Section 5 have duplicated content on flow over complex terrain

The following redundant text was omitted:

Turbulence characteristics in complex terrain are impacted by the wide range of flow phenomena in addition to spatial inhomogeneity frequently resulting in non-equilibrium conditions. As indicated above,...

and the following text was moved from the subsection 4.5 to the end of the section 8.5:

There has been relatively little research into the interactions between downslope wind storms, power production, and turbine loads compared to studies of other extreme wind events. Observational studies, such as those by Sherry and Rival (2015), studied wind ramps associated with Chinook winds downwind in the Canadian Rockies. One of their findings was that turbulence intensity was generally large during ramp events. Kozmar and Grisogono (2020) provided a review of characteristics of downslope wind storms including mountain wave overturning and quasi periodic oscillations in wind speed and elevate turbulence levels result of Kelvin-Helmholtz instabilities that are particularly relevant for wind energy applications. They point out the need for updating engineering standards to account for large wind velocity fluctuations associated with downslope wind storms.

Please notice that the subsection 4.5 addresses downslope wind storm phenomena, while the section 5 addresses observations related to wind energy.

14. Line 465: I am unclear what the authors mean when they say that computing turbulent fluxes requires Taylor's hypothesis

The sentence was replaced with the following one:

By assuming that turbulence is frozen in time as it advects by a sensor, i.e. using Taylor's hypothesis (c.f., Wyngaard, 2010), one can use high-rate time series of measurements at a point in space and interpret them as a spatial record.

15. Line 567: "Ideally, assuming a steady laminar flow, the power produced in this region is given theoretically by:" – "Ideally, assuming a steady laminar uniform flow, the power produced in this region is given theoretically by:"

The text was modified as suggested by the reviewer:

Ideally, assuming a steady laminar uniform flow, the power produced in this region is given theoretically by:

16. Line 592: Of relevance: recent evidence suggests the rotor equivalent wind speed model does not fully capture the effects of the wind profile shape (i.e. wind shear and veer), [4. 5]

The following sentences and references suggested by the reviewer were added:

In addition to wind shear, wind veer also impacts power production (Mata et al., 2024). Vratsinis et al. (2025) analyzed performance of a large offshore wind farm and concluded that the IEC-defined rotor equivalent wind speed does not represent the full effects of shear and veer on a large offshore wind turbine.

17. Section 7.1: Given the focus of this review paper on turbulence, and the discussion of wind tunnel tests, the authors should consider confronting the issue of dynamic similarity, especially Reynolds number, and how that affects the interpretation of wind tunnel tests [6, 7]

The following text was added to the end of the subsection 7.1:

While these wind tunnel studies were characterized by Reynolds numbers several orders of magnitude lower than those characteristic for conditions under which utility-scale turbines operate, Miller et al. (2018, 2019) used a high-pressure wind tunnel to achieve dynamic similarity to study both vertical and horizontal axis wind turbine power performance. Conducting experiments at a range of rotor diameter based Reynolds numbers they have shown that the scale effects can significantly impact horizontal axis turbine performance. However, they observed Reynolds number invariance of the power coefficient as a function of tip speed ratio for a cord-length based Reynolds number greater than $3.5 \times 10^6$ for a horizontal axis wind turbine and $1.5 \times 10^6$ for a vertical axis wind turbine.

18. Section 7.3.1: Motivation is given via Great Plains, but LLJs can also be quite important in coastal regions for offshore wind [8]

In subsection 7.3.1 the impact of LLJs on wind power is discussed, therefore the de Jong et al. (2024) reference better fits in subsection 4.1 where coastal LLJs are discussed. We have therefore added de Jong et al. (2024) reference to subsection 4.1. In the subsection 4.1 we have also added Olsen et al. (2024) reference that addresses North Sea and Baltic LLJs.

19. Line 725: This discussion of the scale of wind turbines and farms could be

relocated to the introduction

TBD: The discussion was moved to the Introduction, where the following text was added:

> For example, offshore, where the size limits have not been reached yet, wind turbines are approaching 300 meters (e.g., the Haliade-X turbine, 260 m, or the Vestas V236-15.0 MW, 280 m), beyond the frequently shallow marine ABL. Furthermore, modern wind farm clusters are expanding to areas of thousands square kilometers (e.g., Hornsea area 7240 $km^2$, Minnesota wind farms about 5000 $km^2$) spanning a wide range of atmospheric scales.

20. Line 765: Reference formatting

Reference formatting was corrected.

21. Line 923: Typographical error

Typographical error was corrected.

22. Section 8.1 is comprehensive and well-written. Related to the discussion in paragraph starting at Line 935: There has been recent discussion as to whether failure rates may be increasing, and high profile failure events in the past several years are gaining more attention. The authors may consider a brief summary of knowledge gaps that could be related to these failures. Growing turbine size and veer are mentioned already. Aeroelasticity and the coupling between SIV/VIV and anisotropic/intermittent ABL turbulence are pertinent.

Following reviewer's suggestion the following paragraph was added to the subsection 8.1

> Flow conditions, characterized by anisotropic, intermittent turbulence result in dynamic instability leading to flow-induced vibrations and blade flutter causing unsteady aerodynamic forces. Dimitrov et al. (2017) quantified the impact of turbulence length scales and anisotropy associated with normal and extreme turbulence on fatigue and extreme loads. They concluded that compared to the observed standard deviation of turbulence in the IEC 61400-1 Ed.3 standard is underestimated. Perturbations due to turbulence can result in large vortex-induced vibrations (Grinderslev et al., 2022).

As the size of wind turbines increases the impact of vortex-induced vibrations (Grinderslev et al., 2023).Naqash and Alam (2025) provide a review of impact of flow-induced vibrations on turbine blade loads, fatigue, and failures.

23. Line 974: Reference formatting

Reference formatting was corrected.

24. Line 981: More recent and relevant publication [9]

The following text and the reference were added:

Studying yaw control for a wind turbine and a wind farm, Damiani et al. (2018) concluded that turbulence intensity is one of the primary causes of damage equivalent loads.

25. Line 981: "Furthermore, they state that turbulence length scales are smaller in complex terrain, which is why turbine fatigue loads tend to be higher in complex terrain." I imagine it would be challenging to make such a general statement, especially in light of the discussion earlier on AGW and the lack of a unified standard on what is "complex terrain"

The sentence was modified to more accurately represent the finding in Riziotis and Voutsinas (2000):

Furthermore, they state that the higher turbulence intensity is the main cause of higher turbine fatigue loads in complex terrain.

26. Line 992: "Englberger et al. (2020) used LES to study what controls downwind wake deflection and found the blade rotation when combined with directional shear (wind veer or backing) resulted in a significant wake deflection." This is a good reference but seems out of place in a section about fatigue loads. There are many more studies on how ABL phenomena and turbine operation (shear, stability, Coriolis effects, yaw, ...) affect wakes in general, beyond blade rotation+shear. More generally, this review does not describe wakes/farm scale processes, so this reference is somewhat isolated.

Following reviewers suggestion the sentence and accompanying reference were removed.

Englberger et al. (2020) used LES to study what controls downwind wake deflection and found the blade rotation when combined with directional shear (wind veer or backing) result in a significant wake deflection.

Since we do not focus on wake induced turbulence, we have also removed the following sentence referencing wake-wake interactions

Given that the impact of turbine loading is highly non-linear, additional fundamental insight into wake-wake interactions is required to improve predictions of the turbulence conditions inside extended wind farms or the interaction between wind farms.

27. Figure 15: Unclear what Steps 1, 2, and 3 are

The following sentence was added to the caption of Figure 15

Steps 1 through 3 represent modifications of the inflow by varying wind shear (STEP 1), veer (STEP 2), and turbulence level (STEP 3).

28. Line 1030: This paragraph has high overlap with the previous section on ABL turbulence models

The paragraph was moved to the section 6 "Modeling of ABL Flows" which now includes the following text:

TurbSim (Kelley and Jonkman, 2007) is one of the most commonly used  tools that can rapidly generate a range of idealized turbulent fields. To meet the new design needs, TurbSim must be extended to provide a wider range of realistic turbine inflows. Turbulent inflow can also be generated using LES; however, such simulations are computationally expensive. Therefore, there is still a need for a faster alternative. In addition to a tool like TurbSim, recent developments in Artificial Intelligence and Machine Learning (AI/ML), in particular state-of-the-art physics-informed deep learning approaches, provide an opportunity to develop ML models using vision transformers that could generate realistic turbulent fields at a fraction of a cost of an LES (e.g., Stengel et al., 2020; Dettling et al., 2025). Alternatively, coupled simulations could be used to create a public database of ABL flows, similar to the Johns Hopkins

Turbulence Database (Johns Hopkins University, 2021; Zhu et al., 2025).

As larger modern turbines are exposed to a wider range of atmospheric conditions, there is a need to better represent the turbulent inflow that impacts their performance. To resolve ABL turbulence, from the largest boundary layer eddies into the inertial range of turbulence characterized by Kolmogorov -5/3 spectrum (Kolmogorov, 1941), we can employ large-eddy simulations (LESs).

29. Section 8.3: Relevant to the discussion on 1059, recent LES indicates that wake added turbulence depends on ABL stability [10] which is not well addressed by existing empirical models

Considering the reviewer's recommendation that the manuscript focus on the impact of atmospheric turbulence on the first row of turbines, and given that other papers in the series address wake-generated turbulence and its impacts, we have decided to remove the subsection 8.3 from the manuscript.

30. Line 1089: "however, the disadvantage is that the high Reynolds number characteristic for atmospheric flows is not possible to achieve" This statement is not strictly correct, as demonstrated in Refs. [6, 7]. Also, shear and stability is possible to achieve [e.g. 11, 12], but veer, LLJs, and AGWs are certainly more challenging.

The statement was corrected as suggested and the reference added and it now reads as follows:

While there are studies that replicated some of the atmospheric conditions (e.g., Chamorro and Porté-Agel, 2010), the disadvantage is that unless a wind tunnel is pressurized (Miller et al., 2019), it is not possible to achieve the high Reynolds number characteristic for atmospheric flows. In addition, a range of atmospheric stabilities and Coriolis effects and their consequences (e.g., veer, LLJs, AGWs) are difficult or impossible to replicate.

31. Line 1123: "Turbine incidents and failures are underreported due to legal and other proprietary considerations." Strong statement that may be true, but would require references/proof to include in this paper

Instead of the quoted sentence the following text and a reference were added at the end of the subsection 8.5:

Turbine incidents and failures are underreported. There are only a few data mining studies of wind turbine failures and accidents based on textual analysis of news reports (e.g., Ertek and Kailas, 2021). There is a need for creation of a comprehensive database for better assessment of the impact of extreme events on individual wind turbines and wind farms and for more accurate risk assessment.

32. Line 1151: The sentence is incomplete/cutoff

The sentence was completed as follows:

While this enables a level of confidence in the statistical prediction of extreme wind for wind turbine design, considering the potential impact, the standard should be continuously evaluated based on available observations and high-resolution simulations.

**References**

Aird, J. A., Barthelmie, R. J., Shepherd, T. J., and Pryor, S. C. (2022). Occurrence of low-level jets over the eastern us coastal zone at heights relevant to wind energy. Energies, 15:445.

Arias-Arana, D., Fochesatto, G. J., Jimenez, R., and Ojeda, C. (2024). Locally stationary wavelet analysis of nonstationary turbulent fluxes. Boundary-Layer Meteorology, 190:33.

Banta, R. M., Pichugina, Y. L., Kelley, N. D., Jonkman, B., and Brewer, W. A. (2008). Doppler lidar measurements of the Great Plains low-level jet: Applications to wind energy. In IOP Conference Series: Earth and Environmental Science, volume 1, page 012020.

Bonner, W. D. (1968). Climatology of the low level jet. Mon. Weather Rev., 96:833–850.

Chamorro, L. P. and Porté-Agel, F. (2010). Effects of thermal stability and incoming boundary-layer flow characteristics on wind-turbine wakes: A wind-tunnel study. Boundary-Layer Meteorol., 136:515–533.

Charney, J. G. (1971). Geophysical turbulence. J. Atmos. Sci., 28:1087–1095.

Clifton, A. and Wagner, R. (2014). Accounting for the effect of turbulence on wind turbine power curves. J. Phys. Conf. Ser., 524.

Damiani, R., Dana, S., Annoni, J., Fleming, P., Roadman, J., van Dam, J., and Dykes, K. (2018). Assessment of wind turbine component loads under yaw-offset conditions. Wind Energy Science, 3(1):173–189.

Davis, N. N., Badger, J., Hahmann, A. N., Hansen, B. O., Mortensen, N. G., Kelly, M., Larsén, X. G., Olsen, B. T., Floors, R., Lizcano, G., Casso, P., Lacave, O., Bosch, A., Bauwens, I., Knight, O. J., van Loon, A. P., Fox, R., Parvanyan, T., Hansen, S. B. K., Heathfield, D., Onninen, M., and Drummond, R. (2023). The global wind atlas - a high-resolution dataset of climatologies and associated web-based application. Bull. Am. Meteorol. Soc., 104:E1507–E1525.

Debnath, M., Moriarty, P., Krishnamurthy, R., Bodini, N., Newsom, R., Quon, E., Lundquist, J. K., Letizia, S., Iungo, G. V., and Klein, P. (2023). Characterization of wind speed and directional shear at the awaken field campaign site. Journal of Renewable and Sustainable Energy, 15(3):033308.

Dettling, S., Haupt, S. E., Brummet, T., Kosovic, B., Gagne, D. J., and Jimenez, P. A. (2025). Downscaling from mesoscale to microscale in complex terrain using a compound generative adversarial network. Artificial Intelligence for the Earth Systems, 4:X–Y.

Dimitrov, N., Natarajan, A., and Mann, J. (2017). Effects of normal and extreme turbulence spectral parameters on wind turbine loads. Renewable Energy, 101:1180–1193.

Duarte, H. F., Leclerc, M. Y., and Zhang, G. (2012). Assessing the shear-sheltering theory applied to low-level jets in the nocturnal stable boundary layer. Theoretical and Applied Climatology, 110:359–371.

Elliot, D. L. and Cadogan, J. B. (1990). Effects of wind shear and turbulence on wind turbine power curves. Technical report, Pacific Northwest Lab. Richland, WA, No. PNL-SA-18354; CONF-900989-2.

Emeis, S. (2014). Wind speed and shear associated with low-level jets over northern germany. Meteorologische Zeitschrift, 23(3):295–304.

Englberger, A., Lundquist, J. K., and Dörnbrack, A. (2020). Changing the rotational direction of a wind turbine under veering inflow: a parameter study. Wind Energy Science, 5(4):1623–1644.

Ertek, G. and Kailas, L. (2021). Analyzing a decade of wind turbine accident news with topic modeling. Sustainability, 13(22).

Frisch, U. (1995). Turbulence: The Legacy of A. N. Kolmogorov. Cambridge University Press, Cambridge, UK.

Grinderslev, C., Houtin-Mongrolle, F., Nørmark Sørensen, N., Raimund Pirrung, G., Jacobs, P., Ahmed, A., and Duboc, B. (2023). Forced-motion simulations of vortex-induced vibrations of wind turbine blades – a study of sensitivities. Wind Energy Science, 8(10):1625–1638.

Grinderslev, C., Nørmark Sørensen, N., Raimund Pirrung, G., and González Horcas, S. (2022). Multiple limit cycle amplitudes in high-fidelity predictions of standstill wind turbine blade vibrations. Wind Energy Science, 7(6):2201–2213.

Hallgren, C., Arnqvist, J., Nilsson, E., Ivanell, S., Shapkalijevski, M., Thomasson, A., Pettersson, H., and Sahlée, E. (2022). Classification and properties of non-idealized coastal wind profiles – an observational study. Wind Energy Science, 7(3):1183–1207.

Howell, J. F. and Mahrt, L. (1997). Multiresolution flux decomposition. Boundary-Layer Meteorology, 83:117–137.

Johns Hopkins University (2021). Johns hopkins turbulence databases. Accessed: March 4, 2025.

Kaimal, J. C. (1973). Turbulence spectra, length scales and structure parameters in the stable surface layer. Bound.-Layer Meteorol., 4:289–309.

Kelley, N. D. and Jonkman, B. (2007). Overview of the turbsim stochastic inflow turbulence simulator. Technical Report NREL/TP-500-41137, National Renewable Energy Lab (NREL), Golden, CO (United States).

Kolmogorov, A. N. (1941). The local structure of turbulence in incompressible viscous fluid for very large reynolds numbers. Doklady Akademiia Nauk SSSR, 30:301–305.

Kozmar, H. and Grisogono, B. (2020). Characteristics of downslope windstorms in the view of the typical atmospheric boundary layer. In Hangan, H. and Kareem, A., editors, The Oxford Handbook of Non-Synoptic Wind Storms, chapter 5, pages 85–114. Oxford University Press, Oxford.

Lehner, M. and Rotach, M. W. (2023). The performance of a time-varying filter time under stable conditions over mountainous terrain. Boundary-Layer Meteorology, 188:523–551.

Leishman, J. G. (2002). Challenges in modelling the unsteady aerodynamics of wind turbines. Wind Energy, 5(2-3):85–132.

Lettau, H. and Davidson, B. (1957). Exploring the Atmosphere's First Mile Proceedings of the Great Plains Turbulence Field Program, 1. Pergamon Press, London, UK, 1 edition. pp.578.

Lilly, D. K. (1983). Stratified turbulence and the mesoscale variability of the atmosphere. Journal of Atmospheric Sciences, 40(3):749 – 761.

Mata, S. A., Pena Martínez, J. J., Bas Quesada, J., Palou Larrañaga, F., Yadav, N., Chawla, J. S., Sivaram, V., and Howland, M. F. (2024). Modeling the effect of wind speed and direction shear on utility-scale wind turbine power production. Wind Energy, 27(9):873–899.

Miller, M. A., Duvvuri, S., Brownstein, I., Lee, M., Dabiri, J. O., and Hultmark, M. (2018). Vertical-axis wind turbine experiments at full dynamic similarity. Journal of Fluid Mechanics, 844:707–720.

Miller, M. A., Kiefer, J., Westergaard, C., Hansen, M. O. L., and Hultmark, M. (2019). Horizontal axis wind turbine testing at high reynolds numbers. Phys. Rev. Fluids, 4:110504.

Morales, A., Wächter, M., and Peinke, J. (2012). Characterization of wind turbulence by higher-order statistics. Wind Energy, 15:391–406.

Moriarty, P., Bodini, N., Letizia, S., Abraham, A., Ashley, T., Bärfuss, K. B., Barthelmie, R. J., Brewer, A., Brugger, P., Feuerle, T., Frére, A., Goldberger, L., Gottschall, J., Hamilton, N., Herges, T., Hirth, B., Hung, L.-Y. L., Iungo, G. V., Ivanov, H., Kaul, C., Kern, S., Klein, P., Krishnamurthy, R., Lampert, A., Lundquist, J. K., Morris, V. R., Newsom, R., Pekour, M., Pichugina, Y., Porté-Angel, F., Pryor, S. C., Scholbrock, A., Schroeder, J., Shartzer, S., Smiley, E., Vöhringer, L., Wharton, S., and Zalkind, D. (2024). Overview of preparation for the american wake experiment (awaken). J. Renew. Sustain. Energy.

Murthy, K. and Rahi, O. (2017). A comprehensive review of wind resource assessment. Renewable and Sustainable Energy Reviews, 72:1320–1342.

Naqash, T. M. and Alam, M. M. (2025). A state-of-the-art review of wind turbine blades: Principles, flow-induced vibrations, failure, maintenance, and vibration suppression techniques. Energies, 18(13).

Park, J., Basu, S., and Manuel, L. (2014). Large-eddy simulation of stable boundary layer turbulence and estimation of associated wind turbine loads. Wind Energy, 17(3):359–384.

Puccioni, M., Moss, C. F., Solari, M. S., Roy, S., Iungo, G. V., Wharton, S., and Moriarty, P. (2024). Quantification and assessment of the atmospheric boundary layer height measured during the awaken experiment by a scanning lidar. Journal of Renewable and Sustainable Energy, 16(5):053304.

Riziotis, V. A. and Voutsinas, S. G. (2000). Fatigue loads on wind turbines of different control strategies operating in complex terrain. J. Wind Eng. Ind. Aerodyn., 85(3):211–240.

Shaw, W. J., Berg, L. K., Debnath, M., Deskos, G., Draxl, C., Ghate, V. P., Hasager, C. B., Kotamarthi, R., Mirocha, J. D., Muradyan, P., Pringle, W. J., Turner, D. D., and Wilczak, J. M. (2022). Scientific challenges to characterizing the wind resource in the marine atmospheric boundary layer. Wind Energy Sci., 7(6):2307–2334.

Sherry, M. and Rival, D. (2015). Meteorological phenomena associated with wind-power ramps downwind of mountainous terrain. J. Renew. Sustain. Energy, 7(3):033101.

Smedman, A.-S., Högström, U., and Hunt, J. C. R. (2004). Effects of shear sheltering in a stable atmospheric boundary layer with strong shear. Quarterly Journal of the Royal Meteorological Society, 130(596):31–50.

Stengel, K., Glaws, A., Hettinger, D., and King, R. N. (2020). Adversarial super-resolution of climatological windand solar data. PNAS, 375:16805–16815.

Treviño, G. and Andreas, E. (1996). On wavelet analysis of nonstationary turbulence. Boundary-Layer Meteorology, 81:27–288.

Veers, P., Bottasso, C. L., Manuel, L., Naughton, J., Pao, L., Paquette, J., Robertson, A., Robinson, M., Ananthan, S., Barlas, T., Bianchini, A., Bredmose, H., Horcas, S. G., Keller, J., Madsen, H. A., Manwell, J., Moriarty, P., Nolet, S., and Rinker, J. (2023). Grand challenges in the design, manufacture, and operation of future wind turbine systems. Wind Energy Science, 8(7):1071–1131.

Vratsinis, K., Marini, R., Daems, P.-J., Pauscher, L., van Beeck, J., and Helsen, J. (2025). Impact of inflow conditions and turbine placement on the performance of offshore wind turbines exceeding 7 mw. Wind Energy Science Discussions, 2025:1–18.

Whiteman, C. D., Bian, X., and Zhong, S. (1997). Low-level jet climatology from enhanced rawinsonde observations at a site in the southern Great Plains. J. Appl. Meteor. Climatol., 36:1363–1376.

Wyngaard, J. C. (2010). Turbulence in the atmosphere. Cambridge University Press.

Zhu, X., Xiao, S., Narasimhan, G., Martinez-Tossas, L. A., Schnaubelt, M., Lemson, G., Yao, H., Szalay, A. S., Gayme, D., and Meneveau, C. (2025). Jhtdb-wind: a web-accessible large-eddy simulation database of a wind farm with virtual sensor querying. Wind Energy Science Discussions, 2025:1–28.

---

## Author Comment (AC2)

**Reviewer #2 Response Letter for WES-2025-42**
**Impact of atmospheric turbulence on performance and loads of wind turbines: Knowledge gaps and research challenges**

Branko Kosović, Sukanta Basu, Jacob Berg, Larry K. Berg, Sue E. Haupt,
Xiaoli G. Larsen, Joachim Peinke, Richard J. A. M. Stevens,
Paul Veers, and Simon Watson

**Response to Reviewer #2**

We thank the reviewer for their time and evaluation of our paper. We have carefully read these comments (shown in blue font) and provided below point-by-point responses (shown in magenta font) and the modified text (shown in red font) below. For context, in some instances we included text that was not modified (shown in black font).

This manuscript provides an extensive review of the role of the atmosphere in power capture and loads of wind turbines. This manuscript includes essential information for a general reader to be informed about the main phenomena associated with this wind energy topic, and it can be a valuable resource for our research community as well.

We thank the reviewer for their positive remarks.

1. Each section tackles, in more or less depth, a specific related topic by providing a summary of the associated recent literature. Finally, a very brief outlook on the related research is provided in the final section 8.6. I would propose to restructure each section including a summary of the recent research achievements (rather than listing the executed works), then illustrating the current research gaps, and the research projects/tasks needed to address those gaps. This writing approach is sometimes outlined in a few sections, but generally not implemented in most of the sections.

In the revised manuscript we followed reviewer's suggestions. Since the manuscript can be viewed as consisting of two parts: the first part including relevant background, while the second part addresses the impact of atmospheric turbulence on wind power production and loads, to address reviewer's comment we have therefore focused on the second part. We have added paragraphs addressing research gaps to the subsection 7.3.1

> Although LLJs and their impacts on wind energy have been studied extensively in specific regions (e.g. Emeis, 2014; Aird et al., 2022), there is still a lack of observed wind speed and direction profiles associated with LLJs, particularly for offshore and coastal conditions Shaw et al. (2022). Towers typically only reach 100-200 m, and frequently sodars are ineffective in the layer near the LLJ nose due to the lack of shear-produced turbulence. While profiling (floating) lidars can provide more information, they are expensive and not routinely used and typically have a vertical range of approximately 200 m. These measurements should also provide more information about the turbulence structures near and above the LLJ nose.

and the subsection 8.5

Turbine incidents and failures are underreported. There are only a few data mining studies of wind turbine failures and accidents based on textual analysis of news reports (e.g., Ertek and Kailas, 2021). There is a need for creation of a comprehensive database of failures for better assessment of the impact of extreme events on individual wind turbines and wind farms and for more accurate risk assessment.

2. Some effort should be made to homogenize this extensive manuscript. Currently, it reads as a collection of various drafts written by different authors with different writing styles connected by their content. As I understand this was a necessary strategy to work on such an extensive manuscript, at the same time, I believe an extra effort should be made to homogenize the writing, avoid potential overlaps, and cross-reference different sections when possible.

We have made effort to homogenize the manuscript by eliminating subsection 8.2.2 and 8.3 in the original manuscript, merging subsection 3.1 and 3.2 into a new subsection 3.1 and subsection 3.7 and 3.9 into a new subsection 3.6, as well as partially rewriting or extending several sections (e.g., 4.1 "Low-level Jets," 5. "Observing ABL Flows," 6. "Modeling of ABL Flows," 7.1 "Power Curves," 8.2 "Impact of Atmospheric Phenomena on Fatigue Loads," 8.5 "Extreme Events and Loads," so that the overall structure of the manuscript is more uniform.

3. The manuscript is very extensive and, sometimes, some discussions are rather shallow and could be omitted (see details below). I would suggest revising critically the manuscript to identify those sections/parts that can be removed, merged, or shortened without omitting important information for the reader. Some detailed comments are reported in the following.

Following reviewer's comment we have eliminated subsections 8.2.2 "Global Intermittency and Coherent Structures" and 8.3 "Wind Farm Generated Turbulence." We have merged subsection 3.2 "Turbulence Quantities of ABL flows with subsection 3.1 into the new subsection 3.1 "Mean and Turbulence Quantities of ABL flows."

1. L57 The turbulent motions....To my knowledge (e.g. PERRY, A.E. & MARUSIC, I. 1995 A wall-wake model for the turbulence structure of boundary layers. Part 1. Extension of the attached eddy hypothesis. J. Fluid Mech. 298, 361–388.; HÖGSTRÖM,U., HUNT, J.C.R. & SMEDMAN, A.S. 2002 Theory and measurements for turbulence spectra and variances in the atmospheric neutral surface layer. Boundary-Layer Meteorol. 103 (1), 101–124; Van der Hoven (1957) for the spectral gap) turbulent motions have a specific spectral footprint and are restricted to scales smaller than those belonging to the mesoscale range. I think you should replace the adjective turbulent with atmospheric.

As suggested by the reviewer we have replaced "turbulent" with "atmospheric".

The sentence now reads as follows:

> The atmospheric motions exhibit three distinct kinetic energy scaling ranges, starting from the largest planetary waves and synoptic scales through mesoscale to microscales in the ABL depicted in Fig. 1.

2. L79 – You can merge it with the previous paragraph.

The paragraph was merged.

3. L108 – Check for typos.

Typo corrected.

4. L192-194 – Specify the criterion used in Kelley et al. (2006) to identify neutral conditions.

The report states only that:

> "Neutral conditions ($Ri_g = 0$) represent a flow in which turbulence is being generated only through the action of wind shear; buoyancy has no influence."

We have therefore decided to remove the statement and the related citation.

5. Sect. 3.9 is very disconnected from the rest of the discussion. Maybe it can be removed.

Subsection 3.9 was removed. The text defining statistical analysis tools was added to the beginning of subsection 3.8 where we think it represents a proper introduction to spectra and cospectra of turbulent quantities.

6. L464 – Please add that sonic anemometers typically measure virtual temperature as well. This physical parameter is leveraged for the estimation of the friction velocity and Obukhov length through the eddy-covariance method.

We have added the following sentences:

> When combined with independent temperature measurements, sonic anemometers can provide high-rate measurements of acoustic temperature which represents a good approximation of a virtual temperature which accounts for the water vapor in the air. By simultaneously measuring velocity components and virtual temperature, using the eddy-covariance method sonic anemometers can provide

sensible heat fluxes. Using measurements of momentum fluxes and sensible heat fluxes one can estimate the Obukhov length.

7. L559 – Add here the reference to the IEC standards, which is provided at L 585, instead.

Reference was added.

8. L655 – Provide details on the Langevin equation.

The Langevine equation and definition of terms in the equation were added:

Using highly fluctuating data at 1 Hz or higher, it could be shown that power characteristics can be defined using stochastic process modeling representing the evolution of a random value based on the Langevin equation 1.

$$\frac{dP(t)}{dt} = D_1(P, u) + \sqrt{D_2(P, u)}\Gamma(t) \tag{1}$$

Here, $P(t)$ denotes the power output, $D_1(P, u)$ is the drift coefficient, $D_2(P, u)$ is the diffusion coefficient, and $\Gamma(t)$ denotes the zero-mean Gaussian noise. The coefficients are functions of the power output $P$ and the wind speed $u$.

9. L656 – Fix references.

References were fixed.

10. L721 – Wind Energy, no need for capital letters.

Capital letters were changed to lower case.

11. Sect. 8.2.2 does not provide a clear explanation of the phenomenon described. I would suggest removing it.

Global intermittence of a stably stratified boundary layer is an important physical phenomenon that can impact turbine loads. The phenomenon is briefly described in the subsection 3.1.8 where references are provided. We therefore think that this topic should be addressed. We followed reviewer's suggestion and removed the subsection 8.2.2, but included the following text in the subsection 8.1 "Impact of Atmospheric Phenomena on Fatigue Loads."

In Section 3.7, the global intermittency phenomenon was briefly introduced. Even though these phenomena are often present in the atmosphere, only a few studies have described their impacts on wind turbine loading. Using observational data from the Long-Term Inflow and Structural Test (LIST) project, Kelley (2011) documented severe transient loading events associated with turbulent bursting events (see Fig. 15 for an example). In an LES study, Park et al. (2015) reported the presence of global intermittency in stable boundary layers. They found that these structures led to strong asymmetric forces on the rotor and, in turn, produced increased tower-top yawing moments.

12. Similarly for Sect. 8.3. The discussion is very generic and no critical information is provided.

We have removed the subsection 8.3.

**References**

Aird, J. A., Barthelmie, R. J., Shepherd, T. J., and Pryor, S. C. (2022). Occurrence of low-level jets over the eastern us coastal zone at heights relevant to wind energy. Energies, 15:445.

Emeis, S. (2014). Wind speed and shear associated with low-level jets over northern germany. Meteorologische Zeitschrift, 23(3):295–304.

Ertek, G. and Kailas, L. (2021). Analyzing a decade of wind turbine accident news with topic modeling. Sustainability, 13(22).

Kelley, N. D. (2011). Turbulence-turbine interaction: The basis for the development of the TurbSim stochastic simulator. Technical Report NREL/TP-5000-52353, National Renewable Energy Laboratory, Golden, CO.

Park, J., Manuel, L., and Basu, S. (2015). Toward isolation of salient features in stable boundary layer wind fields that influence loads on wind turbines. Energies, 8:2977–3012.

Shaw, W. J., Berg, L. K., Debnath, M., Deskos, G., Draxl, C., Ghate, V. P., Hasager, C. B., Kotamarthi, R., Mirocha, J. D., Muradyan, P., Pringle, W. J., Turner, D. D., and Wilczak, J. M. (2022). Scientific challenges to characterizing the wind resource in the marine atmospheric boundary layer. Wind Energy Sci., 7(6):2307–2334.

---

## Author Comment (AC3)

**Response to a Public Comment on WES-2025-42**
**Impact of atmospheric turbulence on performance and loads of wind turbines: Knowledge gaps and research challenges**
**Response to Etienne Cheynet**

Branko Kosović, Sukanta Basu, Jacob Berg, Larry K. Berg, Sue E. Haupt,
Xiaoli G. Larsen, Joachim Peinke, Richard J. A. M. Stevens,
Paul Veers, and Simon Watson

**Response to Etienne Cheynet**

We thank Prof. Cheynet for his comprehensive evaluation of our paper and valuable suggestions. The comments shown in blue font have been carefully considered, with our responses provided in magenta font and modified text shown in red font below. For context, unmodified text is shown in black font.

**Point 1: Definition of turbulence**

It may be worthwhile to provide a clear definition of turbulence in this section. In wind turbine design and micrometeorology, turbulence is typically understood as three-dimensional and is clearly delineated by the spectral gap; motions on the low-frequency side of this gap are generally considered "non-turbulent" However, in mesoscale meteorology, such non-turbulent motions are sometimes described as two-dimensional turbulence. These differing definitions may create confusion for readers from diverse research backgrounds. It could be useful to clarify whether the analysis follows one convention or recognizes both. This could be done by referencing established literature or explaining how each definition applies within the context of the study.

To more clearly define the scope of the review we have expanded the following sentence in the introduction to more clearly define turbulence within the context of the manuscript:

> When considering turbulence impact on wind energy we adopt a broad view of atmospheric turbulence that is not focused only on irregular, chaotic, three-dimensional, small-scale motions in an ABL, but also includes larger-scale atmospheric forcings associated with quasi-geostrophic turbulence (e.g., Charney, 1971) and mesoscale phenomena (e.g., Lilly, 1983) that modulate turbulent flows in the ABL. Considering quasi-geostrophic turbulence is motivated by resulting ABL turbulence deviating from commonly made assumptions of stationarity, homogeneity, and Gaussianity.

**Point 2: Abstract**

The sentence "Large-scale atmospheric circulations modulate the boundary layer turbulence, characterized by coherence and intermittence" appears to combine concepts that are not typically treated within the same theoretical framework. For example, coherence ($\neq$ coherent structure), a two-point statistical measure, used in signal processing, is typically applied under assumptions of stationarity and homogeneity. However, "intermittence" refers to transient events with

sometimes extreme deviations from stationarity. These concepts do not comfortably coexist, and their combination here appears arbitrary. A clearer and safer formulation might be: "Large-scale atmospheric circulations and Earth surface characteristics modulate boundary layer turbulence."

We did not modify the sentence "Large-scale atmospheric circulations modulate the boundary layer turbulence, characterized by coherence and intermittence." In atmospheric boundary layers and boundary layers in general turbulence and coherent structures coexist and they are not independent of each other. Some of the examples are convective eddies and horseshoe vortices. Therefore, coherence and coherent structures are not independent concepts. Furthermore, coherence can be defined in the context of non-stationary flow and does not require the assumption of stationarity (e.g., Xue et al. 2025, GRL, 52, e2025GL114978. https://doi.org/10.1029/2025GL114978).

**Point 3: Manuscript organisation**

The manuscript might benefit from a clearer structure based on the SGHET framework (stationary, Gaussian, homogeneous, ergodic turbulence), which has been used for wind turbine design. Sections 3.1 to 3.7 largely fall within this framework and could be presented as representing the current paradigm. Sections 3.8 and 3.9 begin to move beyond SGHET assumptions, but the transition is not clearly marked. Section 4 combines both SGHET-compatible and non-SGHET phenomena without always distinguishing between them. Section 4.2 is particularly interesting, as it addresses sub-mesoscale motions that lie outside the SGHET framework and are not accounted for in the idealized picture of a spectral gap separating microscale turbulence from mesoscale flows. This spectral gap is not always observed, suggesting potential coupling between scales. This coupling is often neglected in structural loading, but one that could be important for large offshore wind turbine design. Framing the manuscript around a progression from the SGHET framework to its limitations and potential extensions could improve clarity.

We have considered several approaches to organizing a complex topic treated in the manuscript. Each of them had some advantages and disadvantages. We agree that a possible organization of the manuscript could be by distinguishing between the stationary, Gaussian, homogeneous, ergodic turbulence (SGHET) paradigm that has been a dominant in analysis of turbulence impacts related to wind energy and non-SGHET. However, in our opinion, significant disadvantage of the proposed reorganization of the manuscript is that such organization would result in an unbalanced manuscript since a non-SGHET paradigm for wind energy applications is still in relatively early stage of the development. Instead, throughout the manuscript we emphasis the need to move beyond the SGHET framework.

**Point 4: Section 3.7 - Spectra and coherence**

Within the SGHET framework, one-point spectra and coherence are the two main turbulence statistics — and arguably the only ones needed for wind load modelling. Indeed, turbulence intensity and integral length scales can be retrieved from the wind spectra. While there has been extensive work on modelling one-point spectra over the past sixty years, the authors have chosen to focus on a limited subset.

I tested the Tchen-Mikkelsen model a few years ago but could not reproduce the published results, despite the model's simplicity and appeal. Based on a private communication with Prof. Mikkelsen over six years ago, I believe there may have been an issue with the formulation of the equations in their original paper. I am not sure whether this has since been corrected. It is worth noting that Hu et al. (2018) also reported systematic deviations from the Kaimal model in the inertial subrange when using the Tchen-Mikkelsen formulation, which, by design, should not occur.

The final paragraph of Section 3.7 introduces the Davenport coherence model, a foundational contribution that perhaps deserves its own equation number and a proper citation to Davenport (1962). The value of the decay coefficient reported in line 273 (a $\approx$ 60) appears erroneous. Panofsky and Dutton (1984) reported values more typically in the range of a $\approx$ 12–15. The decay coefficient itself depends on the velocity component, atmospheric stability and the type of separation (longitudinal, vertical, or transverse). Bowen et al. (1983) also document a dependency on the separation distance and measurement height, which was also observed at the FINO1 platform (Cheynet, 2018). Solari and Piccardo (2001) provide a helpful overview of this parameter.

It is important to recall that the Davenport model was developed specifically for microscale turbulence, and decay coefficients reported in that context should be interpreted accordingly. While applying the Davenport model to mesoscale motions is acceptable, comparing decay coefficients for mesoscale and microscale motions, as done in lines 274–276 (page 13) does not make much sense ("comparing apples and oranges").

The text was modified as suggested. Text about Tchen-Mikkelsen spectral model was omitted. The equation for the magnitude of the coherence was added as well as related Davenport (1962) reference. The text was also modified to make sure that there is no intention to compare the Davenport model parameters between fully developed 3D turbulence and quasi-geostrophic turbulence (to avoid the appearance of "comparing apples and oranges"). The discussion of coherence was expanded and now includes Davenport's model as a standalone

equation and the following text:

> Davenport (1962) estimated the parameter $a = 7$ for separation in both cross-wind and vertical directions. However, further studies demonstrated that the parameter $a$ is not constant but it depends on the atmospheric stability (e.g., Panofsky and Mizuno, 1975). While coherence analyses based on observations focused on mean wind direction, Berg et al. (2016) demonstrated how turbulence-resolving numerical simulations can be used to analyze coherence of three velocity components. They compared simulated non-Gaussian velocities to Gaussian fields and showed that their coherences are similar and found that as the separation increases the largest coherence switches from vertical to cross-wind component. While the longitudinal coherence is less important for a wind turbine design it is important for the turbine control (e.g., Schlipf et al., 2013). Thedin et al. (2023) used turbulence-resolving simulation driven by large-scale forcing derived from a mesoscale simulation to analyze coherence of three velocity components in three spatial directions and pointed to limitation of numerical simulations that do not resolve high-frequency fluctuations. For large-scale, quasi-geostrophic turbulence Vincent et al. (2013) analyzed coherence as function of separation and angle with respect to a mean wind direction using observations and mesoscale simulations. They extended a form of Davenport's coherence model to large separations that represent the coherence at mesoscale.

**Point 5: Coherence vs coherent structures**

I think it would also be helpful for the authors to clearly distinguish between the terms "coherence" and "coherent structures". Coherence is a correlation function in the frequency space. It is a concept from signal processing used as a statistical measure, for example, to characterize the spatial correlation of wind velocity fluctuations. The coherence is used, among others, to generate spatially correlated turbulent wind fields for aeroelastic codes. Coherent structures: a more abstract term used to describe organized motions in a fluid. These terms refer to fundamentally different concepts, though they are sometimes conflated in the literature.

We have tried to make the distinction between coherent structures and coherence clear throughout the manuscript. However, we do not agree that coherence can be defined only within the SHGE turbulence framework (e.g. Chatterjee and Peet, 2021, Physics of Fluids). We have also modified one of the recommendations to address this comment:

> The characterization and quantification of effects of atmospheric stability, non-homogeneity, and coherent structures on turbulence nonstationarity,  its coherence, and length scales and their impacts on the aerodynamic performance and wind turbine loads is still lacking.

**Point 6: Section 5**

Maybe this section (atmospheric turbulence observations) could also be reorganized around the concept of the SGHET framework. This section could address the following question: which sensors allow for observations that go beyond this framework?

This section discusses sonic anemometer measurements and draws a parallel between Taylor's hypothesis and the eddy-covariance method. However, this comparison may risk some misinterpretation. The eddy-covariance method does not inherently rely on Taylor's hypothesis. It computes turbulent fluxes directly from time series and does not convert temporal measurements into spatial ones. In contrast, Taylor's hypothesis is generally used to infer spatial statistics from temporal data.

Section 5 should also make a clearer distinction between profiler lidars and scanning lidars. These are different instruments with different purposes. In my experience, this distinction is often overlooked. I have worked a little with scanning lidars in complex terrain, coastal sites and offshore, with both long-range pulsed and short-range continuous waves for the study of atmospheric turbulence. In my experience, the main takeaway is that long-range scanning lidars allow for qualitative analysis of turbulence. However, such lidars would struggle to quantify turbulence, especially spectral statistics, which are more useful than integral statistics. This is due to large probe volumes and low sampling frequencies. Short-range scanning lidars (e.g., WindScanners) are more promising, but unfortunately less commonly used and their useful scanning range is limited to 150-200 m.

Thus, section 5 could address a few more important knowledge gaps and research challenges: (1) it could explain if and how remote sensing could help move beyond the SGHET framework. (2) It could highlight key limitations of scanning lidars: probe volume averaging that can be large, low sampling rate, limited reliability, high cost, reduced performance if the flow across the probe volume is heterogeneous, and the fact that they measure the along-beam component only, which complicates the analysis of 3D turbulence.

The statement about Taylor's hypothesis was modified to explicitly state that the hypothesis is invoked when temporal measurements are used to infer spatial correlations. We considered expanding the discussion about lidars, but decided that this could lead to significant expansion of the manuscript beyond its current scope.

**Point 7: On the turbulence intensity and its relevance to wind loading**

A common misconception in wind loading on structures (turbines, tower, bridges, etc...) is that the turbulence intensity is based on the standard deviation of the wind speed. For non-yawed turbines, it should be the standard deviation of the longitudinal component. The IEC 61400-1 standard itself is ambiguous on that point as it defines the turbulence intensity first based on the standard deviation of the wind speed and later on, using the longitudinal wind velocity component.

For wind loading on structures, the turbulence intensity is, fortunately, not absolutely necessary. Within the SGHET framework, only three wind statistics are needed to generate a spatially correlated wind field: the mean wind speed, the one-point velocity spectra and the coherence of turbulence. The turbulence intensity can be directly retrieved from the velocity spectra and the mean wind speed. In the Eurocode (EN 1991-1-4: Eurocode 1), the turbulence intensity is defined based on the roughness length, which makes more sense from a modelling viewpoint. However, if I remember properly, the Eurocode is only usable for ultimate limit-state design (strong wind), for which the atmosphere is assumed neutral.

Another weakness of turbulence intensity as a statistic is that it is inversely proportional to wind speed. This property is not desirable for a non-dimensional turbulence metric usable in wind loading. In my opinion, the possible over-reliance on turbulence intensity in wind turbine design is a weakness of the IEC standard. The turbulence intensity is a tricky quantity to use: it is widely used, it is easy to measure and interpret, but has multiple definitions depending on the user's background. Finally, it has limited physical meaning and is less informative than the velocity spectra. A possible alternative could be the use of standard deviation profiles, which would depend on surface roughness, atmospheric boundary layer depth and thermal stratification of the atmosphere. However, this would require new measurement techniques for validation, with sensor heights extending beyond those of traditional mast-based observations.

Based on the comment, the following sentence was added to the first paragraph of the subsection "Impact of Atmospheric Phenomena on Fatigue Loads:"

> However, the IEC 61400 standard does not define turbulence intensity consistently. First it defines it based on the wind speed and later based on the longitudinal velocity.

Also, in the second paragraph we added the following statement:

> By definition turbulence intensity is inversely proportional to wind speed and therefore under certain extreme wind conditions, such as downslope wind storms (Pehar et al., 2019), it is not a good predictor of fatigue loads.

**Point 8: On the assumption of stationarity and Gaussianity**

In non-stationary (intermittent) flows, non-Gaussianity often arises as a direct consequence of the lack of stationarity. Skewness and kurtosis are two commonly used metrics to quantify deviations from Gaussian behaviour. They rely on the assumption of stationarity. Therefore, their calculation in intermittent flows has limited physical meaning. In other words, once the flow is non-stationary, traditional statistical moments may no longer be applicable, and alternative analysis tools are needed. This raises two key questions: (1) Under which conditions do we observe stationary, non-Gaussian flows in the atmosphere, and how do they affect wind turbine loading? (2) Which tools can be used to study non-stationary atmospheric flows in the context of wind turbine design?

The second question has started being addressed in wind engineering since the 2010s, where researchers often decompose the flow into stationary and non-stationary components using tools such as empirical mode decomposition. Similar approaches could be valuable for advancing wind turbine load analysis beyond the SGHET framework. Maybe I should clarify that the field of wind engineering and wind energy are overlapping but distinct.

We disagree with the implication that intermittent flows are necessarily non-stationary. Turbulent flows characterized by intermittency (long tail probability distributions) can be stationary. This intermittency (of different turbulence properties, e.g., velocity increments) is distinguished from global intermittence observed in stably-stratified atmospheric boundary layers frequently a consequence of breaking Kelvin-Helmholtz waves. We make the distinction between turbulence intermittency and global intermittency clear at the end the subsection 3.6 "Statistical Hierarchy, Spectra, and Coherence":

> This statistical intermittency must be distinguished from the global intermittency induced by large coherent structures such as, for example, Kelvin-Helmholtz billows.

**Point 9: On the Integral length scales (section 3.6)**

Integral length scales (ILS) are useful in wind tunnel studies. For mast-based measurements, ILS are typically estimated in the streamwise (x) direction. Following Panofsky and Dutton (1984), page 176, the use of ILS in atmospheric studies should be avoided due to their lack of reliability. I tend to agree with them. In my opinion, estimates of ILS are fairly reliable for the vertical velocity component, but much less so for the horizontal components. As the authors correctly point out, this is largely due to the influence of large-scale eddies on the auto-correlation function. Obtaining better estimates would require a longer

time series, but this can conflict with the assumption of stationarity, potentially compromising the validity of the ILS estimation itself.

It can be noted that there are multiple methods to estimate the ILS, not limited to the use of the autocorrelation function. These methods can lead to quite different values. For example, in wind engineering, the so-called von Kármán spectrum uses ILS as an input parameter, which can be estimated by least-squares fitting to measured power spectral densities. Overall, I think the section could reflect more critically on the relevance of ILS for wind turbine design. Are they truly useful? If they are not reliable, should we consider alternative length scales? And if so, which ones? These are open questions, and while I don't claim to have clear answers, I believe they are important to raise.

ILSs are important in characterizing turbulence. They are parameters in IEC spectral models. They are also used in turbulence modeling and therefore, at least indirectly, relevant for wind turbine or wind farm design. We have added the following text to the end of the subsection:

> Related to sparsity of data needed to estimate ILSs is the challenge to determine them from the data. Considering these challenges, a different way to estimate relevant turbulence length scales would be beneficial. LES can provide data needed to estimate all the integral length scales. Stanislawski et al. (2023) used LES of aytime ABLs under different atmospheric stability conditions to study the effect of turbulent inflow ILSs on wind turbine loads. They found that loads increase with increasing length scales. Hodgson et al. (2025) analyzed LESs of a flow through a wind turbine array and concluded that the power output of a wind farm depends integral lengths scales of turbulent inflow.

**Point 10: Section 8.1**

Over the years, I have frequently seen authors refer to Kaimal et al. (1972) as the source of both the so-called IEC-Kaimal spectral model and the exponential decay model for coherence. However, a close reading of Kaimal et al. (1972) shows that coherence was not investigated in that study. The foundational work on turbulence coherence was conducted earlier by Panofsky and co-authors, as well as by Davenport during the 1960s and 1970s. In addition, the one-point spectral model presented in Kaimal et al. (1972) differs significantly from the version adopted in the IEC standard.

Based on the comment the text was modified as follows:

> ... the Kaimal spectral (Kaimal et al., 1972) with the exponential coherence model by Davenport (1961).

**Point 11: Lines 882–884**

The statement that the IEC Kaimal model becomes height-independent above 60 meters is "not physically realistic" might benefit from a more nuanced phrasing. This height-independence is a reasonable simplification that reflects the properties of the mixing layer. Above the surface layer, spectral characteristics often no longer scale with height, and assuming continued height dependence may be less realistic. Paradoxically, the IEC simplification may offer a more accurate representation than surface-layer spectral models that enforce height scaling throughout.

As can be seen from Figure 6 in the manuscript, the longitudinal integral length scale of the streamwise velocity component varies with height computed based on LES of a neutrally stratified ABL. This length scale is a parameter in Kaimal spectrum as outlined in IEC standard where it is prescribed as constant above 60 m. Similarly, the longitudinal integral length scale of the streamwise velocity in a stably stratified ABL also varies with height. We have modified the text to qualify the statement:

> ...this assumption is not physically realistic for modern rotors, in particular when operating in a relatively shallow stably-stratified ABL.

**Point 12: Lines 885–888**

The logical connection between the two sentences in this paragraph seems flawed. The sentence beginning with "However, wind turbines in the field are subject to atmospheric turbulence..." appears to contrast with the previous sentence describing how turbulence is simulated using stationary, Gaussian wind fields. But there is no contradiction here. Simulations based on the SGHET framework are intended to approximate atmospheric turbulence.

A formulation that corrects that issue and highlights the limits of the SGHET framework would read as "According to IEC standards, standard industry tools such as TurbSim (Kelley and Jonkman, 2007) and the Mann turbulence generator (e.g., Dimitrov et al., 2024) generate stationary homogeneous Gaussian turbulent wind fields. While these models are widely used for design and simulation, actual atmospheric turbulence experienced by wind turbines can exhibit strong nonstationarity and non-Gaussian characteristics, which may significantly affect power output, structural loading, and fatigue life."

We have rewritten the paragraph about TurbSim to more clearly distinguish coherent structures included in TurbSim from random coherent turbulence as follows:

> Working towards this goal, researchers from

 the National Renewable Eenergy Laboratory (NREL) (Jonkman, 2009) implemented in TurbSim a capability to include coherent structures that reflect the proper spatiotemporal turbulent velocity field relationships seen in instabilities associated with nocturnal boundary layer flows (e.g., breaking Kelvin-Helmholtz waves) and which are not represented well by the IEC Normal Turbulence Models. TurbSim provides the ability to efficiently generate randomized coherent turbulent structures produced by one of the non-neutral spectral models that are superimposed on the more random background turbulent field characterized by non-zero coherence .

**Point 13: Lines 897–902**

The paragraph describing TurbSim may give a misleading impression regarding its relationship to the SGHET framework. While the tool does allow for the superposition of randomized coherent turbulent structures, the underlying turbulence field is still generated within the SGHET paradigm. That is, it remains stationary, Gaussian, and homogeneous, constructed from predefined spectra and coherence functions. It is important to clarify that coherence structures are always present in TurbSim-generated fields due to the use of a coherence function; this is not unique to the added structures. Coherent structures refer to organized, persistent patterns of motion. They can be turbulent or non-turbulent motion. These structures are spatially and temporally correlated regions of the flow, e.g. vortices, shear layers, or streaks, that carry a significant portion of energy and contribute to the transport of momentum, heat, or scalars. Unlike statistical coherence, which describes correlations between signals, coherent structures are physical features within the turbulent flow.

The superposition of optional deterministic coherent structures can indeed help represent flows beyond the scope of traditional models like the IEC Normal Turbulence Model. In that sense, TurbSim offers a useful extension. Maybe the paragraph could better distinguish between the base SGHET-generated field and the additional non-SGHET structures. This clarification would help avoid confusion regarding what constitutes a true move beyond the SGHET assumptions.

We agree that coherent structures are physical feature of a flow, and they can be integral components of a turbulent flow, or they can modulate turbulent flows (e.g. tropical cyclones). The coherence is a quantitative measure characterizing turbulent flow. While coherence is commonly defined as a Fourier transform of a correlation function it is possible to define coherence of an inhomogeneous turbulent flow using different, local basis functions based on its physical manifestation through correlation functions.

**Point 14: Lines 912–923**

The discussion on quad-coherence and its impact on wind loading is interesting but appears somewhat speculative. As discussed in Cheynet et al. (2022), quad-coherence typically may have little to no influence on wind loading when linearized load models are used. Its relevance for nonlinear loading remains less clear. A recent study by Wang et al. (2025) investigated the role of out-of-phase fluctuations on wind-induced forces on floating bridges, which might offer some insight here.

It is also worth noting that specific modelling of quad-coherence is not strictly required for wind field simulation using the IEC Kaimal spectrum with exponential decay. As shown in (eqs. 28-29 Cheynet et al., 2022), quad-coherence can arise implicitly through the use of a complex exponential phase term in the coherence function. This produces vertical quad-coherence but not lateral quad-coherence unless yaw misalignment is introduced. Similarly, the uniform shear model without blockage (Mann, 1994) also exhibits vertical but not lateral quad-coherence.

Given this, it might be helpful to refine the paragraph to reflect that while quad-coherence can be present and included in some models (like the Mann model), its practical significance for wind turbine loading, particularly in standard design methodologies, remains uncertain.

The following text was added to address the comment:

> The algorithm windSim4D developed by Cheynet et al. (2022) includes quad-coherence and relaxes Taylor's frozen turbulence requirement. Cheynet et al. (2022) indicated that quad-coherence does not affect linearized wind load estimates.

**Point 15: Lines 928–934**

An important aspect to consider when combining mesoscale and microscale motions is that such models generally assume the two scales are uncoupled, relying on the presence of a clear spectral gap. If I remember well, the model by Syed and Mann (2024) follows this assumption and was developed specifically for neutral conditions; it may also be valid under stable stratification. However, under convective (unstable) conditions, the separation between scales often breaks down, and interactions between mesoscale and microscale motions become apparent. In such cases, the assumption of scale independence may no longer hold, and the applicability of this type of model becomes uncertain. I

believe this limitation should be more clearly highlighted in the discussion, especially given the relevance of convective boundary layers in wind energy research.

The following sentence was added to address the comment:

> Further developments are needed to include conditions when mesoscale peak is not pronounced such as in the presence of mesoscale convective circulations.

**Point 16: Section 8.3**

I believe this is one of the most important sections of the manuscript and may deserve further elaboration. A significant research gap in wind loading, particularly from a structural dynamics perspective, is the continued emphasis on undisturbed, upstream flow conditions. In practice, many wind turbines — especially in large farms — operate in the wake of other turbines, where the flow may not be stationary, Gaussian, or homogeneous. This calls for a move beyond the SGHET framework commonly used in wind turbine design.

Most turbulence generators assume at least statistical stationarity or spatial homogeneity. However, in wind farm wakes, these assumptions are often violated, raising the question of whether our current tools are still appropriate. There is a clear need for new turbulence characterization methods that can account for turbulence in the wake of turbines, particularly in the context of wind loading and fatigue. While I am not an expert on FAST.Farm (I could misunderstand it), it appears to offer a promising middle ground by capturing key wake dynamics and turbulence advection in a computationally efficient framework that may be suitable for load analysis.

Following a comment by one of the reviewers we have removed subsection 8.3. We agree that the topic of wake generated turbulence is important and deserves significant attention. In fact, it is treated in another paper in preparation for the Grand Challenges series. In this manuscript we focus on the atmospheric boundary layer turbulence encountered by the first row of turbines.

**Point 17: Section 8.6**

I think this section should begin by clearly restating the prevailing paradigm under which turbulence is used in wind energy, namely, the assumption of stationary, Gaussian, and homogeneous turbulence. Framing the conclusion around this paradigm would help emphasize the need to move beyond it. Specific pathways for doing so, whether through new models, observational techniques, or analytical tools, would make the call to action more concrete. As it stands, the list of recommendations feels somewhat disjointed, and organizing it around a

clear logical structure would significantly improve its clarity.

The list of recommendations is organized based on the current structure of the manuscript (and not SGHET – non-SGHET structure) to emphasis the need to consider the whole range of atmospheric scales of motion when addressing turbulence impacts on power production and wind turbine loads. Therefore, the list starts with the need to consider and observe mesoscale motions that modulate ABL turbulence followed by how they affect properties of ABL turbulence when assessing impacts on wind farms, etc.

**Point 18: Line 1185**

The phrase "mesoscale-generated turbulence associated with low-level jets, convective cells, convective rolls, and gravity waves" seems to conflate turbulence with larger-scale organized motions. In wind engineering and micrometeorology, such features are not considered turbulence themselves but rather mesoscale flow structures that can interact with or trigger turbulence under certain conditions. It may be helpful to clarify this distinction to avoid confusion between mesoscale motions and (microscale) turbulent fluctuations.

To clarify the statement the phrase "mesoscale-generated turbulence associated with low-level jets, convective cells, convective rolls, and gravity waves" was modified to:

> Mesoscale-modulated turbulence associated with low-level jets, mesoscale convective circulations (convective rolls and convective cells), and gravity waves is observed onshore and offshore.

**Point 19: Line 1190**

This line attempts to address multiple concepts: atmospheric stability, turbulence characteristics, and turbine performance, in a single sentence, which results in ambiguity. It also groups "turbulence," "coherent structures," and "length scales" together in a way that conflates concepts at different levels. Turbulence is a flow regime, coherent structures are organized motions that may or may not be part of turbulent flow, and length scales are statistical measures used to characterize turbulence. These should be more clearly distinguished, as they represent different aspects of atmospheric dynamics.

The sentence was modified as follows:

> The characterization and quantification of effects of atmospheric stability, non-homogeneity, and coherent structures on turbulence non-stationarity,  its coherence, and length scales and

their impacts on the aerodynamic performance and wind turbine loads is still lacking.

**References**

Berg, J., Natarajan, A., Mann, J., and Patton, E. G. (2016). Gaussian vs non-gaussian turbulence: impact on wind turbine loads. *Wind Energy*, 19(11):1975–1989.

Chatterjee, T. and Peet, Y. T. (2021). Streamwise inhomogeneity of spectra and vertical coherence of turbulent motions in a finite-size wind farm. *Phys. Rev. Fluids*, 6:114601.

Cheynet, E., Daniotti, N., Bogunović Jakobsen, J., Snæbjörnsson, J., and Wang, J. (2022). Unfrozen skewed turbulence for wind loading on structures. *Applied Sciences*, 12(19).

Davenport, A. G. (1961). The spectrum of horizontal gustiness near the ground in high winds. *Q. J. R. Meteorol. Soc.*, 87(372):194–211.

Davenport, A. G. (1962). The response of a slender line-like structure to a gusty wind. *Proceedings of the Institution of Civil Engineers*, 23:389–408.

Hodgson, E. L., Troldborg, N., and Andersen, S. J. (2025). Impact of freestream turbulence integral length scale on wind farm flows and power generation. *Renewable Energy*, 238:121804.

Jonkman, B. J. (2009). TurbSim User's Guide: Version 1.50. Technical Report NREL/TP-500-46198, National Renewable Energy Laboratory, Golden, CO.

Kaimal, J. C., Wyngaard, J. C., Izumi, Y., and Coté, O. R. (1972). Spectral characteristics of surface-layer turbulence. *Q. J. R. Meteorol. Soc.*, 98:563–589.

Panofsky, H. A. and Mizuno, T. (1975). Horizontal coherence and pasquill's beta. *Boundary-Layer Meteorol*, 9:247–256.

Pehar, B., Zlomušica, E., and Zalihić, S. (2019). The turbulence intensity of the wind bora. In Karabegović, I., editor, *New Technologies, Development and Application*, pages 369–376. Springer.

Schlipf, D., Cheng, P. W., and Mann, J. (2013). Model of the correlation between lidar systems and wind turbines for lidar-assisted control. *Journal of Atmospheric and Oceanic Technology*, 30(10):2233 – 2240.

Stanislawski, B. J., Thedin, R., Sharma, A., Branlard, E., Vijayakumar, G., and Sprague, M. A. (2023). Effect of the integral length scales of turbulent inflows on wind turbine loads. *Renewable Energy*, 217:119218.

Thedin, R., Quon, E., Churchfield, M., and Veers, P. (2023). Investigations of correlation and coherence in turbulence from a large-eddy simulation. *Wind Energy Science*, 8(4):487–502.

Vincent, C. L., Larsén, X. G., Larsen, S. E., and Sørensen, P. (2013). Cross-spectra over the sea from observations and mesoscale modelling. *Bound.-Layer Meteorol.*, 146:297–318.

---

## Author Comment (AC4)

We thank Prof. Arnquist for his detailed and constructive comments.

Line 24 (now 25) – Misspelling was corrected.

Figure 2 – While the suggested modification could enhance the figure, we do not think that it is an essential detail for the general information that the figure conveys.

Line 109 (line 118) – Misspelling was corrected.

Line 110 (now 117) – Here and in other instances the word "destroy" was replaced with "suppress" which is more precise than suggested "prevent," since buoyancy actively suppresses turbulence advected from another location.

Figure 3. c) – The figure is from Kalverla et al. (2017) and we cannot modify it. The scale is likely because convective cases (Ri_b > 0) are relatively weak.

Line 270 (now 367) – The statement about the spectral analysis by Sim et al. (2023) was improved.

Line 364 (now 816) – The paragraph was moved to the section "Impacts of Certain Atmospheric Phenomena," subsection "Low-level Jets" and references to Emeis (2014)

Line 366 (now 822) – The statement about the sodar observations is modified as suggested and now reads as:

"Towers typically only reach 100-200 m, and frequently sodars are ineffective in the layer near the LLJ nose due to the lack of shear produced turbulence."

Line 370 (now 448) – The statement was modified and references added and now it reads as follows:

"Several attempts to analyze spectral features associated with LLJ structure and relate them to spectra observed in canonical stably-stratified ABLs without a jet (Kaimal 1973) did not result in consistent findings. While Smedman (2004) and  Hallgren (2022) found that low frequencies of the streamwise velocity spectra associated with LLJs are suppressed, however, these results are not consistent with some other studies (Duarte 2012)."

Line 376 (now 457) – While in atmospheric sciences terminology "convection" implies unstable conditions we have modified the preceding sentence to make this clearer:

"Significant differences in temperature and humidity between relatively warmer sea surface and colder overlying air, combined with large wind shear, can result in helical roll vortices (Lemone.1973)."

Line 500 (now 582) – Mann (2017) reference was added.

Line 558 (now 660) – The acronym IEC was corrected.

Line 713 (now 840) – Misspelling was corrected.

Line 905 (now 1032 and 1034) – Mirocha et al. (2018, WES) and Sim et al. (2023, Sci. Reports) were added to support the statements.

Line 983 (now 1142) – As suggested the difference between turbulence magnitude and intensity was pointed out as follows:

"Here, we point that the turbulence magnitude defined by Equation (3) may stay constant through the row of wind turbines, while the intensity of turbulence, a non-dimensional value, would increase because the wind speed in the wake may decrease."